# SIRT4 regulates antiviral and autoimmune responses by promoting cGAS-mediated signaling pathways

Bo Yang [1,2,3,4,6], Yanjie Zhang[2,6], Saiyu Wang[2,4,6], Yufei Wu[2], Zilu Diao[2], Qunmei Zhang[5], Chen Lu[2,4], Mengyang Shen[2,4], Xuewei Zhang[1], Shujun Ma[3], Chunsheng Yang[1], Jinyong Pei[2], Hongxia Xing[1], Yinming Liang [4✉] & Jie Wang [1,2,3,4✉]

## Abstract

Cyclic guanosine monophosphate (GMP)-AMP synthase (cGAS) is a critical cytosolic DNA sensor, whose activity can be regulated by acetylation. Here, we show that nicotinamide adenine dinucleotide (NAD$^+$)-dependent lysine deacetylase SIRT4 interacts with cGAS and positively regulates innate immune responses triggered by DNA viruses or cytoplasmic DNA. Overexpression of SIRT4 inhibits HSV-1 infection, whereas knockdown of *SIRT4* has the opposite effect. Deficiency of *SIRT4*, or treatment with a SIRT4 inhibitor, impairs antiviral innate immune signaling in response to DNA viruses or cytoplasmic DNA, both in vitro and in vivo. Moreover, SIRT4 inhibitor treatment attenuates type I interferon signaling in *Trex1*-deficient cells and in peripheral blood mononuclear cells (PBMCs) from patients with systemic lupus erythematosus (SLE). Mechanistically, SIRT4 deacetylates cGAS and enhances its association with double-stranded DNA. Collectively, our study identifies SIRT4 as a positive regulator of cGAS-mediated innate immune signaling pathways, which advances the understanding of the regulation of cGAS activity.

**Keywords** Antiviral Innate Immune Responses; DNA Virus; Signal Transduction; Acetylation; Autoimmune Responses
**Subject Categories** Immunology; Post-translational Modifications & Proteolysis; Signal Transduction

## Introduction

Innate immune responses act as the first line of host defense against invading pathogens, relying on the recognition of pathogen-associated molecular patterns (PAMPs) by a set of host pattern recognition receptors (PRRs) (Tenthorey et al, 2022). Through binding with conserved PAMPs derived from various microbial pathogens, PRRs are activated and initiate a series of signaling cascades, resulting in the synthesis and secretion of type I interferons (IFN-I) and pro-inflammatory cytokines, which lead to the expression of other antiviral proteins and the full activation of antiviral immune responses (Dvorkin et al, 2024). Upon viral infection, conventional PRRs, such as Toll-like receptors (TLRs), the retinoic acid-induced gene I (RIG-I)-like receptors (RLRs), and a series of cytosolic DNA sensors, are engaged in sensing of viral DNAs, RNAs, and proteins, contributing to the restriction of viral invasion (Webb and Fernandez-Sesma, 2022).

The nucleotidyl transferase (NTase) enzyme cyclic guanosine monophosphate (GMP)-AMP synthase (cGAS), which is widely expressed in most cell types, has been identified and demonstrated to be critical for the recognition of various DNA viruses, certain retroviruses, and intracellular bacteria (Flavell and Sefik, 2024). Normally, cGAS is kept inactive in resting cells and can be activated by dsDNAs and DNA-RNA hybrids. Upon binding to double-stranded DNA (dsDNA), cGAS undergoes a conformational change to allow the synthase for CDN 2'-5' 3'-5' cyclic GMP-AMP (2'-3'-cGAMP, hereafter referred to as cGAMP), which acts as a second messenger that binds to the signaling adaptor stimulator of IFN genes (STING) (Gao et al, 2013; Sun et al, 2013). Activated STING serves as a platform to recruit and activate TANK-binding kinase 1 (TBK1) and IKKε, leading to the phosphorylation and activation of the transcription factor IFN regulatory factor 3 (IRF3) and NF-kB, thus inducing the expression of IFN-I and pro-inflammatory factors (Guey and Ablasser, 2022).

Although cGAS plays a critical role in the immune response against infections, its abnormal activation by self-DNA is a significant cause of various severe autoimmune diseases, including systemic lupus erythematosus (SLE), rheumatoid arthritis (RA), and Aicardi-Goutières syndrome (AGS) (Hu et al, 2022). SLE is a chronic autoimmune disease characterized by high levels of autoantibodies and multi-organ damage. Its pathogenesis is not yet fully understood, and effective treatments remain limited (Zucchi et al, 2023). A hallmark serological feature of SLE is the presence of autoantibodies targeting self-nucleic acids (particularly self-dsDNA) and nucleic acid-binding proteins. These autoantibodies accumulate in tissues, leading to inflammation and the production of type I interferons (IFNs) (Shrivastav and Niewold,

[1]Department of Rehabilitation Medicine, The Third Affiliated Hospital of Xinxiang Medical University, Xinxiang, China. [2]Xinxiang Key Laboratory of Inflammation and Immunology, School of Medical Technology, Xinxiang Medical University, Xinxiang, China. [3]Henan Collaborative Innovation Center of Molecular Diagnosis and Laboratory Medicine, Xinxiang Medical University, Xinxiang, China. [4]Henan Key Laboratory of Immunology and Targeted Drug, Xinxiang Medical University, Xinxiang, China. [5]Clinical Laboratory, The First Affiliated Hospital of Xinxiang Medical University, Weihui, Henan, China. [6]These authors contributed equally: Bo Yang, Yanjie Zhang, Saiyu Wang. ✉E-mail: yinming.liang@gris.org.cn; jiewang618@xxmu.edu.cn

2013). In recent years, growing evidence has linked cGAS to the pathological processes of SLE. In peripheral blood mononuclear cells (PBMCs) of SLE patients, the expression of cGAS and cGAMP is upregulated, and their levels correlate strongly with disease activity (An et al, 2017). Approximately 1–2% of SLE patients carry mutations in TREX1, a nuclease responsible for degrading cytoplasmic DNA. In mice, *Trex1* deficiency leads to the accumulation of cytoplasmic DNA, triggering spontaneous type I interferon-dependent autoimmune disease. Notably, the deletion of either cGAS or STING completely abolishes all detectable pathological and molecular phenotypes in these *Trex1*-deficient mice (Gao et al, 2015). In addition, the activation of cGAS-STING signaling is implicated in ageing-related inflammation and neurodegeneration (Gulen et al, 2023).

Considering the essential role of cGAS in antiviral responses, autoimmunity, neurodegeneration, and ageing-related decline, it is not surprising that the ligand-binding ability, activity and stability of cGAS are tightly regulated by various posttranslational modifications (PTMs) to avoid aberrant activation (Zahid et al, 2020). To date, numerous post-translational mechanisms, including proteolysis, methylation, acetylation, glutamylation, ubiquitylation, sumoylation and phosphorylation, have been described, and certain enzymes catalyzing these PTMs have been identified to play a role in cGAS-mediated immune responses (Hong et al, 2022; Zahid et al, 2020).

Recently, accumulating evidence indicates that acetylation plays a critical role in regulating the activity of cGAS (Hong et al, 2022). It has been demonstrated that the acetylation at Lys384, Lys394, or Lys414 of cGAS suppresses cGAS activity, and that histone deacetylases (HDACs), such as HDAC1 and HDAC3, are involved in the activation of cGAS in response to DNA challenges by deacetylating cGAS (Dai et al, 2019). Conversely, the lysine acetyltransferase 5 (KAT5) catalyzes the acetylation of cGAS in its N-terminal domain at multiple lysine residues, including K47, K56, K62, and K83, thereby promoting the DNA-binding ability of cGAS (Song et al, 2020). In addition, acetylation can also regulate the cGAS activity through the modulation of cGAS binding proteins. For example, SIRT2 deacetylates G3BP1 at K257, K276, and K376, which disrupts cGAS-G3BP1 interaction, and thereby inhibits the DNA binding ability and droplet formation of cGAS (Li et al, 2023). Subsequently, Barthez et al reported that SIRT2 interacts with cGAS directly, deacetylates cGAS, and suppresses ageing-associated cGAS activation and inflammation (Barthez et al, 2025).

Sirtuins (SIRTs), a family of nicotinamide adenine dinucleotide (NAD⁺)-dependent lysine deacetylases, comprise SIRT1-7 in mammals. Increasing evidence indicates that SIRTs are involved in numerous biological processes, such as inflammation, metabolism, oxidative stress, mitochondrial function, DNA repair, and apoptosis, and that SIRT modulators are considered potential therapeutic strategies for many diseases (Nandave et al, 2023; Wu et al, 2022). Several SIRTs have been reported to be involved in the cGAS-STING signaling pathway, including SIRT2, SIRT3, and SIRT6 (Guo et al, 2024; Li et al, 2023; Simon et al, 2019; Zhou et al, 2024). Among the seven SIRTs, SIRT3, SIRT4, and SIRT5 are mainly localized in the mitochondria (Li et al, 2018). A well-known function of SIRT4 is its regulation of insulin secretion through interaction with glutamate dehydrogenase (GDH), insulin-degrading enzyme (IDE) and the ADP/ATP carrier proteins

(ANT2 and ANT3) (Zaganjor et al, 2017). In addition, SIRT4 has also been identified as an important regulator of lipid homeostasis, amino acid catabolism, nucleotide metabolism, intestine fibrosis, aerobic glycolysis, and ferroptosis (Hu et al, 2019; Li et al, 2018; Liu et al, 2023; Tucker et al, 2024; Xue et al, 2023; Zaganjor et al, 2021). Early studies suggested that among the seven members of the SIRT family, SIRT1, SIRT2, and SIRT3 are robust deacetylases, while SIRT6 and SIRT7 also exhibit deacetylase activity. SIRT5 primarily catalyzes lysine desuccinylation, demalonylation, and deglutarylation. In contrast, SIRT4 lacks detectable histone deacetylase activity but possesses substrate-specific deacetylase activity (Kumar and Lombard, 2017). The absence of robust enzymatic activity has posed challenges for studying the biological functions or developing modulators of SIRT4 (Li et al, 2018). However, recent studies have increasingly reported that SIRT4 can regulate critical physiological and pathological processes by directly deacetylating specific target proteins (Lv et al, 2025; Yu et al, 2025; Zhang et al, 2022). Moreover, specific inhibitors of SIRT4 have been identified, which sheds light on the functional study of SIRT4 and suggesting its potential as a novel drug target (Pannek et al, 2024). Nevertheless, the role of SIRT4 in antiviral innate immune responses remains unclear to date.

In this study, our findings suggest that SIRT4 is a critical positive regulator of antiviral innate immune responses against DNA viruses and cytosolic DNA. We demonstrate that SIRT4 interacts with cGAS and deacetylates cGAS directly, thereby promoting the binding of cGAS with cytosolic DNA and the generation of cGAMP. *SIRT4* deficiency impairs DNA virus- or cytosolic DNA-triggered signaling. Additionally, treatment with a SIRT4 inhibitor attenuated the type I IFN signaling response in DNA virus-infected cells, as well as in *Trex1*-deficient cells and the PBMCs from SLE patients. Overall, these observations reveal a novel role of SIRT4 in cGAS-mediated signaling pathway and provide a potential therapeutic strategy for modulating antiviral and autoimmune responses.

# Results

## SIRT4 interacts with cGAS

Because acetylation contributes to the inhibition of cGAS activity and cGAS is deacetylated in response to DNA treatment (Dai et al, 2019), we wondered whether proteins of the Sirtuin family were involved in regulating cGAS acetylation and the cGAS-mediated antiviral response. To investigate the role of Sirtuin family proteins in the DNA virus-induced signaling pathway, the potent Sirtuin inhibitor NAM, was used to treat HSV-1-infected PMA-THP1 cells. The results indicated that cGAS acetylation was increased upon NAM treatment (Fig. 1A), suggesting that Sirtuin family proteins were involved in regulating cGAS acetylation during DNA virus stimulation. To identify which member of the Sirtuin family was responsible for cGAS deacetylation, we screened for Sirtuin family proteins that interact with cGAS by co-immunoprecipitation. As shown in Fig. 1B,C and Appendix Fig. S1A, SIRT4 co-immunoprecipitated with cGAS. It appeared that under unstimulated conditions, SIRT2 interacted only very weakly to endogenous cGAS, but after HSV-1 stimulation, this binding became even less detectable (Appendix Fig. S1A).

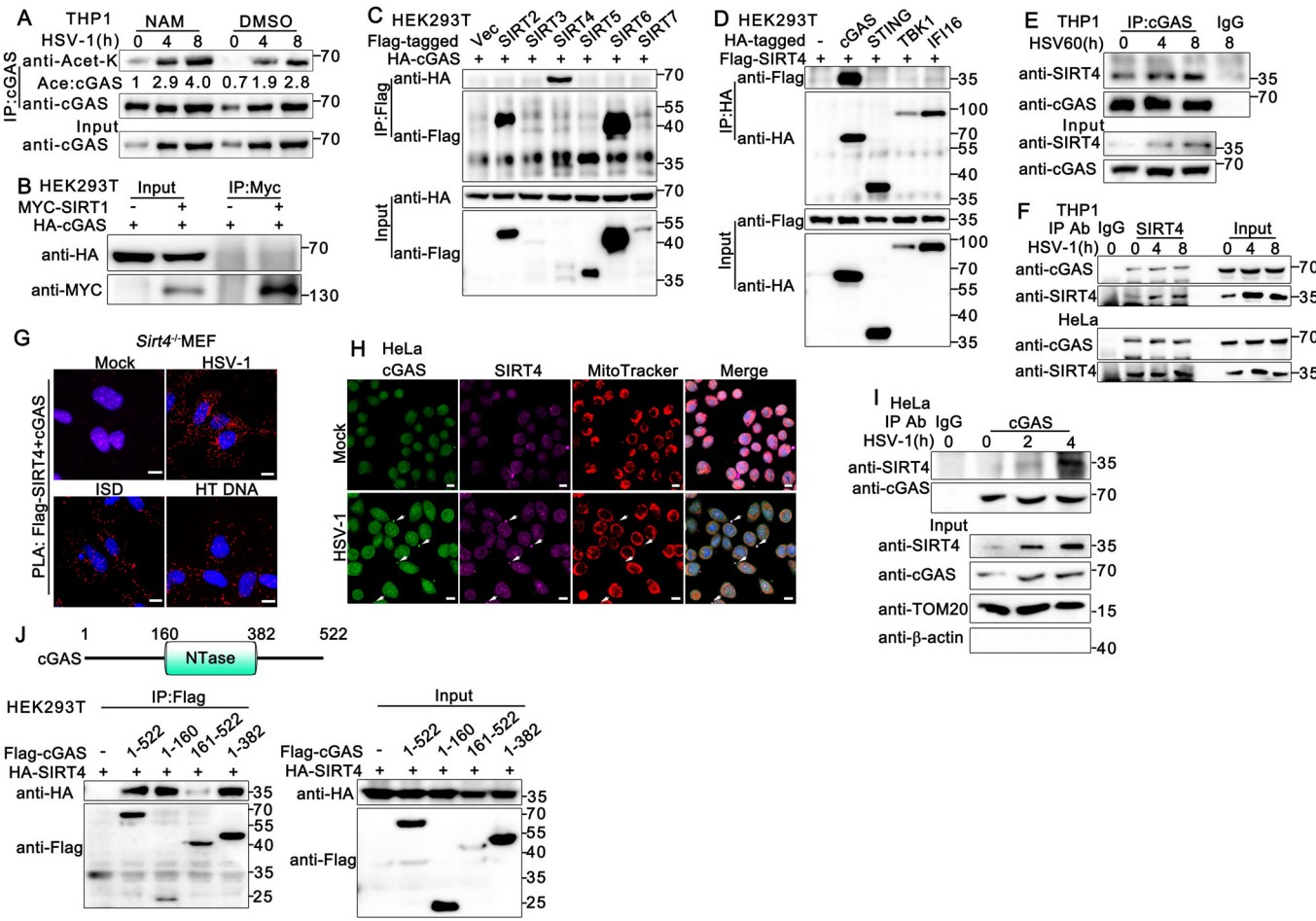

**Figure 1. SIRT4 interacts with cGAS.**

(A) PMA-THP1 cells were infected with HSV-1 (MOI = 1) for the indicated periods, using NAM (5 mM) or DMSO as a control. Cells were then lysed and subjected to immunoprecipitation (IP) and immunoblot (IB) analysis. A representative immunoblot is shown. (B–D) HEK293T cells were transfected with various combinations of plasmids as indicated. After 24 h, immunoprecipitation (IP) and immunoblot (IB) analysis were performed. A representative immunoblot is shown. (E) PMA-THP1 cells were transfected with HSV60 (1 μg/ml) for the indicated periods. Cell lysates were then subjected to immunoprecipitation (IP) and immunoblot (IB) assays as indicated. A representative immunoblot is shown. (F) PMA-THP1 (top) or HeLa (bottom) cells were infected with HSV-1 (MOI = 1) for the indicated periods. Cell lysates were then subjected to immunoprecipitation (IP) and immunoblot (IB) assays as indicated. A representative immunoblot is shown. (G) Sirt4-deficient MEFs were transfected with Flag-SIRT4 for 24 h, and then infected with HSV-1 (MOI = 1), transfected with ISD (1 μg/ml) or HT DNA (1 μg/ml) for 8 h, or left unstimulated. In situ PLA assays were performed to examine the co-localization of SIRT4 and cGAS. SIRT4-cGAS complex, red; nuclei, blue. Scale bars, 10 μm. (H) HeLa cells were infected with HSV-1 (MOI = 1) for 2 h. Immunofluorescence assays were performed using anti-cGAS (green) and anti-SIRT4 (purple). Nuclei were stained with DAPI. Mitochondria were stained with MitoTracker. Scale bar, 10 μm. Arrows indicate cGAS puncta. (I) HeLa cells were infected with HSV-1 (MOI = 1) for the indicated periods. Mitochondria isolated from cells were subjected to immunoprecipitation (IP) and immunoblot (IB) assays as indicated. A representative immunoblot is shown. (J) Schematic diagram of full-length cGAS (top). NTase, nucleotidyl transferase domain. HEK293T cells were transfected with the indicated plasmids. At 24 h after transfection, immunoprecipitation (IP) and immunoblot (IB) assays were performed as indicated (bottom). A representative immunoblot is shown. Source data are available online for this figure.

We noticed that SIRT4 expression levels were increased upon HSV-1 infection or HSV60 transfection (Fig. EV1A), indicating that SIRT4 might be involved in DNA virus infection. Co-immunoprecipitation assays also suggested that SIRT4 specifically interacted with cGAS, but not with other key molecules in the cGAS signaling pathway, such as STING, TBK1, and IFI16 (Fig. 1D). Endogenous SIRT4-cGAS interaction was detected by co-immunoprecipitation assays in PMA-THP1 and HeLa cells following HSV-1 infection or HSV60 transfection (Fig. 1E,F). Co-localization between SIRT4 and endogenous cGAS was observed in PLA assays (Fig. 1G). In addition, SIRT4 and cGAS were both detected in mitochondria, and SIRT4 co-localized and co-immunoprecipitated with cGAS in the mitochondrial fraction, consistent with the report that SIRT4 is a known mitochondrial protein (Li et al, 2018) (Fig. 1H,I; Appendix Fig. S1B,C). Confocal imaging also revealed that SIRT4 and cGAS could form puncta after HSV-1 infection (Fig. 1H), consistent with previously reported results indicating that cGAS exhibits liquid-liquid phase separation (LLPS) with DNA (Du and Chen, 2018; Shi et al, 2022). Next, we generated several truncated mutants of cGAS, and the 1-160aa region of cGAS was suggested to be important for the SIRT4-cGAS interaction (Fig. 1J). Taken together, our findings suggest that SIRT4 interacts with cGAS.

## SIRT4 positively regulates antiviral innate immune responses against DNA viruses

Given the essential role of cGAS in host defense against DNA viruses, we next investigated the effect of SIRT4 on HSV-1 infection. SIRT4 was overexpressed in HaCaT cells, and the role of SIRT4 in antiviral innate immune responses was evaluated. As shown in Fig. EV1B,C, SIRT4 overexpression promoted HSV-1- or HSV60-induced activation of TBK1, and IRF3. Consistently, the overexpression of SIRT4 markedly potentiated HSV-1-triggered transcription of downstream antiviral genes including *IFNB*, *CXCL10*, *IFIT1*, and *IL-6*, whereas the production of viral protein UL30 was reduced by SIRT4 transfection (Fig. EV1D), indicating that exogenous SIRT4 positively regulated HSV-1-induced innate immune responses. However, SIRT4 overexpression did not significantly affect the response triggered by cGAMP (Appendix Fig. S1D–F).

Next, we explored the role of endogenous SIRT4 in HSV-1 infection. Three pairs of siRNA (S1, S2, and S3) specifically targeting *SIRT4* were used to inhibit SIRT4 expression. Immunoblot and real-time PCR assays indicated that all three siRNA pairs effectively suppressed SIRT4 expression, and the maximal inhibition was observed with S2 transfection (Figs. 2A and EV1E). Thus, S2 and S3 were used in the following experiments. When transfected into PMA-THP1 cells, both S2 and S3 promoted HSV-1 infection and inhibited HSV-1- or HSV60-induced production of IFN-β and TNF-α at the protein level (Figs. 2B,C and EV1F). Consistently, *SIRT4* knockdown by S2 inhibited HSV-1- or exogenous DNA-induced production of *IFNB*, *CXCL10*, and *TNF* at the mRNA level (Fig. EV1G,H). In addition, after HSV-1 infection or HSV60 transfection, compared to control cells, *SIRT4*-silenced cells exhibited decreased activation of TBK1, IRF3 and p65 (Fig. 2D and EV1I). Furthermore, compared to cells transfected with a scrambled control (SC), S2-transfected cells showed reduced nuclear translocation of IRF3 and p65 upon HSV-1 infection (Fig. EV1J). Overall, these results suggest that SIRT4 positively regulates antiviral innate immune responses against DNA viruses and exogenous DNA.

## SIRT4 deficiency reduces antiviral innate immune responses against DNA viruses in PMA-THP1 cells

To confirm the role of SIRT4 in antiviral host defense, we generated *SIRT4*-deficient PMA-THP1 cells using the CRISPR/Cas9 system. As shown in Fig. 2E, SIRT4 expression was markedly reduced in *SIRT4*-deficient PMA-THP1 cells. Plaque assay analysis demonstrated that *SIRT4* deficiency promoted HSV-1 infection, but did not affect VSV infection (Fig. 2F). Consistently, using GFP-HSV-1 infection, immunoblot and fluorescence assays further indicated that SIRT4 inhibited HSV-1 infection (Fig. 2G,H). Upon HSV-1 infection or cytoplasmic DNA transfection, compared to control wild-type cells, *SIRT4*-deficient PMA-THP1 cells exhibited impaired innate immune responses, including reduced production of type I IFN, pro-inflammatory cytokines, and IFN induced antiviral proteins, as well as diminished activation of TBK1, IRF3, and p65 (Figs. 2I,J and EV2A–E), but showed no effect on cGAMP-induced activation of innate immune responses (Fig. EV2E). Taken together, these data suggest that *SIRT4* deficiency impairs DNA

virus or cytoplasmic DNA-triggered innate immune responses in PMA-THP1 cells.

To further investigate the effect of SIRT4 on virus-triggered innate immune responses, we explored the genes and pathways regulated by SIRT4 using RNA-sequencing analysis. In HSV-1-infected PMA-THP1 cells, differential expression analysis identified 248 upregulated genes (log2 > 1) and 674 down-regulated genes (log2 < −1) in response to *SIRT4* deficiency (Appendix Fig. S1G). Subsequently, Gene-set-enrichment analysis (GSEA) indicated that the core-enriched genes were related to cytosolic DNA sensing-related pathways (Appendix Fig. S1H). In addition, differential expression analysis further showed that *SIRT4* deficiency led to the downregulation of many type I IFN-induced IFN-stimulated genes (ISGs) and pro-inflammatory cytokines upon HSV-1 infection (log2 ≥ 1) (Appendix Fig. S1I).

Overall, our findings suggest that *SIRT4* deficiency impairs antiviral responses in PMA-THP1 cells.

## SIRT4 protects mice from HSV-1 infection

Next, we examined the effect of *SIRT4* deficiency on HSV-1 infection in a mouse model. *Sirt4*-deficient mice were generated using the CRISPR/Cas9 system (Appendix Fig. S2A–C). As shown in Fig. 3A,B, after HSV-1 infection, compared to wild-type group, *Sirt4*-deficient mice exhibited a significantly lower survival rate when the mice were challenged with a lethal dose of HSV-1 treatment, and greater weight loss following a low-dose of HSV-1 infection. When we examined the antiviral innate immune responses, ELISA analysis showed that *Sirt4* deficiency significantly reduced HSV-1-triggered secretion of IFN-β, and IL-6 in both serum and bronchoalveolar lavage fluids (BALF) (Fig. 3C,D). Consistently, impaired innate immune responses were observed in multiple organs of *Sirt4*-deficient mice compared to wild-type mice after HSV-1 infection (Fig. 3E–G; Appendix Fig. S2D). In addition, real-time PCR assays indicated that *Sirt4* deficiency enhanced HSV-1 infection, as shown by increased HSV-1 genomic DNA copies in these organs (Fig. 3F,G). Furthermore, H&E and CD68 staining revealed that *Sirt4*-deficient mice exhibited more severe tissue damage and macrophage infiltration in the lungs than wild-type mice in response to HSV-1 infection (Fig. 3H; Appendix Fig. S2E). However, we observed no significant difference in innate host defense between wild-type and *Sirt4*-deficient mice following VSV infection (Fig. 3I; Appendix Fig. S2F–H), suggesting that SIRT4 does not regulate RNA virus-triggered antiviral innate immune responses in mice. Taken together, our findings suggest that *Sirt4* deficiency impairs DNA virus-induced innate immune responses and promotes DNA virus infection.

## Sirt4 deficiency reduces antiviral innate immune responses against DNA viruses in primary cells

To further investigate the protective role of SIRT4 during HSV-1 infection in mice, we isolated and cultured bone marrow-derived macrophages (BMDMs), bone marrow-derived dendritic cells (BMDCs), or mouse embryonic fibroblasts (MEFs) from wild-type and *Sirt4*-deficient mice and examined the effect of *Sirt4* deficiency on host defense against DNA viruses and cytosolic DNA. In HSV-1-infected or exogenous cytosolic DNA-stimulated BMDMs, *Sirt4* deficiency led to reduced IFN-β secretion

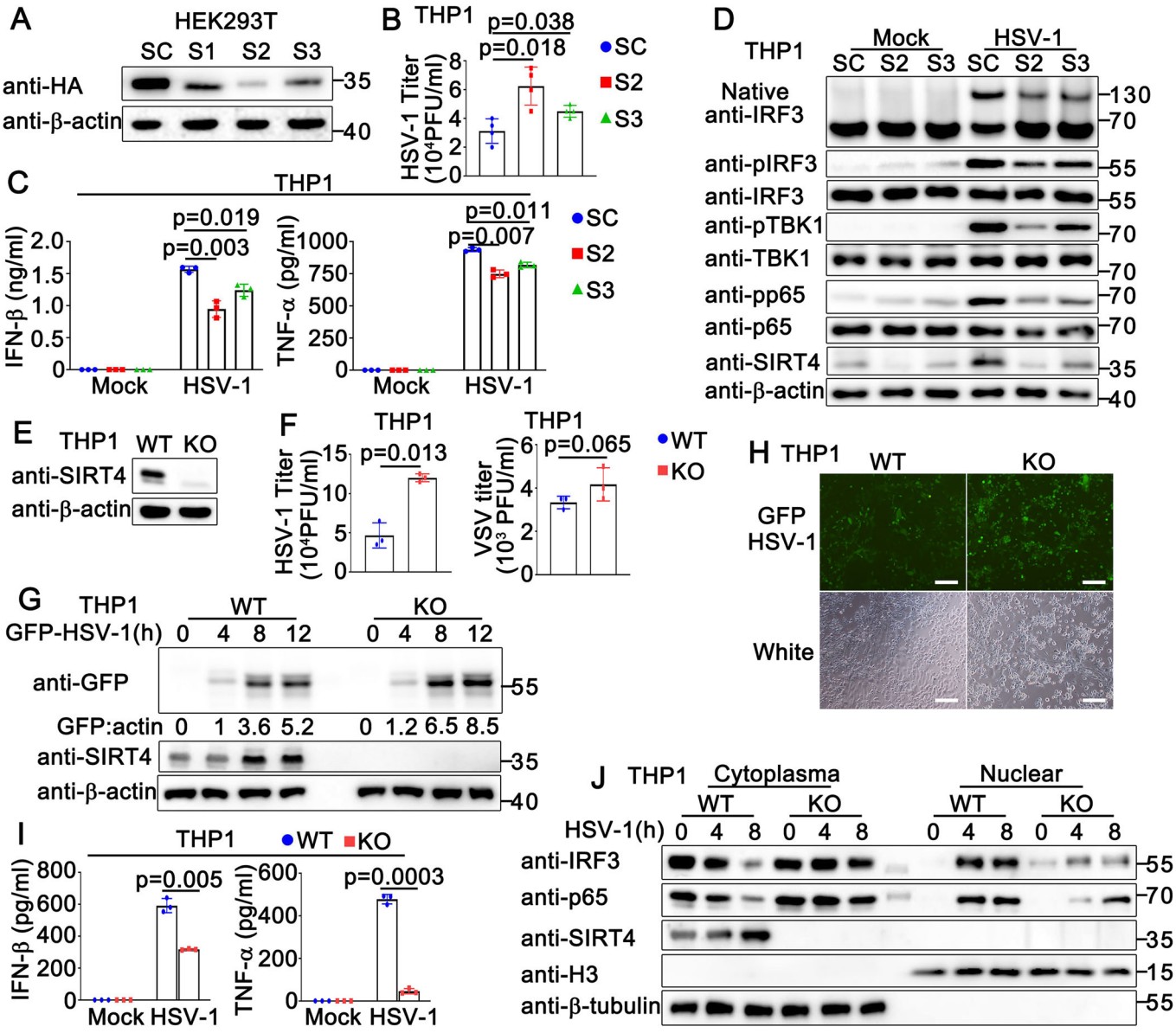

**Figure 2. SIRT4 positively regulates antiviral innate immune responses against DNA viruses.**

(A) HEK293T cells were transfected with HA-SIRT4 and then transfected with control siRNA (SC) or *SIRT4*-specific siRNA (S1, S2, and S3). At 24 h after transfection, cells were lysed for immunoblot assays. A representative immunoblot is shown. (B) PMA-THP1 cells were transfected with control siRNA (SC) or *SIRT4*-specific siRNA (S2 and S3). At 24 h after transfection, cells were infected with HSV-1 (MOI = 1) for 24 h. HSV-1 titers were determined by standard plaque assays. Results are mean ± SD, n = 4 biological replicates. Unpaired *t* test was used for statistical analysis. (C) PMA-THP1 cells were transfected with control siRNA (SC) or *SIRT4*-specific siRNA (S2 and S3). At 24 h after transfection, cells were infected with HSV-1 (MOI = 1) for 24 h. Supernatants were collected and analyzed by ELISA assays. Results are mean ± SD, n = 3 biological replicates. Unpaired *t* test was used for statistical analysis. (D) PMA-THP1 cells were transfected with control siRNA (SC) or *SIRT4*-specific siRNA (S2 and S3). At 24 h after transfection, cells were infected with HSV-1 (MOI = 1) for 8 h. Cells were then lysed for Native PAGE and SDS-PAGE analysis. Both Native PAGE and SDS-PAGE samples originate from the same experiment and were processed in parallel. A representative immunoblot is shown. (E) Wild-type (WT) and *SIRT4*-deficient (KO) PMA-THP1 cells were analyzed by immunoblot assays. A representative immunoblot is shown. (F) Wild-type (WT) and *SIRT4*-deficient (KO) PMA-THP1 cells were infected with HSV-1, or VSV (MOI = 1) for 24 h. HSV-1 or VSV titers were determined by standard plaque assays. Results are mean ± SD, n = 3 biological replicates. Unpaired *t* test was used for statistical analysis. (G, H) Wild-type (WT) and *SIRT4*-deficient (KO) PMA-THP1 cells were infected with HSV-1 (MOI = 1) for 24 h. Immunoblot (G) or fluorescence (H) assays were performed. Nuclei were stained with DAPI. Scar bars, 500 μm. Both Native PAGE and SDS-PAGE samples originate from the same experiment and were processed in parallel. A representative immunoblot (G) or image (H) is shown. (I) Wild-type (WT) and *SIRT4*-deficient (KO) PMA-THP1 cells were infected with HSV-1 (MOI = 1) for 24 h. Supernatants were collected and analyzed by ELISA assays. Results are mean ± SD, n = 3 biological replicates. Unpaired *t* test was used for statistical analysis. (J) Wild-type (WT) and *SIRT4*-deficient (KO) PMA-THP1 cells were infected with HSV-1 (MOI = 1) for the indicated periods. Cells were then fractionated into cytosolic and nuclear fractions. Immunoblot assays were performed as indicated. A representative immunoblot is shown. Source data are available online for this figure.

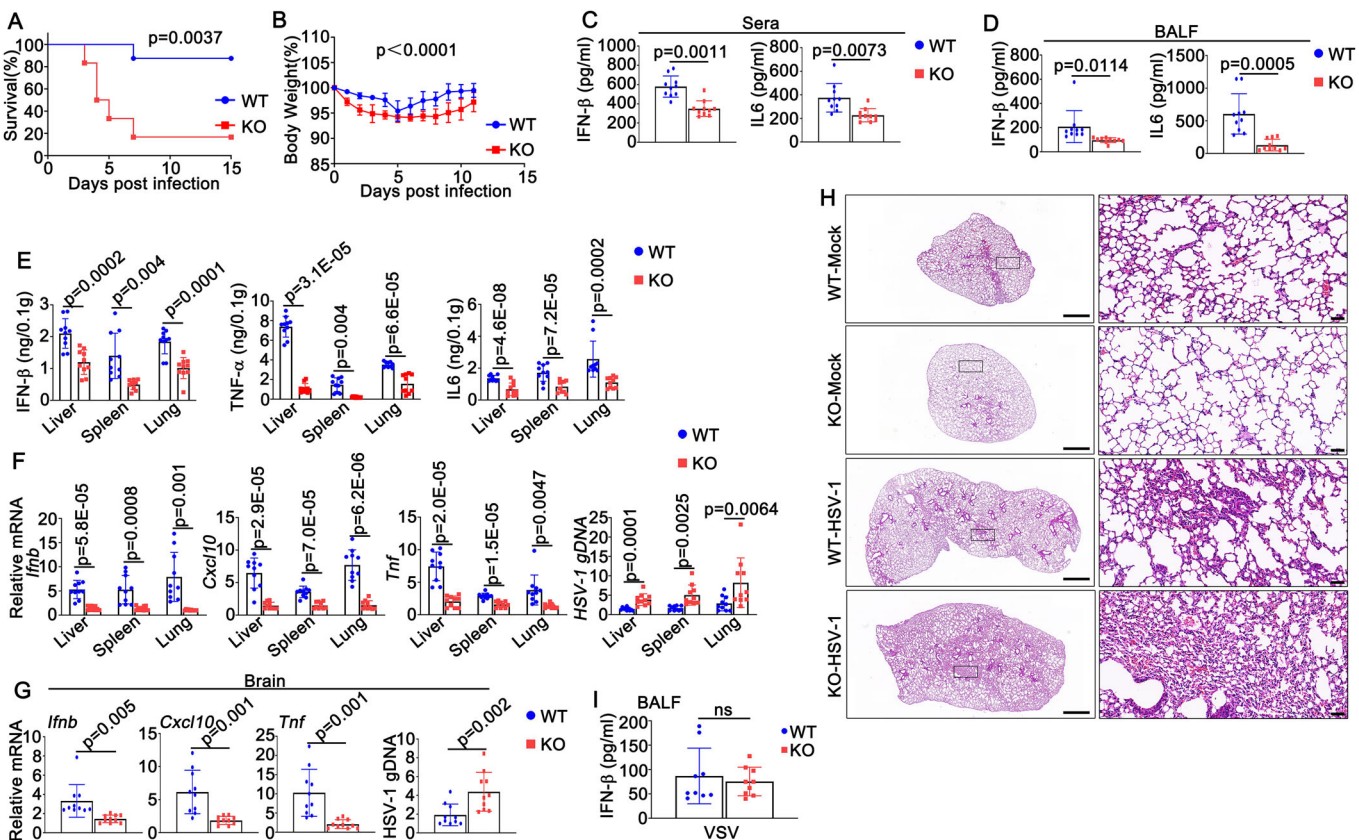

**Figure 3. SIRT4 protects mice from HSV-1 infection.**

(**A**) Sex and age-matched wild-type (WT) ($n = 8$) and *Sirt4*-deficient (KO) ($n = 6$) mice (8-week-old) were intravenously infected with HSV-1 ($2 \times 10^7$ PFU). The survival of these mice was monitored for 15 days. Each drop in the survival curve represents a death event. Log-rank (Mantel–Cox) test was used for statistical analysis. (**B**) Sex and age-matched wild-type (WT) ($n = 6$) and *Sirt4*-deficient (KO) ($n = 5$) mice (8-week-old) were intravenously infected with HSV-1 ($1 \times 10^7$ PFU). The body weight loss was monitored for 12 days. Two-way ANOVA was used for statistical analysis. (**C**) Levels of IFN-β, and IL-6 in serum from wild-type (WT) and *Sirt4*-deficient (KO) mice ($n = 10$, 8-week-old) were measured by ELISA 6 h after intravenous infection with HSV-1 ($1 \times 10^7$ PFU). Results are mean ± SD. Each data point represents one mouse. Unpaired *t* test was used for statistical analysis. (**D**) Wild-type (WT) and *Sirt4*-deficient (KO) mice ($n = 10$, 8-week-old) were intranasally infected with HSV-1 ($1 \times 10^7$ PFU) for 24 h. Bronchoalveolar lavage fluid (BALF) was collected and analyzed by ELISA. Results are mean ± SD. Each data point represents one mouse. Unpaired *t* test was used for statistical analysis. (**E, F**) Wild-type (WT) and *Sirt4*-deficient (KO) mice ($n = 10$) were intravenously infected with HSV-1 ($1 \times 10^7$ PFU) for 24 h. Lungs, livers, and spleens were then harvested and analyzed by ELISA (**E**) or real-time PCR (**F**). Results are mean ± SD. Each data point represents one mouse. Unpaired *t* test was used for statistical analysis. (**G**) Wild-type (WT) and *Sirt4*-deficient (KO) mice ($n = 10$) were intravenously infected with HSV-1 ($1 \times 10^7$ PFU) for 4 days. Brains were then harvested and analyzed by real-time PCR assays. Results are mean ± SD. Each data point represents one mouse. Unpaired *t* test was used for statistical analysis. (**H**) Sex and age-matched wild-type (WT) and *Sirt4*-deficient (KO) mice were intravenously infected with HSV-1 ($1 \times 10^7$ PFU) for 24 h. Lung sections were analyzed by H&E staining. Scale bars, 1000 μm (left), 50 μm (right). One representative image for each group is shown. (**I**) Wild-type (WT) and *Sirt4*-deficient (KO) mice ($n = 9$, 8-week-old) were intranasally infected with VSV ($5 \times 10^7$ PFU) for 24 h. BALFs were collected and analyzed by ELISA. Results are mean ± SD. Each data point represents one mouse. Unpaired *t* test was used for statistical analysis. Source data are available online for this figure.

(Fig. EV3A,B). Similar results were observed from BMDCs (Appendix Fig. S3A,B). Consistent with these observations, *Sirt4*-deficient BMDMs exhibited impaired production of *Ifnb*, *Cxcl10*, *Ifit1*, and *Tnf* in response to HSV-1 infection or exogenous DNA transfection, including HSV60, VACV70, ISD, and poly(dA:dT) (Fig. EV3C,D; Appendix Fig. S3C). In addition, compared with wild-type control cells, the activation of innate immune responses in *Sirt4*-deficient BMDMs or BMDCs was significantly decreased after HSV-1 infection or exogenous DNA transfection, as indicated by the phosphorylation of TBK1, IRF3, and p65, and decreased formation of IRF3 dimers (Fig. EV3E–G; Appendix Fig. S3D–F).

Considering that the innate immune responses also occur in non-immune cells, we next examined antiviral host defense against DNA viruses and cytosolic DNA in MEFs. As expected, we found

that *Sirt4* deficiency reduced the production of type I IFN, inflammatory cytokine, and IFN-stimulated genes (ISGs), as well as the activation of antiviral innate immune signaling pathways in response to HSV-1 infection or cytosolic DNA stimulation (Fig. 4A–F). However, in both human and mouse cells, cGAMP- or RNA virus-triggered innate immune responses were not significantly affected by loss of SIRT4 expression (Figs. 4A,F,G, EV2E, and EV3B,D,G; Appendix Figs. S1D–F and S3C,D), suggesting that SIRT4 specifically influences DNA virus-induced signaling pathway and likely acts upstream of cGAMP. In addition, the impairment of antiviral innate immune responses activation caused by *Sirt4* deficiency was reversed by SIRT4 transfection (Fig. 4H), further confirming the positive regulatory role of SIRT4 in DNA virus-triggered signaling pathways.

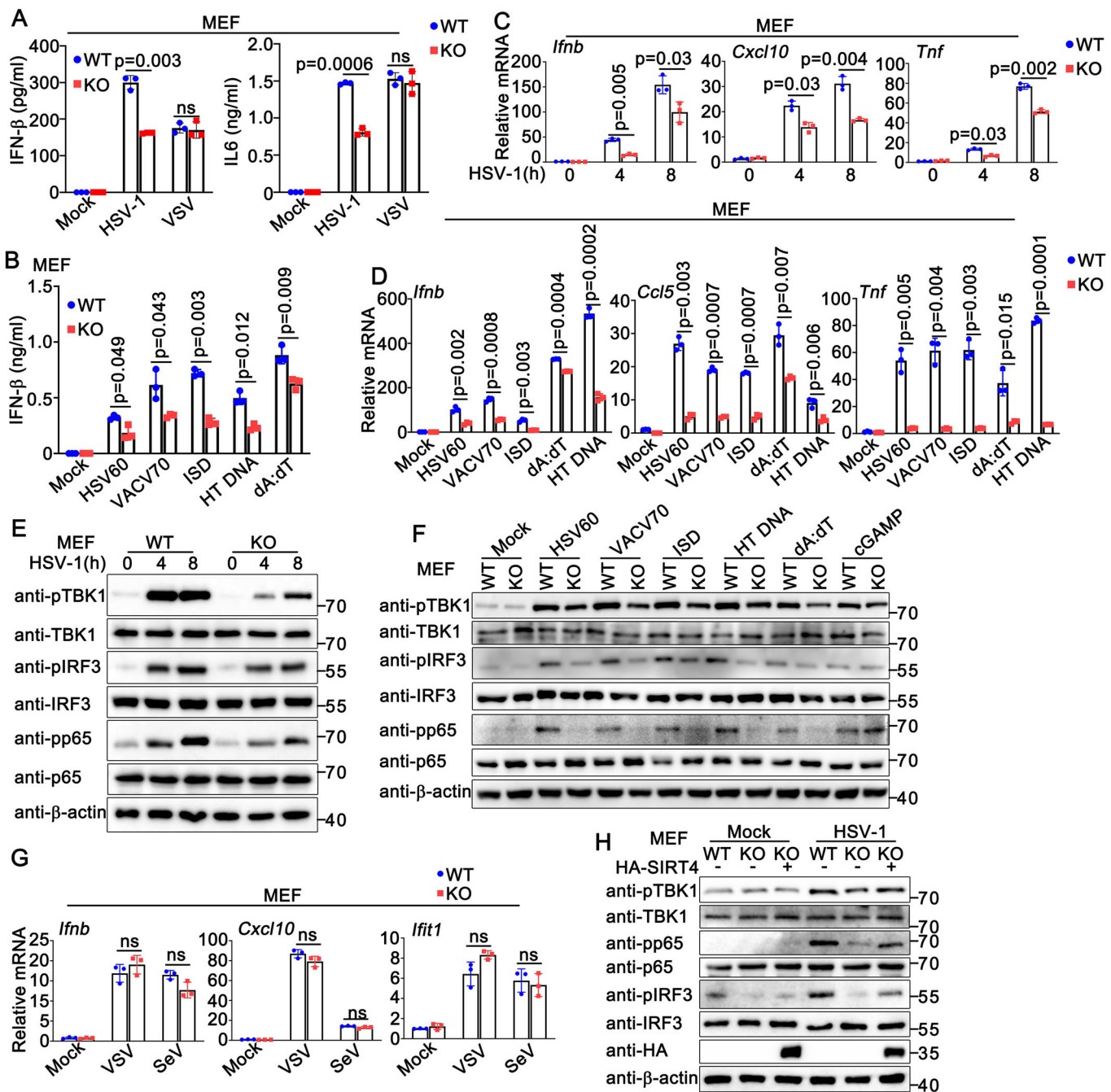

Taken together, our observations indicate that *Sirt4* deficiency impairs host innate immune responses triggered by DNA viruses or cytosolic DNA, but not by cGAMP or RNA viruses.

## SIRT4 promotes the activation of cGAS

Considering our findings that SIRT4 interacted with cGAS, we first explored whether the role of SIRT4 in innate immunity is dependent on cGAS. We examined the effect of SIRT4 on the activation of innate immune responses in wild-type or *cGAS*-deficient HeLa cells. As shown in Fig. EV4A,B, upon HSV-1 infection, SIRT4 overexpression or knockdown modulated the

activation of TBK1, IRF3, and p65 in wild-type HeLa cells, but not in *cGAS*-deficient HeLa cells. Consistently, in *cGAS*-deficient HeLa cells, no significant difference was observed in the production of type I IFN and pro-inflammatory cytokines after SIRT4 over-expression or knockdown in response to HSV-1 infection (Fig. EV4C,D). It has been reported that SIRT2 negatively regulates cGAS activation by deacetylating G3BP1 (Li et al, 2023). We therefore investigated whether the effect of SIRT4 depends on SIRT2. Results from immunoblot analysis demonstrated that *Sirt4* knockdown still inhibited the activation of innate immune signaling in *Sirt2*-deficient MEFs (Fig. EV4E), suggesting that the role of SIRT4 in HSV-1-induced signaling is independent of SIRT2.

**Figure 4.  *Sirt4* deficiency reduces antiviral innate immune responses against DNA viruses in MEFs.**

(A) Wild-type (WT) and *Sirt4*-deficient (KO) MEFs were infected with HSV-1 (MOI = 1) or VSV (MOI = 1) for 24 h. Supernatants were collected and analyzed by ELISA. Results are mean ± SD, n = 3 biological replicates. Unpaired *t* test was used for statistical analysis. (B) Wild-type (WT) and *Sirt4*-deficient (KO) MEFs were transfected with HSV60 (1 μg/ml), VACV70 (1 μg/ml), ISD (1 μg/ml), HT DNA (1 μg/ml), or poly(dA:dT) (1 μg/ml) for 24 h. Supernatants were collected and analyzed by ELISA. Results are mean ± SD, n = 3 biological replicates. Unpaired *t* test was used for statistical analysis. (C) Wild-type (WT) and *Sirt4*-deficient (KO) MEFs were infected with HSV-1 (MOI = 1) for the indicated periods. Cells were then lysed for real-time PCR assays. Results are mean ± SD, n = 3 technical replicates. Unpaired *t* test was used for statistical analysis. (D) Wild-type (WT) and *Sirt4*-deficient (KO) MEFs were transfected with HSV60 (1 μg/ml), VACV70 (1 μg/ml), ISD (1 μg/ml), poly(dA:dT) (1 μg/ml), or HT DNA (1 μg/ml) for 8 h. Cells were then lysed for real-time PCR assays. Results are mean ± SD, n = 3 technical replicates. Unpaired *t* test was used for statistical analysis. (E) Wild-type (WT) and *Sirt4*-deficient (KO) MEFs were infected with HSV-1 (MOI = 1) for the indicated periods. Cells were then for immunoblot assays. All samples were from the same experiment and processed in parallel. A representative immunoblot is shown. (F) Wild-type (WT) and *Sirt4*-deficient (KO) MEFs were transfected with HSV60 (1 μg/ml), VACV70 (1 μg/ml), ISD (1 μg/ml), HT DNA (1 μg/ml), poly(dA:dT) (1 μg/ml), or cGAMP (1 μg/ml) for 8 h. Cells were then lysed for immunoblot assays. All samples were from the same experiment and processed in parallel. A representative immunoblot is shown. (G) Wild-type (WT) and *Sirt4*-deficient (KO) MEFs were infected with VSV (MOI = 1), or SeV (MOI = 1) for 8 h. Cells were then lysed for real-time PCR assays. Results are mean ± SD, n = 3 technical replicates. Unpaired *t* test was used for statistical analysis. (H) Wild-type (WT) and *Sirt4*-deficient (KO) MEFs were transfected with HA-SIRT4 (+) or control vector (−). At 24 h after transfection, cells were infected with HSV-1 for 8 h. Immunoblot assays were then performed as indicated. All samples were from the same experiment and processed in parallel. One representative replicate experiment is shown in the Western blot panels. Source data are available online for this figure.

Thus, these observations suggest that cGAS is a potential target of SIRT4 in antiviral host defense against DNA viruses.

Next, we examined whether SIRT4 regulates the production of cGAMP, which is synthesized by cGAS. As shown in Fig. 5A, *SIRT4* deficiency impaired HSV-1-triggered production of cGAMP, suggesting that SIRT4 promotes the synthesis of cGAMP by cGAS. Given that cGAS catalyzes the synthesis of cGAMP upon binding to DNA, we investigated whether SIRT4 affects the binding of cGAS to dsDNA. HSV60 or ISD was labeled by biotin and transfected into HEK293T cells expressing Flag-cGAS. Coimmunoprecipitation assays revealed that cGAS interacted with HSV60 or ISD, and these interactions were enhanced by overexpression of SIRT4 (Fig. 5B). Consistently, in PMA-THP1 cells, *SIRT4* knockdown decreased the association of endogenous cGAS with biotin-ISD (Fig. 5C). In addition, *SIRT4*-deficient PMA-THP1 cells exhibited a weaker association of cGAS with biotin-ISD or biotin-HSV60 compared to wild-type cells (Fig. 5D). Since the binding of dsDNA triggers the dimerization of cGAS and its subsequent activation, we next examined the role of SIRT4 in the formation of cGAS dimers. Coimmunoprecipitation assays indicated that cGAS dimerization was enhanced in the presence of SIRT4, further confirming its positive role in the activation of cGAS (Fig. 5E). Taken together, our observations suggest that SIRT4 promotes the activation of cGAS.

## SIRT4 deacetylates cGAS

Considering that SIRT4 has deacetylation activity, we next investigated whether SIRT4 deacetylates cGAS. As shown in Fig. 6A, SIRT4 decreased the acetylation of exogenously expressed cGAS in a dose-dependent manner. In addition, the enzymatically inactive mutant H161Y or delta 28 (an N-terminal 1–28 aa truncated SIRT4 mutant that cannot translocate into mitochondria) did not deacetylate cGAS to the same extent as wild-type SIRT4 (Fig. 6B). These observations were verified in an in vitro acetylation assay (Appendix Fig. S4A,B). We also investigated whether other SIRTs affect the acetylation of endogenous cGAS and found that SIRT4 effectively reduced its acetylation under unstimulated conditions (Appendix Fig. S4C). After HSV-1 infection, SIRT2 and SIRT5 also reduced cGAS acetylation to some extent, but the effect of SIRT4 was more pronounced (Appendix Fig. S4C). Consistently, *SIRT4* knockdown or *SIRT4* deficiency increased the

acetylation of endogenous cGAS (Fig. 6C,D). Next, we investigated whether the effect of SIRT4 on HSV-1-induced innate responses was dependent on its enzymatic activity. In response to HSV-1 infection or HSV60 transfection, transfection of wild-type SIRT4 into *Sirt4*-deficient MEFs rescued the impairment of innate immune signaling caused by *Sirt4* deficiency, whereas the H161Y mutant of SIRT4 did not (Fig. 6E–G), indicating that the catalytic activity of SIRT4 is essential for its role in the regulation of antiviral host defense. It has been reported that acetylation at Lys384, Lys394, or Lys414 helps maintain cGAS in an inactive state (Dai et al, 2019). Therefore, we generated a 3KR mutant (with K-to-R substitutions at Lys384, Lys394, and Lys414 of cGAS) to study the effect of SIRT4 on acetylation at these sites. While SIRT4 decreased the acetylation of wild-type cGAS, it had no significant effect on the acetylation of the 3KR mutant (Fig. 6H). Additionally, SIRT4 did not further enhance innate immune responses induced by the 3KR mutant (Fig. 6I,J). Taken together, our findings suggest that SIRT4 deacetylates cGAS and that its deacetylase activity is critical for its function in cGAS-mediated innate immune responses.

## SIRT4 inhibitor regulates DNA virus- or viral DNA-triggered innate immune responses

Since SIRT4 promotes cGAS-mediated antiviral innate immune responses, we explored whether SIRT4 inhibitors regulate DNA virus- or viral DNA-triggered immune responses. We used SIRT4-IN-1 (Compound 69), a reported selective SIRT4 inhibitor with an $IC_{50}$ of 16 μM that shows no relevant effects on other sirtuin isoforms (Pannek et al, 2024), in the following experiments. As expected, SIRT4-IN-1 treatment increased the acetylation of cGAS both with and without HSV60 transfection (Fig. 7A,B). Moreover, SIRT4-IN-1 treatment inhibited cGAMP synthesis induced by HSV60 transfection (Fig. 7C). In HSV60-transfected PMA-THP1 cells, SIRT4-IN-1 inhibited the production of type I IFNs, pro-inflammatory cytokines, and ISGs in a dose-dependent manner (Figs. 7D and EV5A). SIRT4-IN-1 treatment also attenuated HSV-1 or viral DNA-triggered antiviral innate immune responses (Figs. 7E and EV5B–D), but did not affect cGAMP-induced activation of STING and TBK1 (Fig. 7E). Similar results were observed in HSV60-transfected BMDCs (Fig. 7F,G). Furthermore, no effect of SIRT4-IN-1 on the signaling pathway was observed in cGAS-deficient PMs (Figs. 7H and EV5E). Finally, we examined

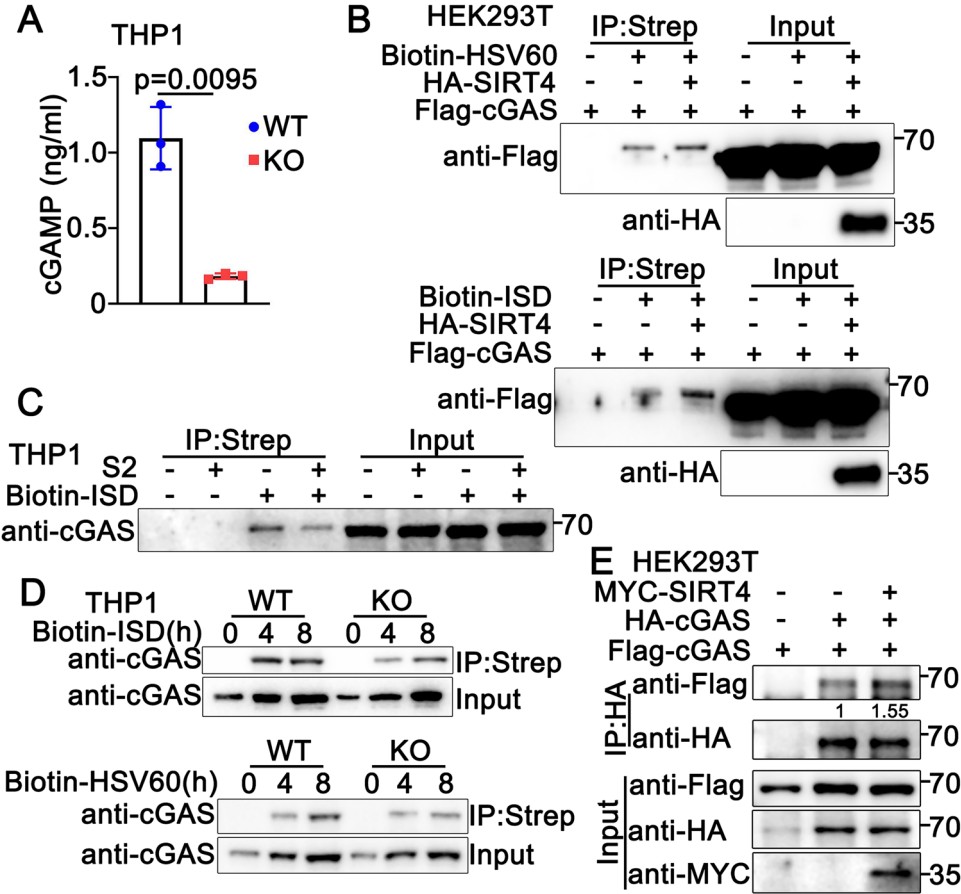

**Figure 5. SIRT4 promotes the activation of cGAS.**

(A) Wild-type (WT) and *SIRT4*-deficient (KO) PMA-THP1 cells were infected with HSV-1 (MOI = 1) for 24 h. Cells were then collected and lysed, and the lysates were analyzed by ELISA. Results are mean ± SD, n = 3 biological replicates. Unpaired *t* test was used for statistical analysis. (B) HEK293T cells were transfected with Flag-cGAS and HA-SIRT4 as indicated. At 24 h after transfection, cells were transfected with 1 μg biotinylated (Biotin)-HSV60 or Biotin-ISD for 8 h. Cell lysates were subjected to immunoprecipitation (IP) with streptavidin beads and analyzed by immunoblotting (IB) with anti-Flag. A representative immunoblot is shown. (C) PMA-THP1 cells were transfected with control siRNA (SC) or *SIRT4*-specific siRNA (S2). At 24 h after transfection, cells were transfected with 1 μg Biotin-ISD for 8 h. Cell lysates were subjected to immunoprecipitation (IP) with streptavidin beads and analyzed by immunoblotting (IB) with anti-cGAS. A representative immunoblot is shown. (D) Wild-type (WT) and *SIRT4*-deficient (KO) PMA-THP1 cells were transfected with 1 μg biotinylated (Biotin)-HSV60 or Biotin-ISD for the indicated periods. Cell lysates were then subjected to immunoprecipitation (IP) and immunoblot (IB) assays as indicated. A representative immunoblot is shown. (E) HEK293T cells were transfected with indicated plasmids for 24 h. Cell lysates were then subjected to immunoprecipitation (IP) and immunoblot (IB) assays as indicated. A representative immunoblot is shown. Source data are available online for this figure.

whether SIRT4-IN-1 suppresses the innate immune responses to DNA viral infection in vivo. Mice were treated with SIRT4-IN-1 and then infected with HSV-1. As shown in Figs. 7I–K and EV5F,G, compared to the control group, HSV-1-infected mice treated with SIRT4-IN-1 showed significantly reduced levels of IFN-β and IL-6 in serum and BALF, along with markedly lower expression of type I IFNs, ISGs, and pro-inflammatory cytokines in mouse tissues. We also found that expression of the HSV-1 viral protein UL30 was significantly increased in the tissues of SIRT4-IN-1-treated mice compared to controls (Figs. 7K and EV5H). Histopathological analysis of lung tissue sections also revealed that SIRT4-IN-1 exacerbated HSV-1-induced lung damage and increased macrophage infiltration (Fig. 7L; Appendix Fig. S4D). Histone deacetylase HDAC3 has been reported to deacetylate cGAS and regulate its activity (Dai et al, 2019). Therefore, we compared the effects of SIRT4 and HDAC3 on cGAS acetylation and downstream signaling

using knockdown and inhibitor approaches, respectively. Our findings indicate that both SIRT4 and HDAC3 can deacetylate cGAS and thereby influence its activity and signaling (Appendix Fig. S5A–H). In summary, our results demonstrate that a SIRT4 inhibitor suppresses DNA virus- or viral DNA-triggered innate immune responses at both cellular and animal levels.

## SIRT4 inhibitor regulates autoimmune responses

Next, we investigated whether a SIRT4 inhibitor could serve as a potential therapeutic agent for diseases associated with aberrant cGAS activation, such as autoimmune disorders. First, PMs and BMDCs were isolated from wild-type and *Trex1*-deficient mice, respectively, and then treated with the SIRT4 inhibitor. Subsequent real-time quantitative PCR and immunoblot analysis showed that the expression of interferons, interferon-related proteins, pro-

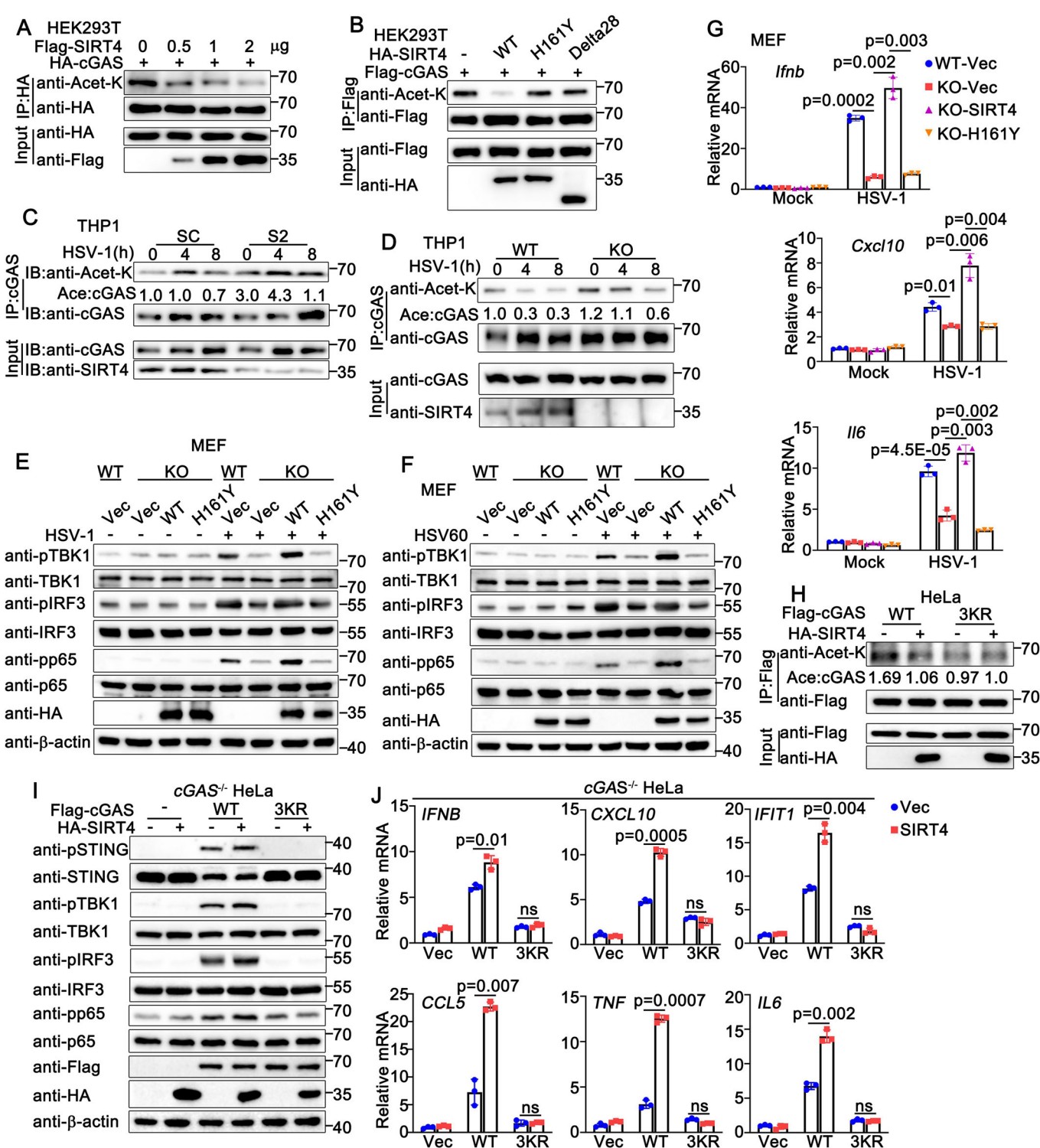

inflammatory factors, as well as the activation of innate immune signaling pathways in *Trex1*-deficient cells, were suppressed by SIRT4-IN-1 (Fig. 8A–C). It has been reported that aspirin can directly acetylate cGAS and effectively suppress self-DNA-induced autoimmunity (Dai et al, 2019). Therefore, we compared the effects of the SIRT4 inhibitor with those of aspirin in *Trex1*-deficient PMs and BMDCs and found that they produced similar effects on

autoimmune responses (Appendix Fig. S5I–K). To further evaluate the role of SIRT4 in SLE, we analyzed the large-scale gene expression dataset GSE45291 (Petri et al, 2019) comprising PBMCs from 302 SLE patients and 27 healthy donors. The analysis revealed significantly higher SIRT4 expression levels in SLE patients compared to healthy controls (Fig. 8D). We then isolated PBMCs from SLE patients. As expected, compared with healthy individuals,

**Figure 6. SIRT4 deacetylates cGAS.**

(A, B) HEK293T cells were transfected with plasmids as indicated. At 24 h after transfection, cells were lysed and subjected to immunoprecipitation (IP) and immunoblot (IB) assays as indicated. A representative immunoblot is shown. (C) PMA-THP1 cells were transfected with control siRNA (SC) or SIRT4-specific siRNA (S2). At 24 h after transfection, cells were infected with HSV-1 (MOI = 1) for the indicated periods. Cells were then lysed and subjected to immunoprecipitation (IP) and immunoblot (IB) assays as indicated. A representative immunoblot is shown. (D) Wild-type (WT) and SIRT4-deficient (KO) PMA-THP1 cells were infected with HSV-1 (MOI = 1) for the indicated periods. Cells were then lysed and subjected to immunoprecipitation (IP) and immunoblot (IB) assays as indicated. A representative immunoblot is shown. (E, F) Wild-type (WT) and Sirt4-deficient (KO) MEFs were transfected with an empty control vector (Vec), wild-type SIRT4, or its mutants. At 24 h after transfection, cells were infected with HSV-1 (MOI = 1) (E) or transfected with HSV60 (1 μg/ml) (F) for 8 h. Immunoblot assays were then performed as indicated. All samples were from the same experiment and processed in parallel. A representative immunoblot is shown. (G) Wild-type (WT) and Sirt4-deficient (KO) MEFs were transfected with an empty control vector (Vec), wild-type SIRT4, or its mutants. At 24 h after transfection, cells were infected with HSV-1 (MOI = 1) for 8 h. Cells were then lysed for real-time PCR assays. Results are mean ± SD, n = 3 technical replicates. Unpaired t test was used for statistical analysis. (H) HeLa cells were transfected with the indicated plasmids for 24 h. Cell lysates were then subjected to immunoprecipitation (IP) and immunoblot (IB) assays as indicated. A representative immunoblot is shown. (I) cGAS-deficient HeLa cells were transfected with indicated plasmids for 24 h. Cell lysates were then subjected to immunoblot assays as indicated. All samples were from the same experiment and processed in parallel. A representative immunoblot is shown. (J) cGAS-deficient HeLa cells were transfected with indicated plasmids for 24 h. Cell lysates were then subjected to real-time PCR assays. Results are mean ± SD, n = 3 technical replicates. Unpaired t test was used for statistical analysis. Source data are available online for this figure.

the expression of SIRT4 was significantly elevated in the PBMCs of SLE patients (Fig. 8E). In addition, SIRT4-IN-1 treatment markedly suppressed the expression of type I interferons and chronic inflammation-related cytokines and proteins in the PBMCs of SLE patients (Fig. 8F). In summary, our findings suggest that SIRT4 inhibitors may serve as a potential therapeutic strategy for cGAS-related autoimmune diseases.

## Discussion

cGAS is critical for the innate immune responses against DNA viruses, and thus the regulation of cGAS activity plays an important role in viral infection. In the current study, we identify SIRT4 as a novel positive regulator of cGAS activity that interacts with and deacetylates cGAS (Appendix Fig. S6). Our findings also indicate that SIRT4 could be a potential target for antiviral treatment or for other cGAS-associated immune diseases.

Recently, accumulating evidence suggests that several deacetylase Sirtuin family members are involved in antiviral innate immune responses, including SIRT1 targeting IRF3 and IFI16 (Qin et al, 2022; Wang et al, 2023), SIRT2 targeting G3BP1 and cGAS (Barthez et al, 2025; Li et al, 2023), SIRT3 targeting MAVS (Liu et al, 2024), and SIRT5 targeting DDX3 and MAVS (He et al, 2021; Liu et al, 2020). It has been reported that acetylation can either promote or inhibit cGAS activity, depending on the specific lysine residue modified (Dai et al, 2019; Song et al, 2020). These reports raise two interesting questions. First, do Sirtuin family members regulate innate immune signaling pathways by targeting cGAS? Second, how do Sirtuin family members regulate cGAS-mediated antiviral host defense?

In our study, we found that treatment with the Sirtuin inhibitor NAM increased the acetylation of cGAS. In addition, coimmunoprecipitation assays indicated that SIRT4 interacts with cGAS. These results identify SIRT4 among the Sirtuin family as a potential regulator of cGAS activity. At the cellular level, we demonstrated that the positive regulatory role of SIRT4 in HSV-1- or cytosolic dsDNA-induced signaling using SIRT4 overexpression, knockdown and knockout. Furthermore, Sirt4-deficient mice exhibited impaired host defense against DNA viruses. Although these findings suggest that SIRT4 regulates DNA virus- or cytosolic dsDNA-triggered innate immune responses, a question remained:

is cGAS the target of SIRT4 in innate immune responses? In addition to the finding that SIRT4 interacts with cGAS, two other results supported our hypothesis. First, SIRT4 increased the production of cGAMP, but did not affect cGAMP-induced antiviral innate immune responses. Second, in cGAS-deficient HeLa cells, SIRT4 overexpression or knockdown no longer regulated HSV-1-induced innate immune signaling. Thus, our findings answer the first question by demonstrating that the Sirtuin family member SIRT4 regulates innate immune signaling by targeting cGAS.

Because it has been reported that SIRT2 indirectly regulates cGAS activity by deacetylating G3BP1, a cGAS-interacting protein (Li et al, 2023), we first excluded the possibility that SIRT4 regulates the cGAS-STING pathway through SIRT2. As observed in Sirt2-deficient MEFs, Sirt4 knockdown still inhibited DNA virus-triggered antiviral responses (Fig. EV4E). Although early studies suggested that SIRT4 might have only very weak deacetylase activity against histone peptide substrates, a growing body of literature has revealed deacetylase activities of SIRT4 toward several non-histone proteins (Tao et al, 2023). In this study, we observed that SIRT4 decreased the acetylation of cGAS (Fig. 6A). The fact that the enzymatically inactive mutant H161Y of SRIT4 could not affect the acetylation of cGAS or rescue HSV-1- or HSV60-induced innate immune responses in Sirt4-deficient MEFs confirmed that the enzymatic activity of SIRT4 is necessary for its role in regulating cGAS acetylation and activity (Fig. 6B,E,F). Dai et al identified that acetylation of cGAS at Lys384, Lys394, or Lys414 helps maintain cGAS in an inactive state (Dai et al, 2019). Therefore, we generated a 3KR mutant of cGAS with mutations at these three lysine residues and observed that SIRT4 could no longer reduce acetylation of the 3KR mutant or enhance its activation of innate signaling (Fig. 6H–J), further indicating that the effect of SIRT4 on cGAS is related to its acetylation. Our data also elucidated the mechanisms by which SIRT4 regulates cGAS activity, including cGAMP production, the binding of cGAS to dsDNA, and formation of cGAS dimers (Figs. 5 and 7C). Dai et al demonstrated that acetylation at a single site of cGAS slightly reduces its DNA binding affinity (Dai et al, 2019), whereas we observed that SIRT4 increased the binding of cGAS to ISD or HSV60. We hypothesize that although acetylation at each individual site of cGAS only slightly affects DNA binding affinity, the combined removal of acetylation at multiple sites may have a significant effect. Thus, our research answers the second question by demonstrating that SIRT4

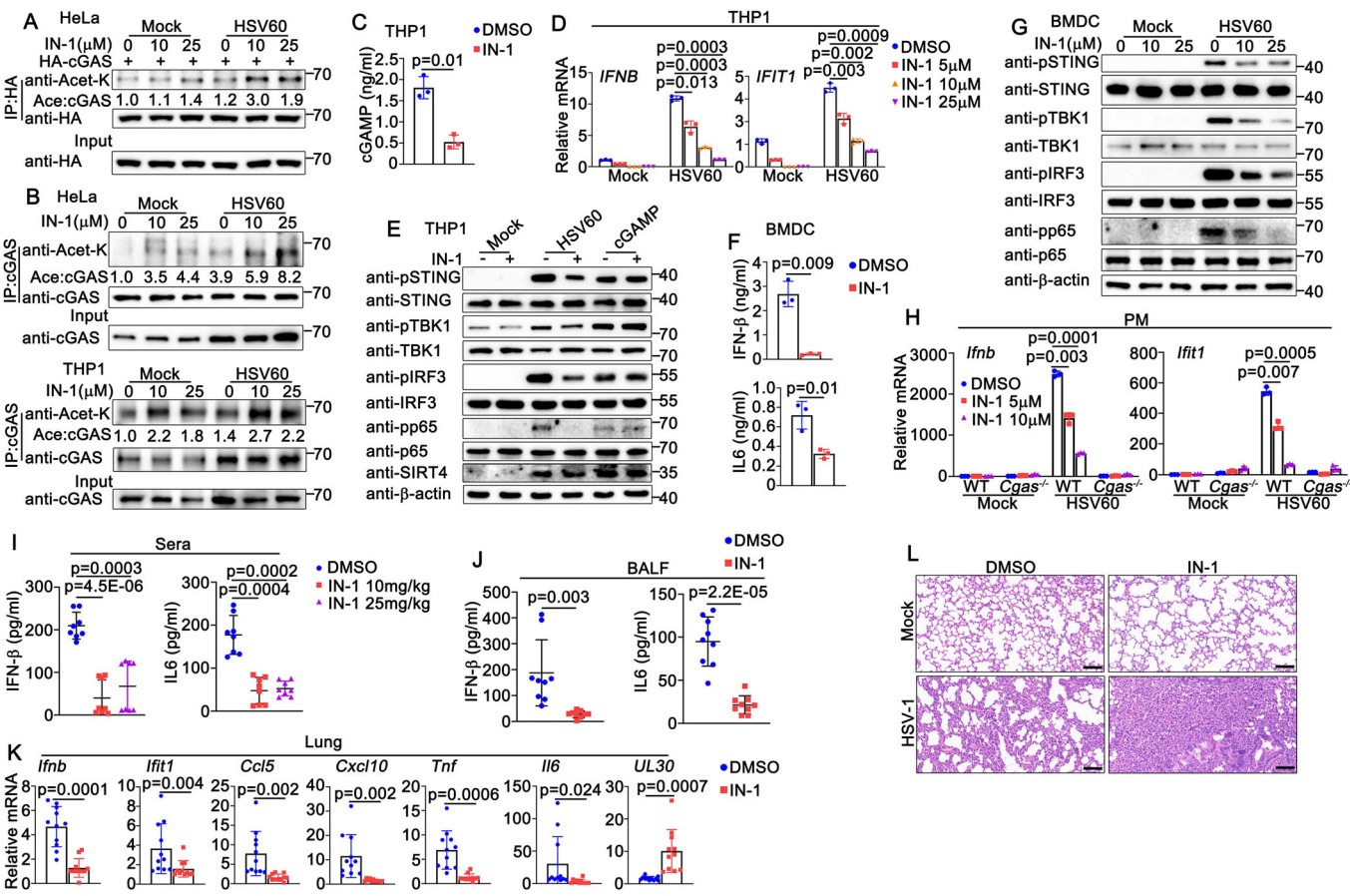

**Figure 7. SIRT4 inhibitor regulates DNA virus- or viral DNA-triggered innate immune responses.**

(A) HeLa cells were transfected with HA-cGAS and treated with SIRT4-IN-1 (0, 10, 25 μM). At 24 h after transfection, cells were transfected with HSV60 (1 μg/ml) for an additional 8 h. Cells were then lysed and subjected to immunoprecipitation (IP) and immunoblot (IB) assays. A representative immunoblot is shown. (B) HeLa (top) or PMA-THP1 (bottom) cells were treated with SIRT4-IN-1 (0, 10, 25 μM) for 12 h, and then transfected with HSV60 (1 μg/ml) for an additional 8 h. Cells were then lysed and subjected to immunoprecipitation (IP) and immunoblot (IB) assays. A representative immunoblot is shown. (C) PMA-THP1 cells were treated with SIRT4-IN-1 (0, 25 μM) for 12 h, and then transfected with HSV60 (1 μg/ml) for 24 h. Cells were then collected and lysed, and the lysates were analyzed by ELISA. Results are mean ± SD, $n = 3$ biological replicates. Unpaired $t$ test was used for statistical analysis. (D) PMA-THP1 cells were treated with SIRT4-IN-1 (0, 5, 10, 25 μM) for 12 h, and then transfected with HSV60 (1 μg/ml) for an additional 8 h. Cells were then lysed for real-time PCR assays. Results are mean ± SD, $n = 3$ technical replicates. Unpaired $t$ test was used for statistical analysis. (E) PMA-THP1 cells were treated with SIRT4-IN-1 (10 μM) or DMSO as a control for 12 h, and then transfected with HSV60 (1 μg/ml), or cGAMP (1 μg/ml) for 6 h. Cells were then lysed for immunoblot assays. All samples were from the same experiment and processed in parallel. A representative immunoblot is shown. (F) BMDCs were treated with SIRT4-IN-1 (10 μM) or DMSO as a control for 12 h, and then transfected with HSV60 (1 μg/ml) for 24 h. Supernatants were collected and analyzed by ELISA. Results are mean ± SD, $n = 3$ biological replicates. Unpaired $t$ test was used for statistical analysis. (G) BMDCs were treated with SIRT4-IN-1 (0, 10, 25 μM) for 12 h, and then transfected with HSV60 (1 μg/ml) for 6 h. Cells were then lysed for immunoblot assays. All samples were from the same experiment and processed in parallel. A representative immunoblot is shown. (H) Wild-type (WT) and Cgas-deficient (KO) PMs were treated with SIRT4-IN-1 (0, 5, 10 μM) for 12 h, and then transfected with HSV60 (1 μg/ml) for 6 h. Cells were then lysed for real-time PCR assays. Results are mean ± SD, $n = 3$ technical replicates. Unpaired $t$ test was used for statistical analysis. (I) Wild-type (WT) mice ($n = 8$, 8-week-old) were treated with SIRT4-IN-1 (0, 10, 25 mg/kg body weight), and then intranasally infected with HSV-1 ($1 \times 10^7$ PFU) for 24 h. Serum was collected and analyzed by ELISA. Results are mean ± SD. Each data point represents one mouse. Unpaired $t$ test was used for statistical analysis. (J) Wild-type (WT) mice ($n = 9$, 8-week-old) were treated with SIRT4-IN-1 (0, 10 mg/kg body weight), and then intranasally infected with HSV-1 ($1 \times 10^7$ PFU) for 24 h. BALF was collected and analyzed by ELISA. Results are mean ± SD. Each data point represents one mouse. Unpaired $t$ test was used for statistical analysis. (K) Wild-type (WT) mice ($n = 11$, 8-week-old) were treated with SIRT4-IN-1 (0, 10 mg/kg body weight), and then intranasally infected with HSV-1 ($1 \times 10^7$ PFU) for 24 h. Lungs were harvested and analyzed by real-time PCR. Results are mean ± SD. Each data point represents one mouse. Unpaired $t$ test was used for statistical analysis. (L) Wild-type (WT) mice were treated with SIRT4-IN-1 (0, 10 mg/kg body weight), and then intranasally infected with HSV-1 ($1 \times 10^7$ PFU) for 24 h. Lung sections were analyzed by H&E staining. Scale bars, 100 μm. One representative image for each group is shown. Source data are available online for this figure.

regulates cGAS-mediated antiviral host defense by modulating the acetylation and activity of cGAS.

Although cGAS recognizes DNA released from mitochondria, it is primarily located in the cytoplasm and the nucleus (Sun et al, 2021). A recent study has suggested that cGAS anchors to the outer mitochondrial membrane and suppresses ferroptosis to promote cancer progression (Qiu et al, 2023). SIRT4 is a well-known

mitochondrial protein, and our data suggest that SIRT4 co-localizes with cGAS in mitochondria (Fig. 1H,I). It is worth exploring whether the mitochondrial localization of cGAS contributes to its function in innate immune signaling and whether SIRT4 also plays a role in tumors by targeting cGAS.

Notably, cGAS can also be activated by endogenous DNA and thus is important in autoimmunity, sterile inflammatory responses

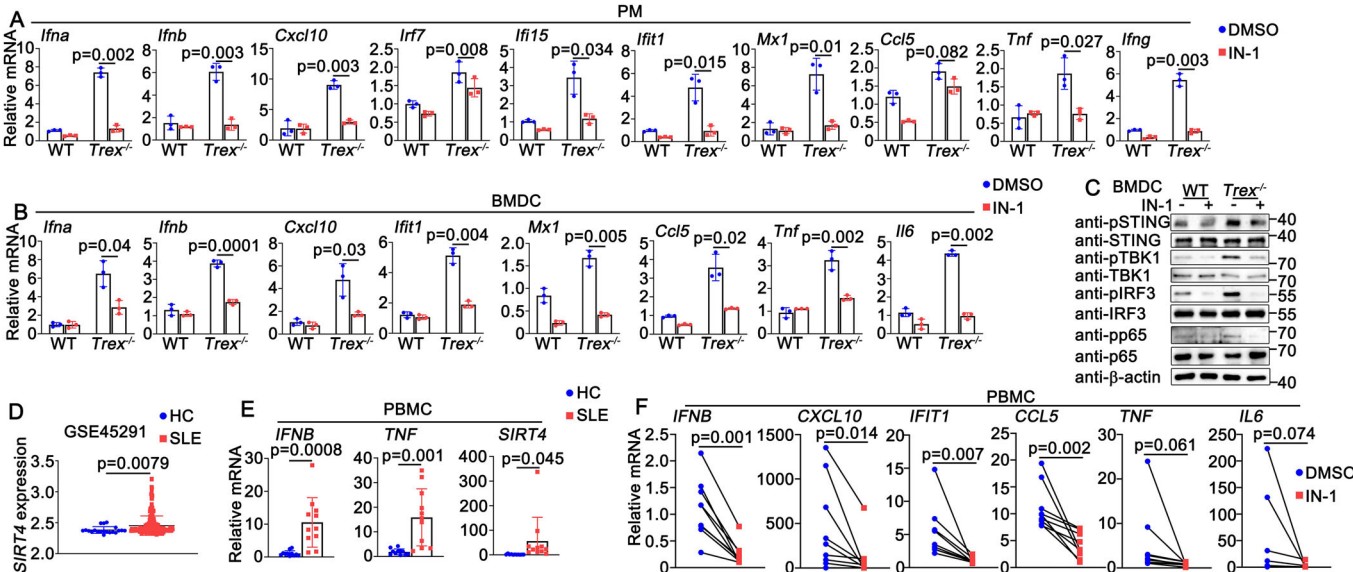

**Figure 8. SIRT4 inhibitor regulates autoimmune responses.**

(A) Wild-type (WT) and *Trex1*-deficient (KO) PMs were treated with SIRT4-IN-1 (0, 10 µM) for 12 h. Cells were then lysed for real-time PCR assays. Results are mean ± SD, $n = 3$ technical replicates. Unpaired *t* test was used for statistical analysis. (B) Wild-type (WT) and *Trex1*-deficient (KO) BMDCs were treated with SIRT4-IN-1 (0, 10 µM) for 12 h. Cells were then lysed for real-time PCR assays. Results are mean ± SD, $n = 3$ technical replicates. Unpaired *t* test was used for statistical analysis. (C) Wild-type (WT) and *Trex1*-deficient (KO) BMDCs were treated with SIRT4-IN-1 (0, 10 µM) for 12 h. Cells were then lysed for immunoblot assays. All samples were from the same experiment and processed in parallel. A representative immunoblot is shown. (D) Comparison of SIRT4 expression in monocytes between SLE patients and healthy donors (HC). Unpaired *t* test was used for statistical analysis. (E) PBMCs were isolated from blood samples of healthy donors or SLE patients ($n = 11$). Cells were lysed for real-time PCR assays. Results are mean ± SD. Each data point represents one subject. Unpaired *t* test was used for statistical analysis. (F) PBMCs were isolated from blood samples of SLE patients ($n = 9$). PBMCs were treated with SIRT4-IN-1 (0, 25 µM) for 12 h. Cells were then lysed for real-time PCR assays. Results are mean ± SD, $n = 3$ technical replicates. Paired *t* test was used for statistical analysis. Source data are available online for this figure.

and cellular senescence (Flavell and Sefik, 2024). Therefore, besides infectious diseases, it is also important to investigate the effect of SIRT4 on other cGAS-associated diseases. In this study, we focused on the potential therapeutic effects of a SIRT4 inhibitor on autoimmune responses. We found that the SIRT4 inhibitor significantly reduced autoimmune responses in *Trex1*-deficient cells as well as in PBMCs from SLE patients (Fig. 8), suggesting that SIRT4 inhibitors could serve as a novel treatment approach for autoimmune-related diseases. The regulatory functions of SIRT4 in different disease models, its protective role during viral infection, and its pathogenic effects in autoimmunity, illustrate that the innate immune response is a double-edged sword. It protects the host against pathogen invasion, but its abnormal and persistent activation can be harmful, thus requiring precise regulation. The activity of SIRT family molecules depends on NAD⁺. However, considering the functions of SIRT4 in antiviral and autoimmune processes, as well as the distinct roles of different SIRT family members in innate immunity, we may need to exercise caution when using NAD⁺ supplements, particularly in patients with autoimmune-related diseases. In addition, the role of SIRT4 inhibitors in other cGAS-mediated inflammatory diseases and ageing requires further research.

In addition, we compared the SIRT4 inhibitor with known inhibitors of cGAS deacetylation, RGFP966 (targeting HDAC3) and aspirin, and found that they produced similar effects on regulating cGAS activity and its signaling pathway (Appendix Fig. S5). This may be attributed to the critical role of cGAS, which requires multiple regulatory mechanisms in vivo. Such multi-layered

regulation may provide diverse therapeutic targets when developing drugs that target cGAS. Therefore, in drug development, it may be useful to investigate whether these inhibitors differ in their effects and side effects across various pathological processes, in order to identify optimal therapeutic agents.

Taken together, our findings demonstrate that the Sirtuin family member SIRT4 promotes DNA virus- or cytosolic DNA-triggered innate immune signaling by targeting cGAS. SIRT4 interacts with and deacetylates cGAS, leading to enhanced cGAS activity and increased cGAMP production. Thus, our findings contribute to understanding the regulation of cGAS activity by post-translational modifications and may provide a new potential target for managing cGAS-associated diseases.

## Methods

**Reagents and tools table**

| Reagent/resource | Reference or source | Identifier or catalog number |
|---|---|---|
| **Experimental models** | | |
| Mouse: C57BL/6 J, *Trex-/-* | Cyagen Biosciences | S-KO-05559 |
| Mouse: C57BL/6 N, *Sirt4-/-* | This study | Methods |
| HEK293T | Stem Cell Bank, Chinese Academy of Sciences. | SCSP-502 |

| Reagent/resource | Reference or source | Identifier or catalog number |
|---|---|---|
| THP1 | Stem Cell Bank, Chinese Academy of Sciences. | SCSP-567 |
| HeLa | Stem Cell Bank, Chinese Academy of Sciences. | SCSP-504 |
| Hela CRISPR/Cas9 cGAS -/- | This study | Methods |
| **Recombinant DNA** | | |
| pcDNA3.0_Flag-SIRT4 | Wang et al, 2023 | N/A |
| pcDNA3.0_Flag-cGAS mutants | This study | N/A |
| pcDNA3.0_HA-SIRT4 mutants | This study | N/A |
| pcDNA3.0_Myc-SIRT1 | Wang et al, 2023 | N/A |
| pcDNA3.0_Flag-SIRT2 | Wang et al, 2023 | N/A |
| pcDNA3.0_Myc-SIRT3 | Wang et al, 2023 | N/A |
| pcDNA3.0_Flag-SIRT5 | Wang et al, 2023 | N/A |
| pEnter_Flag-SIRT6 | Wang et al, 2023 | N/A |
| pEnter_Flag-SIRT7 | Wang et al, 2023 | N/A |
| **Antibodies** | | |
| Mouse anti-HA | Biolend | 901515 |
| Mouse anti-Flag | Sigma-aldrich | F3165 |
| Rabbit anti-STING | Proteintech | 19851-1-AP |
| Rabbit anti-cGAS | Proteintech | 26416-1-AP |
| Mouse anti-MYC | Proteintech | 66004-1-Ig |
| Mouse anti-TBK1 | Proteintech | 67211-1-Ig |
| Rabbit anti-p65 | Proteintech | 10745-1-Ig |
| Rabbit anti-IRF3 | Proteintech | 11312-1-AP |
| Rabbit anti-pSTING | Cell Signaling Technology | 1971S |
| Rabbit anti-pTBK1 | Cell Signaling Technology | 5483 T |
| Rabbit anti-pIRF3 | Cell Signaling Technology | 4947 s |
| Rabbit anti-pp65 | Cell Signaling Technology | 3033 |
| Mouse anti-SIRT4 | Proteintech | 66543-1-Ig |
| Mouse anti-GFP | Proteintech | 66002-Ig |
| Rabbit anti-b-tubulin | Proteintech | 10068-1-AP |
| Rabbit anti-acetylated-lysine | Cell Signaling Technology | 9441S |
| Mouse anti-b-actin | Proteintech | 60008-1 |
| Rabbit anti-Histone H3 | Flarbio | CSB-PA010109LA01HU |
| Rabbit anti-TOM20 | Proteintech | 11802-1-AP |
| Rabbit anti-HDAC3 | Proteintech | 10255-1-AP |
| **Oligonucleotides and other sequence-based reagents** | | |
| PCR primers | This study | Appendix Table S1 |

| Reagent/resource | Reference or source | Identifier or catalog number |
|---|---|---|
| HSV60 | TAAGACACGATGCGAT AAAATCTGTTTGTAAAA TTTATTAAGGGTACAAA TTGCCCTAGC (Forward) | GCTAGGGCAATTTGTACC CTTAATAAATTTTACA AACAGATTTTATCGC ATCGTGTCTTA (Reverse) |
| ISD | TACAGATCTACTAGT GATCTATGACTGATCT GTACATGATCTACA (Forward) | TGTAGATCATGTACAGA TCAGTCATAGATCACTA GTAGATCTGTA (Reverse) |
| VACV70 | CCATCAGAAAGAGGTTA TATATTTTTGTGAGACCAT GGAAGAGAGAGAAAG AGATAAAACTTTT TTACGACT(Forward) | AGTCGTAAAAAAGTTTTA TCTCTTTCTCTCTTCCATG GTCTCACAAAAATATTAAA CCTCTTTCTGATGG (Reverse) |
| *HDAC3* siRNA | Ribobio | siB1032283530 |
| *SIRT4* siRNA-S1 | GGGUUAUUUGUGC CAGCAA (Forward) | UUGCUGGCACAAA UAACCC (Reverse) |
| *SIRT4* siRNA-S2 | CAUCCAGCAUGGUGA UUUU (Forward) | AAAAUCACCAUGCUG GAUG (Reverse) |
| *SIRT4* siRNA-S3 | GCGUGUCUGAAACUG AAUU (Forward) | AAUUCAGUUUCAGAC ACGC (Reverse) |
| The negative control siRNA-NC | UUCUCCGAACGUGUC ACGU (Forward) | ACGUGACACGUUCGGA GAA (Reverse) |
| **Chemicals, enzymes and other reagents** | | |
| SIRT4-IN-1 | MedChemExpress | HY-163316 |
| RGFP966 | Selleck | S7229 |
| PMA | Beyotime Bio technology | S1819 |
| Herring testis (HT) DNA | Sigma-aldrich | D6898 |
| Poly(dA:dT) | InvivoGen | tlrl-patn |
| cGAMP | InvivoGen | tlrl-nacga23 |
| **Software** | | |
| GraphPad Prism (version 8) | GraphPad Software | https://www.graphpad.com/scientificsoftware/prism/ |
| Adobe Photoshop CS6 | Adobe Systems Incorporated | https://www.adobe.com |
| **Other** | | |
| A1R microscope | Nikon | |

## Mice

*Sirt4*-deficient mice were generated on a C57BL/6 background using CRISPR/Cas9-mediated gene editing by the Laboratory of Genetic Regulators in the Immune System at Xinxiang Medical University. Mice were housed in a facility with ad libitum access to food and water and were maintained under a 12-h light/12-h dark cycle. All animal procedures were performed in accordance with guidelines approved by the Animal Care Committee of Xinxiang Medical University, China (XYLL-2021049). Age- and sex-matched wild-type and *Sirt4*-deficient mice were used in the experiments.

## Cell culture, transfection, and stimulation

*cGAS*-deficient HeLa cells were kindly provided by Zhiduo Liu (Shanghai Jiao Tong University). HeLa and HEK293T cells were

cultured in Dulbecco's modified Eagle's medium (DMEM). THP1 cells were maintained in RPMI 1640. "PMA-THP1 cells" refers to THP1 cells that were pretreated with 100 ng/ml PMA for overnight. All cells were cultured in medium supplemented with 10% FBS (Gibco), 4 mM L-glutamine, 100 μg/ml penicillin, and 100 U/ml streptomycin under humidified conditions with 5% $CO_2$ at 37 °C. Transfection of HeLa, HEK293T, THP1, and MEF cells was performed with Lipofectamine 2000 (Invitrogen) according to the manufacturer's instructions.

## Preparations of MEFs

The procedure for generating MEFs has been described previously (Yang et al, 2021). Briefly, mice were anesthetized with isoflurane on embryonic days 12.5–14.5 and sacrificed by rapid cervical dislocation. Embryos were removed, separated from the uterine horns, and washed three times with 10 ml of PBS without calcium or magnesium. After removal of the head and internal organs, the embryos were washed three times with 10 ml of PBS without calcium or magnesium. The embryos were then minced, and 2 ml of trypsin/EDTA per embryo was added. The mixture was incubated, and gently shaken at room temperature for 15 min. The trypsin digestion was stopped by adding 20 ml of complete MEF medium. The undigested tissue pieces were allowed to settle to the bottom, and the supernatant was removed and centrifuged at low speed for 5 min at $300 \times g$. The cell pellet was resuspended in MEF medium and plated in a culture flask. Inspection on day 3 revealed a homogenous layer of fibroblast-like cells. MEFs from passages 1 to 5 were used in the experiments.

## Immunoprecipitation and immunoblot analysis

Immunoprecipitation and immunoblot analysis were performed as described previously (Wang et al, 2017). Briefly, after the treatment, cells were lysed in lysis buffer containing 1.0% (vol/vol) Nonidet P40, 20 mM Tris-HCl, pH 8.0, 10%(vol/vol) glycerol, 150 mM NaCl, 0.2 mM $Na_3VO_4$, 1 mM NaF, 0.1 mM sodium pyrophosphate and a protease inhibitor 'cocktail' (Roche). After centrifugation for 20 min at $14,000 \times g$, supernatants were collected and incubated with the indicated antibodies and protein A/G Plus-agarose immuno-precipitation reagent (sc-2003, Santa Cruz) at 4 °C for 3 h or overnight. After three washes, the immunoprecipitates were boiled in SDS sample buffer for 10 min and subjected to immunoblot analysis. For immunoblot analysis, equal amounts of protein were separated by SDS-PAGE or Native PAGE and transferred onto PVDF membranes. The membranes were blocked with 5% non-fat milk and then incubated overnight at 4 °C with primary antibodies. After washing, membranes were incubated with HRP-conjugated secondary antibodies for 1 h at room temperature. Protein bands were visualized using ECL substrate and detected with an Azure 600 imaging system (Azure Biosystems).

## IRF3 dimerization assay

IRF3 dimerization assay was performed as described previously (Wang et al, 2023). Briefly, cell extracts were prepared in native lysis buffer (50 mM Tris-HCl (pH 8.0), 150 mM NaCl, 1% NP40, 1% protease inhibitor cocktail (Roche), and 1% orthovanadate) by incubation for 30 min at 4 °C. Lysates were loaded onto a native

PAGE gel (without SDS), which was prerun in electrophoresis buffer (25 mM Tris-HCl (pH 8.4) and 192 mM glycine with 0.2% deoxycholate in the cathode chamber and without deoxycholate in the anode chamber) for 60 min at 40 mA, and then electrophoresed for 60 min at 25 mA. The native gel was then analyzed by immunoblot as described above.

## Real-time PCR

Total RNA was extracted from the cultured cells with TRIzol reagent (Invitrogen) as described by the manufacturer. Transcript levels of all genes were quantified by real-time PCR with SYBR Green qPCR Master Mix on a 7500 Fast real-time PCR system (Applied Biosystems). The relative fold induction was calculated using the $2^{-\triangle\triangle Ct}$ method. The primers used for real-time PCR were as previously described (Yang et al, 2023) and are provided in Appendix Table S1.

## ELISA

THP1 cells were infected with viruses or transfected with synthetic nucleic acids for 24 h. Supernatants were collected to measure IFN-β (KE00187, Proteintech) and TNF-α (88-7346-88, Thermo Fisher Scientific). MEF or BMDM cells were infected with viruses or transfected with synthetic nucleic acids for 24 h. Supernatants were collected to measure IFN-β (R&D), TNF-α (88-7323-88, Thermo Fisher Scientific), and IL-6 (88-7064-88, Thermo Fisher Scientific). 8-week-old wild-type mice were infected with HSV-1, VSV or SeV for the indicated time periods. Serum or BALF was then was collected for ELISA.

## cGAMP quantification

To measure 2'3'-cGAMP, a commercial ELISA kit based on competitive binding between 2'3'-cGAMP and 2'3'-cGAMP-HRP was used (Catalog No. 501700, Cayman). Briefly, PMA-THP1 cells were seeded into a 6-well plate. After transfection with HSV60 or infection with HSV-1, cell pellets were lysed using homogenization buffer (10 mM Tris-HCl, pH 7.4; 10 mM KCl; 1.5 mM $MgCl_2$) and centrifuged at 13,000 rpm for 20 min. The supernatant was then heated at 95 °C for 5 min and centrifuged again at 12,000 rpm for 10 min to remove denatured proteins. The supernatant was collected for cGAMP quantification according to the manufacturer's instructions.

## In vitro deacetylation assay

Flag-cGAS was purified from HEK293T cells by immunoprecipitation. HA-SIRT4 and its H161Y mutant were synthesized using a TNT Quick-coupled Transcription/Translation Systems kit (L520A, Promega). The reaction mixture contained 10× Ac buffer (200 mM pH 8.0 HEPES, 10 mM DTT, 10 mM PMSF, and 1 mg/ml BSA), 2× De-Ac buffer (8 mM $MgCl_2$, 100 mM NaCl and 20% glycerol), and Flag-cGAS, with or without 1 mM $NAD^+$. SIRT4 and its H161Y mutant were added as indicated. The reaction mixture was incubated at 37 °C for 2 h and then boiled in the SDS sample buffer.

## RNA interference

SIRT4 siRNA was purchased from GenePharma. PMA-THP1 cells were transfected with siRNA using Lipofectamine 2000 according

to the manufacturer's instructions. At 24 h after transfection, cells were used for further experiments.

## Viruses and infection

Viral infection was performed as described previously (Yang et al, 2023; Yang et al, 2021). Briefly, cells were infected with HSV-1 (KOS strain), VSV, or SeV for 1.5 h. Then cells were washed with PBS and cultured in fresh medium. In the mouse model, age- and sex-matched groups of mice were infected intravenously with HSV-1, or infected intranasally with HSV-1 or VSV.

## Generation of *SIRT4* knockout cell line

The *SIRT4*-deficient THP1 cell line was generated using the CRISPR-Cas9 system as described previously (Yang et al, 2020).

## Confocal microscopy

After transfection, HEK293T cells were fixed with 4% PFA in PBS, permeabilized with TritonX-100, and then blocked with 1% BSA in PBS. Nuclei were stained with 4, 6-diamidino-2-phenylindole (DAPI). Images were acquired using a Nikon A1R microscope.

## In situ PLA

The in situ PLA assay was performed as described previously (Yang et al, 2023). Briefly, the assay was conducted using a DUOLink PLA kit (Sigma). Cells were seeded on Teflon-coated coverslips and cultured overnight. After treatment, cells were fixed in 4% PFA in PBS for 15 min and washed three times with PBS. Next, cells were permeabilized with 0.1% TritonX-100 on ice for 5 min, washed with PBS, blocked in 5% BSA in PBS for 1 h at 37 °C, and incubated with primary antibodies overnight at 4 °C. The following day, cells were washed three times with the wash buffer (DUO82049) and incubated with PLA probe-conjugated secondary antibodies (DUO92001, DUO92005) for 2 h at 37 °C, followed by three washes with wash buffer. Cells were then incubated with the Duolink in situ detection reagents red (DUO92008) for 2 h at 37 °C. Finally, Nuclei were stained with DAPI and coverslips were mounted onto slides. Images were acquired using a Nikon A1R microscopy, and Image Pro Plus 6.0 software was used for quantitative analysis.

## Tissue section staining

The tissue sections were prepared and stained according to the standard operating procedure (SOP) of Servicebio. Briefly, paraffin-embedded sections were dewaxed, hydrated, and subjected to antigen retrieval. Endogenous peroxidase was blocked with 3% $H_2O_2$. After serum blocking, sections were incubated overnight with a primary CD68 antibody at 4 °C, followed by an HRP-conjugated secondary antibody. DAB was used for visualization. For H&E staining, separate sections were stained with hematoxylin and eosin. All slides were then dehydrated, cleared, and mounted with neutral gum for microscopic examination. For each slide, five non-overlapping fields with the highest density of positive staining were selected at high-power magnification (×400) and CD68-positive cells were manually counted by a trained observer who was blinded to the sample groups to obtain the average value.

## RNA-seq analysis

After HSV-1 stimulation, total RNA was extracted from control and *SIRT4*-deficient THP1 cells using Trizol reagent. Library preparation and sequencing were performed by BGI (Shenzhen, China). To investigate phenotypic changes, GO (http://www.geneontology.org/) and KEGG (https://www.kegg.jp/) enrichment analysis of annotated differentially expressed genes were performed by Dr. Tom (https://biosys.bgi.com/). The significance of terms and pathways was assessed by a Q value with a rigorous threshold (Q value ≤ 0.05) and adjusted by the Bonferroni method. All RNA-seq data have been deposited in the NCBI database and are publicly available under accession number PRJNA1162950.

## Nuclear extracts

The nuclear extracts were prepared as described previously (Yang et al, 2018). Briefly, cells were harvested and washed with PBS. Cells were lysed with buffer A (10 mM HEPES, 1.5 mM $MgCl_2 • 6 H_2O$, 10 mM KCl, 0.5 mM DTT, protease inhibitor cocktail, 0.1% Nonidet P-40, pH 7.9). Lysates were placed on ice for 10 min and centrifuged at 10,000 rpm for 5 min at 4 °C to remove cytoplasmic proteins. Nuclear proteins were extracted from the pellet using ice-cold buffer C (20 mM HEPES, 1.5 mM $MgCl_2 • 6 H_2O$, 0.42 M NaCl, 0.2 mM EDTA, 25% glycerol, 0.5 mM DTT, protease inhibitor cocktail, pH 7.9). Insoluble material was removed by centrifugation at 10,000 rpm for 5 min at 4 °C. Protein concentration was measured using the BCA protein assay reagent kit according to the manufacturer's instructions.

## Isolation of peripheral blood mononuclear cells from SLE patients

PBMCs were isolated from the blood of SLE patients using Ficoll-Paque (GE, 17144002). The study was conducted in accordance with the principles of the Declaration of Helsinki and approved by the Institutional Review Board of the Medical Ethic Committee of the First Affiliated Hospital of Xinxiang Medical University, following its guidelines for the protection of human subjects (Approval Number EC-025-516). Informed consent was obtained from all participants, and the experiments adhered to the principles outlined in the WMA Declaration of Helsinki and the Department of Health and Human Services Belmont Report.

## Statistics

All the experiments were repeated at least three times, using independent experimental samples and statistical tests as specified in the figure legends. GraphPad Prism 8 was employed for statistical analysis. A *P* value lower than 0.05 was considered statistically significant.

# Data availability

All RNA-seq data have been deposited in the NCBI database and are publicly available under accession number PRJNA1162950.

The source data of this paper are collected in the following database record: biostudies:S-SCDT-10_1038-S44319-026-00708-5.

## Peer review information

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

## Acknowledgements

This work was supported by grants from the National Natural Science Foundation of China (32170871 and 82371759), the Natural Science Foundation of Henan Province (222300420064, 252300421249), the Program for Science & Technology Innovation Talents in Higher Education of Henan Province (23HASTIT050), the Key Scientific and Technological Projects of Henan Province (232102310154), the 111 Project (No. D20036), and the Zhongyuan Academician Fund from Henan Province (C24185).

## Author contributions

**Bo Yang**: Conceptualization; Resources; Data curation; Formal analysis; Supervision; Funding acquisition; Validation; Investigation; Visualization; Methodology; Writing—original draft; Project administration; Writing—review and editing. **Yanjie Zhang**: Validation; Investigation; Methodology; Writing—review and editing. **Saiyu Wang**: Validation; Investigation; Writing—review and editing. **Yufei Wu**: Validation; Investigation; Writing—review and editing. **Zilu Diao**: Validation; Investigation; Writing—review and editing. **Qunmei Zhang**: Resources; Investigation. **Chen Lu**: Validation; Investigation. **Mengyang Shen**: Validation; Investigation. **Xuewei Zhang**: Resources; Investigation. **Shujun Ma**: Funding acquisition; Investigation. **Chunsheng Yang**: Resources; Investigation. **Jinyong Pei**: Validation; Investigation. **Hongxia Xing**: Resources. **Yinming Liang**: Resources; Supervision; Project administration; Writing—review and editing. **Jie Wang**: Conceptualization; Resources; Data curation; Formal analysis; Supervision; Funding acquisition; Validation; Investigation; Visualization; Methodology; Writing—original draft; Project administration; Writing—review and editing.

Source data underlying figure panels in this paper may have individual authorship assigned. Where available, figure panel/source data authorship is listed in the following database record: biostudies:S-SCDT-10_1038-S44319-026-00708-5.

## Disclosure and competing interests statement

The authors declare no competing interests.

# Expanded View Figures

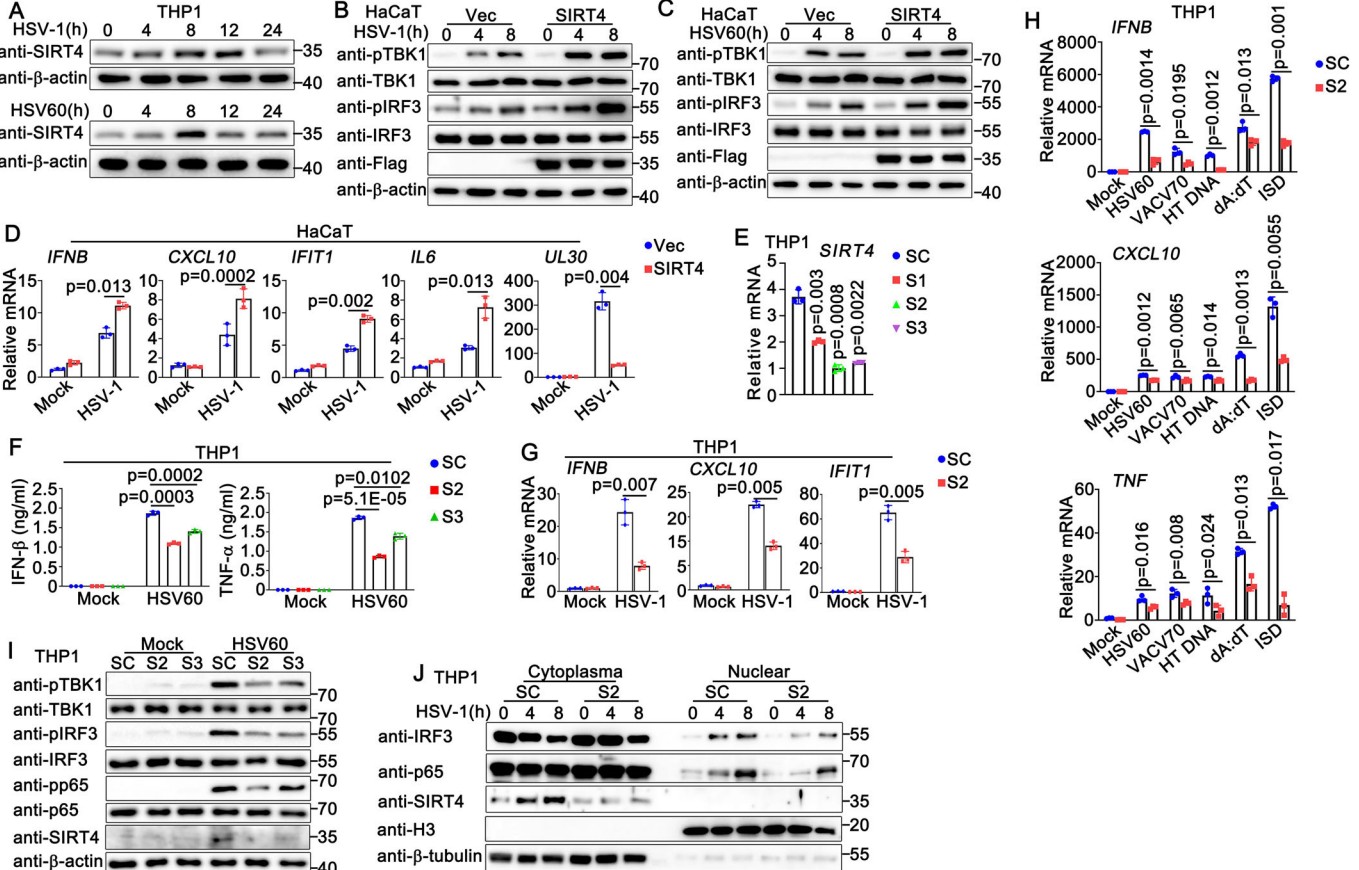

**Figure EV1. SIRT4 positively regulates DNA virus- or viral DNA-triggered innate immune responses.**

(A) PMA-THP1 cells were infected with HSV-1 (MOI = 1) (top) or transfected with HSV60 (1 μg/ml) (bottom) for the indicated periods. Cells were then lysed for immunoblot assays. A representative immunoblot is shown. (B, C) HaCaT keratinocytes were transfected with an empty vector (Vec) or a SIRT4 expressing plasmid for 24 h, and then infected with HSV-1 (MOI = 1) (B) or transefected with HSV60 (1 μg/ml) (C) for the indicated periods. Cells were then lysed for immunoblot assays. All samples were from the same experiment and processed in parallel. A representative immunoblot is shown. (D) HaCaT keratinocytes were transfected with an empty vector (Vec) or a *SIRT4* expressing plasmid for 24 h, and then infected with HSV-1 (MOI = 1) for 8 h. Cells were then lysed for real-time PCR assays. Results are mean ± SD, n = 3 technical replicates. Unpaired *t* test was used for statistical analysis. (E) PMA-THP1 cells were transfected with control siRNA (SC) or *SIRT4*-specific siRNA (S1, S2, and S3). At 24 h after transfection, cells were lysed for real-time PCR assays. Results are mean ± SD, n = 3 technical replicates. Unpaired *t* test was used for statistical analysis. (F) PMA-THP1 cells were transfected with control siRNA (SC) or *SIRT4*-specific siRNA (S2 and S3). At 24 h after transfection, cells were transfected with HSV60 (1 μg/ml) for 24 h. Supernatants were collected and analyzed by ELISA. Results are mean ± SD, n = 3 biological replicates. Unpaired *t* test was used for statistical analysis. (G) PMA-THP1 cells were transfected with control siRNA (SC) or *SIRT4*-specific siRNA (S2). At 24 h after transfection, cells were infected with HSV-1 (MOI = 1) for 8 h. Cells were then lysed for real-time PCR assays. Results are mean ± SD, n = 3 technical replicates. Unpaired *t* test was used for statistical analysis. (H) PMA-THP1 cells were transfected with control siRNA (SC) or *SIRT4*-specific siRNA (S2). At 24 h after transfection, cells were transfected with HSV60 (1 μg/ml), VACV70 (1 μg/ml), HT DNA (1 μg/ml), poly(dA:dT) (1 μg/ml), or ISD (1 μg/ml) for 8 h. Cells were then lysed for real-time PCR assays. Results are mean ± SD, n = 3 technical replicates. Unpaired *t* test was used for statistical analysis. (I) PMA-THP1 cells were transfected with control siRNA (SC) or *SIRT4*-specific siRNA (S2 and S3). At 24 h after transfection, cells were transfected with HSV60 (1 μg/ml) for 8 h. Immunoblot assays were then performed as indicated. All samples were from the same experiment and processed in parallel. A representative immunoblot is shown. (J) PMA-THP1 cells were transfected with control siRNA (SC) or *SIRT4*-specific siRNA (S2). At 24 h after transfection, cells were infected with HSV-1 (MOI = 1) for 8 h. Cells were then fractionated into cytosolic and nuclear fractions. Immunoblot assays were then performed as indicated. A representative immunoblot is shown. Source data are available online for this figure.

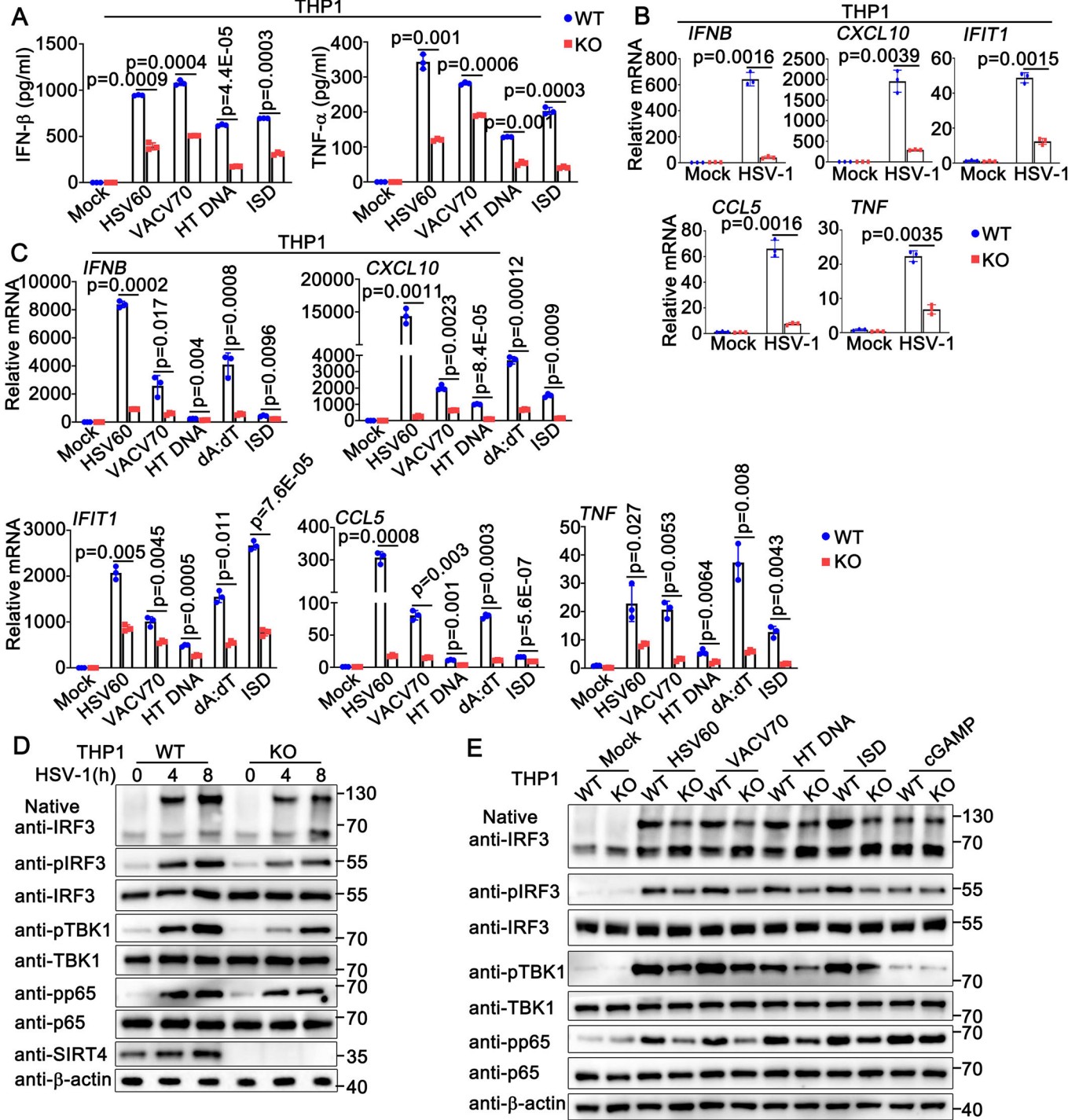

◀ **Figure EV2.** *SIRT4* **deficiency reduces antiviral innate immune responses against cytosolic DNA in PMA-THP1 cells.**

(A) Wild-type (WT) and *SIRT4*-deficient (KO) PMA-THP1 cells were transfected with HSV60 (1 µg/ml), VACV70 (1 µg/ml), HT DNA (1 µg/ml), or ISD (1 µg/ml) for 24 h. Supernatants were collected and analyzed by ELISA. Results are mean ± SD, $n = 3$ biological replicates. Unpaired $t$ test was used for statistical analysis. (B) Wild-type (WT) and *SIRT4*-deficient (KO) PMA-THP1 cells were infected with HSV-1 (MOI = 1) for 8 h. Cells were then lysed for real-time PCR assays. Results are mean ± SD, $n = 3$ technical replicates. Unpaired $t$ test was used for statistical analysis. (C) Wild-type (WT) and *SIRT4*-deficient (KO) PMA-THP1 cells were transfected with HSV60 (1 µg/ml), VACV70 (1 µg/ml), HT DNA (1 µg/ml), poly(dA:dT) (1 µg/ml), or ISD (1 µg/ml) for 8 h. Cells were then lysed for real-time PCR assays. Results are mean ± SD, $n = 3$ technical replicates. Unpaired $t$ test was used for statistical analysis. (D) Wild-type (WT) and *SIRT4*-deficient (KO) PMA-THP1 cells were infected with HSV-1 (MOI = 1) for the indicated periods. Cells were then lysed for Native PAGE and SDS-PAGE analysis. Both Native PAGE and SDS-PAGE samples originate from the same experiment and were processed in parallel. A representative immunoblot is shown. (E) Wild-type (WT) and *SIRT4*-deficient (KO) PMA-THP1 cells were transfected with HSV60 (1 µg/ml), VACV70 (1 µg/ml), HT DNA (1 µg/ml), ISD (1 µg/ml), or cGAMP (1 µg/ml) for 8 h. Cells were then lysed for Native PAGE and SDS-PAGE analysis. Both Native PAGE and SDS-PAGE samples originate from the same experiment and were processed in parallel. A representative immunoblot is shown. Source data are available online for this figure.

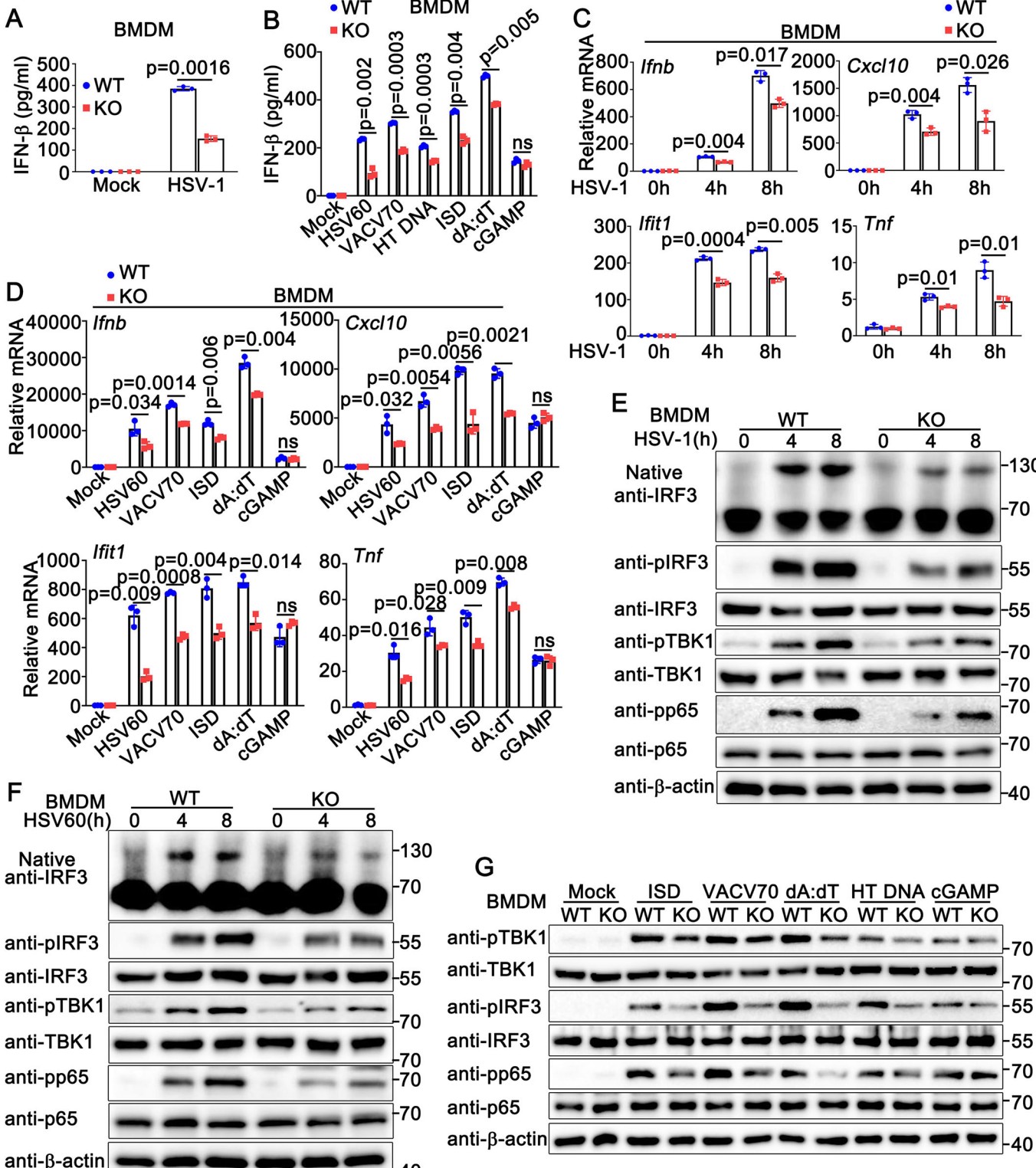

◀ **Figure EV3.** *Sirt4* **deficiency reduces antiviral innate immune responses against DNA viruses in BMDMs.**

(A) Wild-type (WT) and *Sirt4*-deficient (KO) BMDMs were infected with HSV-1 (MOI = 1) for 24 h. Supernatants were collected and analyzed by ELISA. Results are mean ± SD, *n* = 3 biological replicates. Unpaired *t* test was used for statistical analysis. (B) Wild-type (WT) and *Sirt4*-deficient (KO) BMDMs were transfected with HSV60 (1 μg/ml), VACV70 (1 μg/ml), HT DNA (1 μg/ml), ISD (1 μg/ml), poly(dA:dT) (1 μg/ml), or cGAMP (1 μg/ml) for 24 h. Supernatants were collected and analyzed by ELISA. Results are mean ± SD, *n* = 3 biological replicates. Unpaired *t* test was used for statistical analysis. (C) Wild-type (WT) and *Sirt4*-deficient (KO) BMDMs were infected with HSV-1 (MOI = 1) for the indicated periods. Cells were then lysed for real-time PCR assays. Results are mean ± SD, *n* = 3 technical replicates. Unpaired *t* test was used for statistical analysis. (D) Wild-type (WT) and *Sirt4*-deficient (KO) BMDMs were transfected with HSV60 (1 μg/ml), VACV70 (1 μg/ml), ISD (1 μg/ml), poly(dA:dT) (1 μg/ml), or cGAMP (1 μg/ml) for 8 h. Cells were then lysed for real-time PCR assays. Results are mean ± SD, *n* = 3 technical replicates. Unpaired *t* test was used for statistical analysis. (E, F) Wild-type (WT) and *Sirt4*-deficient (KO) BMDMs were infected with HSV-1 (MOI = 1) (E), or transfected with HSV60 (1 μg/ml) (F) for the indicated periods. Cells were then lysed for Native PAGE and SDS-PAGE analysis. Both Native PAGE and SDS-PAGE samples originate from the same experiment and were processed in parallel. A representative immunoblot is shown. (G) Wild-type (WT) and *Sirt4*-deficient (KO) BMDMs were transfected with ISD (1 μg/ml), VACV70 (1 μg/ml), poly(dA:dT) (1 μg/ml), HT DNA (1 μg/ml), or cGAMP (1 μg/ml) for 8 h. Cells were then lysed for immunoblot assays. All samples were from the same experiment and processed in parallel. A representative immunoblot is shown. Source data are available online for this figure.

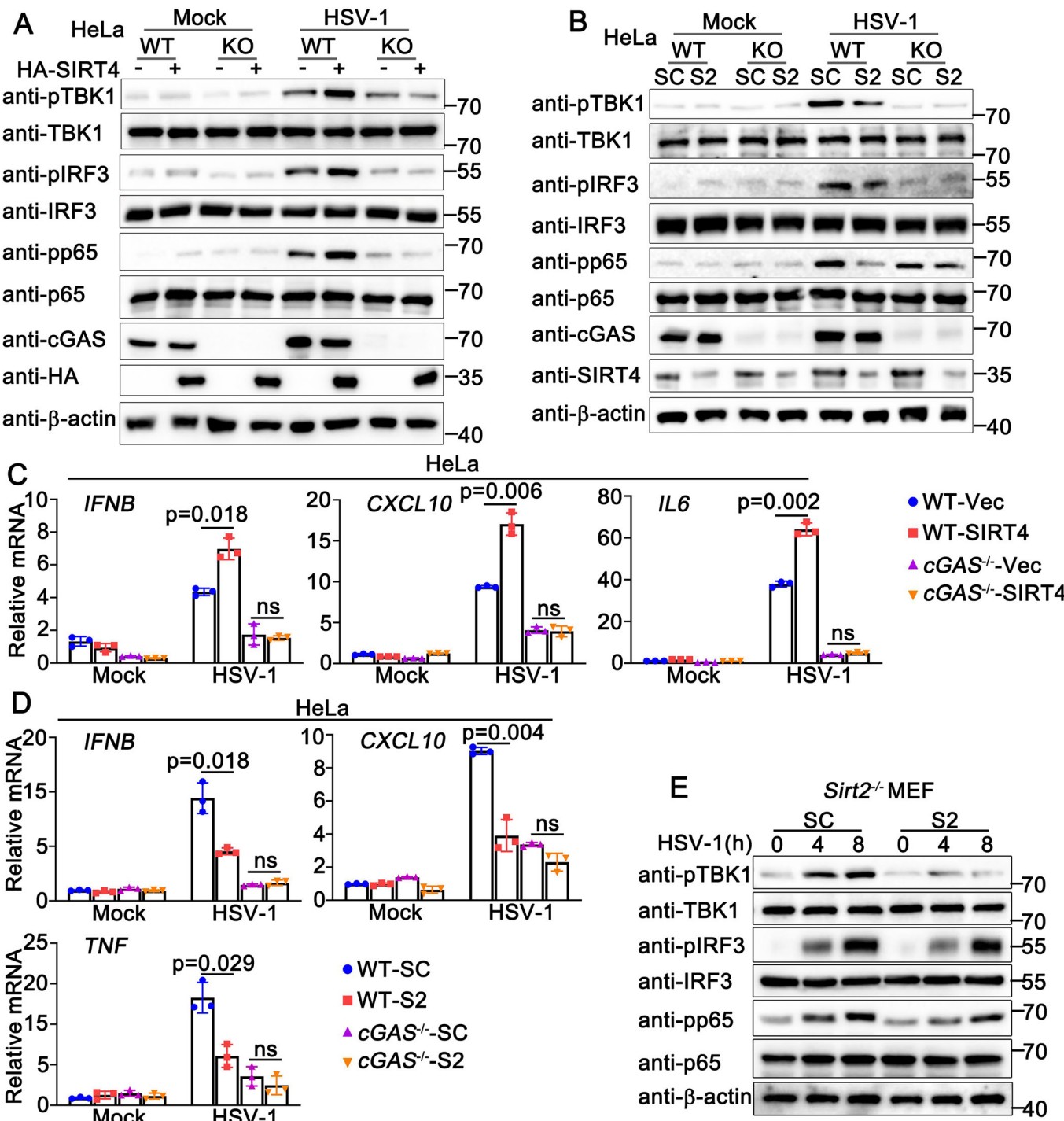

**Figure EV4.   The role of SIRT4 in innate immunity is dependent of cGAS.**

(A) Wild-type (WT) and *cGAS*-deficient (KO) HeLa cells were transfected with HA-SIRT4 (+) or an empty control vector (−). At 24 h after transfection, cells were infected with HSV-1 (MOI = 1) for 8 h. Immunoblot assays were then performed as indicated. All samples were from the same experiment and processed in parallel. A representative immunoblot is shown. (B) Wild-type (WT) and *cGAS*-deficient (KO) HeLa cells were transfected with control siRNA (SC) or *SIRT4*-specific siRNA (S2). At 24 h after transfection, cells were infected with HSV-1 (MOI = 1) for 8 h. Immunoblot assays were then performed as indicated. All samples were from the same experiment and processed in parallel. A representative immunoblot is shown. (C) Wild-type (WT) and *cGAS*-deficient (KO) HeLa cells were transfected with HA-SIRT4 (+) or an empty control vector (−). At 24 h after transfection, cells were infected with HSV-1 (MOI = 1) for 8 h. Cells were then lysed for real-time PCR assays. Results are mean ± SD, $n = 3$ technical replicates. Unpaired $t$ test was used for statistical analysis. (D) Wild-type (WT) and *cGAS*-deficient (KO) HeLa cells were transfected with control siRNA (SC) or *SIRT4*-specific siRNA (S2). At 24 h after transfection, cells were infected with HSV-1 (MOI = 1) for 8 h. Cells were then lysed for real-time PCR assays. Results are mean ± SD, $n = 3$ technical replicates. Unpaired $t$ test was used for statistical analysis. (E) *Sirt2*-deficient MEFs were transfected with control siRNA (SC) or *Sirt4*-specific siRNA (S2). At 24 h after transfection, cells were infected with HSV-1 (MOI = 1) for the indicated periods. Immunoblot assays were then performed as indicated. All samples were from the same experiment and processed in parallel. A representative immunoblot is shown. Source data are available online for this figure.

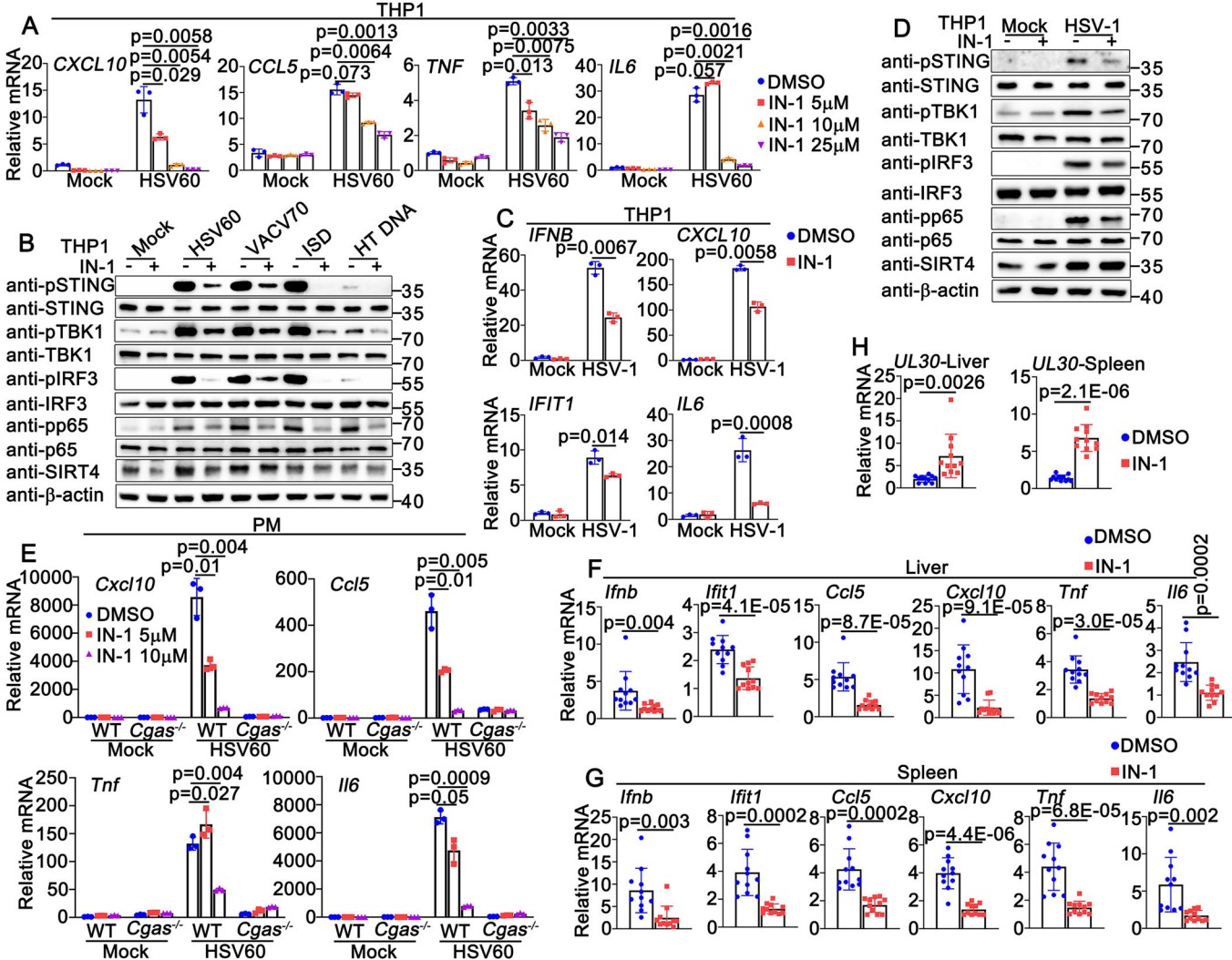

**Figure EV5. SIRT4 inhibitor regulates DNA virus- or viral DNA-triggered innate immune responses.**

(A) PMA-THP1 cells were treated with SIRT4-IN-1 (0, 5, 10, 25 μM) for 12 h, and then transfected with HSV60 (1 μg/ml) for an additional 8 h. Cells were then lysed for real-time PCR assays. Results are mean ± SD, $n = 3$ technical replicates. Unpaired $t$ test was used for statistical analysis. (B) PMA-THP1 cells were treated with SIRT4-IN-1 (10 μM) or DMSO as a control for 12 h, and then transfected with HSV60 (1 μg/ml), VACV70 (1 μg/ml), ISD (1 μg/ml), or HT DNA (1 μg/ml) for 6 h. Immunoblot assays were then performed as indicated. All samples were from the same experiment and processed in parallel. A representative immunoblot is shown. (C) PMA-THP1 cells were treated with SIRT4-IN-1 (10 μM) or DMSO as a control for 12 h, and then infected with HSV-1 (MOI = 1) for 6 h. Cells were then lysed for real-time PCR assays. Results are mean ± SD, $n = 3$ technical replicates. Unpaired $t$ test was used for statistical analysis. (D) PMA-THP1 cells were treated with SIRT4-IN-1 (10 μM) or DMSO as a control for 12 h, and then infected with HSV-1 (MOI = 1) for 6 h. Immunoblot assays were then performed as indicated. All samples were from the same experiment and processed in parallel. A representative immunoblot is shown. (E) Wild-type (WT) and *Cgas*-deficient (KO) PMs were treated with SIRT4-IN-1 (0, 5, 10 μM) for 12 h, and then transfected with HSV60 (1 μg/ml) for 6 h. Cells were then lysed for real-time PCR assays. Results are mean ± SD, $n = 3$ technical replicates. Unpaired $t$ test was used for statistical analysis. (F, G) Wild-type (WT) mice ($n = 11$, 8-week-old) were treated with SIRT4-IN-1 (0, 10 mg/kg body weight), and then intranasally infected with HSV-1 ($1 \times 10^7$ PFU) for 24 h. Livers (F) and spleens (G) were harvested and analyzed by real-time PCR. Results are mean ± SD. Each data point represents one mouse. Unpaired $t$ test was used for statistical analysis. (H) Wild-type (WT) mice ($n = 11$, 8-week-old) were treated with SIRT4-IN-1 (0, 10 mg/kg body weight), and then intranasally infected with HSV-1 ($1 \times 10^7$ PFU) for 24 h. Livers (left) and spleens (right) were harvested and analyzed by real-time PCR. Results are mean ± SD. Each data point represents one mouse. Unpaired $t$ test was used for statistical analysis. Source data are available online for this figure.

