## [Peer Review File · EMBO Reports]

SIRT4 regulates antiviral and autoimmune responses by promoting cGAS-mediated signaling pathways

Bo Yang, Yanjie Zhang, Saiyu Wang, Yufei Wu, Zilu Diao, Qunmei Zhang, Chen Lu, Mengyang Shen, Xuewei Zhang, Shujun Ma, Chunsheng Yang, Jinyong Pei, Hongxia Xing, Yinming Liang, and Jie Wang

Corresponding author(s): Jie Wang (jiewang618@xxmu.edu.cn) , Yinming Liang (yinming.liang@gris.org.cn)

Review Timeline:

Submission Date:	25th Jul 25
Editorial Decision:	18th Aug 25
Revision Received:	11th Nov 25
Editorial Decision:	5th Dec 25
Revision Received:	10th Dec 25
Editorial Decision:	19th Dec 25
Revision Received:	29th Dec 25
Accepted:	23rd Jan 26

Editor: Achim Breiling

Transaction Report:

Dear Prof. Wang,

Thank you for the submission of your manuscript to EMBO reports. I have now received reports from the three referees that were asked to evaluate your study, which can be found at the end of this email. As you will see, the referees think that these findings are of interest. However, they have several comments, concerns, and suggestions, indicating that a major revision of the manuscript is necessary to allow publication of the study in EMBO reports. As the reports are below, and all the referee concerns need to be addressed, I will not detail them here.

Given the constructive referee comments, I would thus like to invite you to revise your manuscript with the understanding that the concerns of the referees must be addressed in the revised manuscript and/or in a detailed point-by-point response. Acceptance of your manuscript will depend on a positive outcome of a second round of review. It is EMBO reports policy to allow a single round of revision only and acceptance of the manuscript will therefore depend on the completeness of your responses included in the next, final version of the manuscript.

- 1) a .docx formatted version of the final manuscript text (including legends for main figures, EV figures and tables), but without the figures included. Figure legends should be compiled at the end of the manuscript text.
- 2) individual production quality figure files as .eps, .tif, .jpg (one file per figure), of main figures and EV figures. Please upload these as separate, individual files upon re-submission.

- 4) a complete author checklist, which you can download from our author guidelines (<https://www.embopress.org/page/journal/14693178/authorguide>). Please insert page numbers in the checklist to indicate where the requested information can be found in the manuscript. The completed author checklist will also be part of the RPF.

- 5) that primary datasets produced in this study (e.g. RNA-seq, ChIP-seq, structural and array data) are deposited in an appropriate public database. If no primary datasets have been deposited, please also state this in a dedicated section (e.g. 'No

primary datasets have been generated and deposited'), see below.

The accession numbers and database should be listed in a formal "Data Availability" section that follows the model below. This is now mandatory (like the COI statement). Please note that the Data Availability Section is restricted to new primary data that are part of this study. This section is mandatory. As indicated above, if no primary datasets have been deposited, please state this in this section

Data availability

6) We now request the publication of original source data with the aim of making primary data more accessible and transparent to the reader. You will receive a separate email with instructions for providing source data with your revised manuscript, including information how to upload and organize the files.

8) Regarding data quantification and statistics, please make sure that the number "n" for how many independent experiments were performed, their nature (biological versus technical replicates), the bars and error bars (e.g. SEM, SD) and the test used to calculate p-values is indicated in the respective figure legends (also for EV and Appendix figures). Please also check that all the p-values are explained in the legend, and that these fit to those shown in the figure. Please provide statistical testing where applicable. Please avoid the phrase 'independent experiment', but clearly state if these were biological or technical replicates. Please also indicate (e.g. with n.s.) if testing was performed, but the differences are not significant. In case n=2, please show the data as separate datapoints without error bars and statistics. See also: <http://www.embopress.org/page/journal/14693178/authorguide#statisticalanalysis>

9) Please add scale bars of similar style and thickness to all microscopic images, using clearly visible black or white bars (depending on the background). Please place these in the lower right corner of the images themselves. Please do not write on or near the bars in the image but define the size in the respective figure legend.

10) Please also note our reference format:

12) We now use CRediT to specify the contributions of each author in the journal submission system. CRediT replaces the author contribution section. Please use the free text box to provide more detailed descriptions and do NOT provide your final manuscript text file with an author contributions section. See also our guide to authors: <https://www.embopress.org/page/journal/14693178/authorguide#authorshipguidelines>

13) All Materials and Methods need to be described in the main text using our 'Structured Methods' format, which is required for all research articles. According to this format, the Methods section should include a Reagents and Tools Table (listing key

reagents, experimental models, software, and relevant equipment and including their sources and relevant identifiers), uploaded as separate file, and a Methods section in which we encourage the authors to describe their methods using a step-by-step protocol format with bullet points, to facilitate the adoption of the methodologies across labs. More information on how to adhere to this format as well as downloadable templates (.doc) for the Reagents and Tools Table can be found in our author guidelines (section 'Structured Methods'):

14) Please order the manuscript sections like this, using only these names:

Title page - Abstract - Keywords - Introduction - Results - Discussion - Methods - Data availability section - Acknowledgements (please put here all the funding information) - Disclosure and Competing Interests Statement - References - Figure legends - Expanded View Figure legends

15) Please make sure that all the funding information is also entered into the online submission system and that it is complete and similar to the one in the acknowledgement section of the manuscript text file.

16) Please confirm that for all Western blot panels (main and EV figures) the loading control was run on the same gel as the other proteins detected. Please note that we discourage comparisons between samples on different gels/blots, even if the samples derive from one experiment, as confounding factors reduce comparability. If unavoidable, the figure legend must state that the samples derive from the same experiment and that gels/blots were processed in parallel. If a 'representative' loading control is shown for multiple gels/blots, the intra-gel controls should be shown in the source data files and the figure legends should describe the data displayed accurately. See our author guidelines:

<https://www.embopress.org/page/journal/14693178/authorguide#datapresentationformat> (section 'Electrophoretic gels and blots').

and

<https://www.embopress.org/image-integrity>

Thus, please also provide the source data files (uncropped images) for the Western blots shown in the EV figures. Please upload these as one ZIPed folder containing all the source data for the EV figures in separate folders.

Finally, please note that corresponding authors are required to supply an institutional e-mail address. Please provide this for co-corresponding authors Yinming Liang.

I look forward to seeing a revised form of your manuscript when it is ready.

Yours sincerely,

Referee #1:

Yang et al., this study reports that SIRT4 physically interacts with cGAS, deacetylates it, and thereby enhances cGAS DNA binding, dimerization, cGAMP production, and downstream STING-TBK1-IRF3 signaling. Functionally, SIRT4 protects against HSV-1 infection in cells and mice, while a SIRT4 inhibitor suppresses cGAS-driven signatures ex vivo (Trex1-deficient cells and SLE PBMCs). Overall, the work links SIRT4 to cGAS regulation and innate immunity, highlighting its potential as a disease-relevant node. Several mechanistic claims still need stronger experimental support, and some conclusions should be presented

more cautiously.

Major comments:

1. The therapeutic implications are confusing. In the HSV-1 model, Sirt4 KO clearly worsens infection, showing SIRT4 is protective. Yet in Trex1-deficient cells and SLE PBMCs, SIRT4 inhibition is presented as beneficial. The authors should discuss this apparent contradiction and explain how SIRT4 can be both protective (in infection) and pathogenic (in autoimmunity).
2. Sirtuin activity is NAD⁺-dependent, and many studies show that NAD⁺ supplementation is generally beneficial for health and host defense. The authors should discuss how their findings fit with this broader literature, since their model seems to suggest the opposite effect.
3. In Fig. 1A, acetyl-cGAS is compared but total cGAS also increases. Acetylation should be normalized to immunoprecipitated cGAS.
4. In Fig. 5B, the 24-h cGAMP rescue is too long to argue that SIRT4 acts upstream of cGAMP. Shorter time points (1-2 h) are needed.
5. Fig. 4H and 10K: lung H&E suggests more infiltration in KO mice, staining or quantification of macrophages would strengthen the data.
6. In Fig. 4, the infection phenotype in Sirt4^{-/-} mice is interpreted as consistent with impaired cGAS signaling, but this is not directly demonstrated. To strengthen this link, the authors could add pathway readouts in vivo (p-STING or p-TBK1 in lung tissue, or WB) to show that cGAS-STING signaling is indeed blunted in vivo.
7. To show directly that SIRT4 removes acetyl groups from cGAS, the authors should test purified SIRT4 on acetylated cGAS in vitro. Including conditions with and without NAD⁺, and with a catalytic mutant (H161Y), would make the evidence more solid.

Minor comments:

1. Introduction should also mention aging-related cGAS activation.
2. The statement 'no sirtuin reported to deacetylate cGAS' should be revised, as Barthez et al., Cell Reports 2025, reported a direct interaction between SIRT2 and cGAS. Please update the text to reflect this.
3. Quantification of WB bands is missing in some key figures.
4. Fig. 4I: IFN-β shows a downward trend, but with n=3 it is difficult to interpret; additional mice would likely clarify significance.
5. Fig. 9H: in the 3KR mutant, acetyl-cGAS is already very low, making further "decrease" difficult to interpret.
6. In the Discussion, update the overview of sirtuins in innate immunity to include Barthez et al. Cell Reports 2025 (SIRT2-cGAS) alongside SIRT1/3/5 family reports.

Referee #2:

The manuscript titled "SIRT4 regulates antiviral and autoimmune responses by promoting cGAS mediated signaling pathway" by Bo Yang et al report SIRT4 as a new regulator for cGAS and explored its function in several experimental models. The authors find that SIRT4 specially interacted with cGAS and mediated its deacetylation, which leads to the enhancement of the association between cGAS with dsDNA. The authors also used genetic and biochemical approaches to demonstrate the specific regulatory role of SIRT4 in the cGAS-Sting pathway. Overall, the findings of this manuscript meet the scope of EMBO Reports, are innovative to some extent, and provide sufficient evidence to support the conclusions.

The reviewer supports the publication of this article in EMBO Reports, but there are some issues that need to be addressed.

1. The Sirtuin family has several members that have been reported to play roles in the cGAS-STING pathway, such as Sirt2, Sirt3 and Sirt6 (EMBO Rep. 2023 Dec 6;24(12):e57500. Redox Biol. 2024 Aug;74:103224. Cell Metab. 2019 Apr 2;29(4):871-885.e5. J Biol Chem. 2024 Aug;300(8):107554.). Although their targets may be not cGAS, the authors need to summarize the roles of this family in the cGAS-STING signaling pathway in the Introduction section in details. Additionally, the authors need to further demonstrate the differences between the actions of Sirt4 and those of other members of this family.
2. The binding experiments conducted by the authors were mainly based on an overexpression system. The authors need to investigate whether cGAS interacts with and is modified by other members of the SIRT family during viral infection. Additionally, the authors need to investigate the co-localization of SIRT4 and cGAS during the process of viral infection.
3. SIRT4 is primarily localized in the mitochondria. So, how does it interact with cGAS on the mitochondria? Is the assistance of other proteins required? Given that cGAS is primarily located in the cytoplasm, what fraction of the total cGAS pool is represented by the mitochondrial-localized cGAS?
4. The results of Figure 2 and Figure 3 should be combined because they address the same issue, which would make the presentation more concise.
5. The authors mentioned in the last part that the SIRT4 inhibitor regulates autoimmune responses. The reviewer suggests that the authors test the effects of the SIRT4 inhibitor or SIRT4 knockdown/knockout in an animal model, such as an SLE animal model. This work would not take a long time but could increase the reliability of the conclusions.
6. The authors need to add the molecular weights of the proteins in their Figures.

Referee #3:

The manuscript „ SIRT 4 regulates antiviral and autoimmune responses by promoting cGAS mediated signaling pathway" describes the identification of the lysine deacetylase SIRT4 as a novel regulator of cGAS activity. The authors show that SIRT4, but not other sirtuin family members, can remove acetyl-groups from key lysine residues and thus facilitate cGAS activity, both in the human and murine system.

The work presented here is situated in a busy field of research and will be of interest to many cell biology, immunology, and virology researchers. These data contribute to our knowledge of the multitude of molecular mechanisms that govern and regulate cGAS activity. The exploratory experiments by the authors regarding treatment of sterile inflammatory diseases, such as SLE, by inhibition of SIRT4 further extends the readership of the manuscript.

The authors show high-quality data that generally well-support the claims made. However, a few points need to be addressed:

Major points:

1. In the paper originally describing the acetylation of three lysine residues (K384, K394 or K414) of human cGAS to regulate its activity (Dai et al. 2019), these authors showed the histone deacetylase HDAC3 to be responsible for deacetylating cGAS in THP1 cells. Ablation of HDAC3 expression via siRNAs prevented de-acetylation on protein level and substantially reduced cGAMP production. This is in direct contrast with the results reported here, where SIRT4 knockout in THP1 cells reduces cGAMP production to the same extent (compare Fig. S3E in Dai et al. and Fig. 8B this manuscript). The authors ought to in the very least address this discrepancy in their discussion, but really should address this experimentally by repeating experiments using siRNAs against SIRT4 and HDAC3 in comparison. Here it would be good to include transfection agent-only conditions, since lipofectamine treatment alone can have effects on cGAS pathway activation. In addition, the authors should compare the effects of the SIRT4 inhibitor they used with small molecules targeting HDAC3.
2. In Figure 4D, Sirt4-KO mice show reduced IFN- β in BALF after HSV-1 infection when compared to WT mice. The same analysis after VSV infection shows very similar results (Fig. 4I). While statistical significance differs, with three data points (should be 4 in 4D?) the results do not allow the conclusion that RNA virus-induced interferon activation is unaffected by SIRT4. The authors need to change their wording and acknowledge that IFN responses to VSV might be affected by SIRT4 as well, albeit indirectly. If these data/samples exist, they should show data on IFN gene expression in tissues and cytokine levels in sera from mice treated with VSV, similar to what is shown in most of Fig. 4 for HSV-1.
3. Control experiments, where cells are treated with cGAMP, are only shown for murine cells (Fig. 5). The authors should demonstrate that THP1 cells and HaCaT cells, WT and SIRT4 KO/overexpression, also display unchanged responses to STING agonist treatment.

Minor points:

4. Related to point 1, Dai et al. used treatment of cells with aspirin to enforce increased cGAS acetylation and thus reduced cGAS activity. It would be very informative if the authors could compare the SIRT4 inhibitor they used with aspirin treatment in the context of autoimmune responses (Fig. 11).
5. The data in Fig. 8A and 8B are not referenced anywhere in the results text.
6. In Figure 9I, overexpression of WT cGAS in the absence of SIRT4 leads to phosphorylation of IRF3 (i.e. PRR pathway activation), while expression of the 3KR mutant does not. In contrast, indistinguishable IFNB1 transcript levels can be observed for both constructs in Fig. 9J. Could the authors comment on this discrepancy?
7. The manuscript would benefit from some grammatical and semantic proof-reading to ease understanding in certain instances. The authors should make sure that they refer to SIRT4 as a "deacetylase" and not "deacylase" as these are two distinct processes. In addition, the authors should be consistent in correctly referring to human and mouse gene names and italicize where necessary. Many figure panels that describe mouse data have gene names in all capital letters (i.e. human). Where mouse gene names are used, they should start with a capital letter.
8. The figure legends need to be more precise when describing how many times experiments were performed independently, and whether technical replicates of one experiment or pooled data are shown. What is written under statistics in the methods section is not always congruent with figure legends. Regarding mouse experiments, were they repeated? If yes, the authors should show data from all individual animals.
9. The authors ought to describe their methodology in more detail. For many assays, only a reference but no technical detail is provided. This is not sufficient for being able to reproduce the data shown here. For the RT-qPCR primers used in the study, a reference is missing entirely.
10. In the introduction, IKK-epsilon is referred to as IKKi, whereas it is called IKKe in the graphical abstract. This should be consistent.

Referee #1:

Yang et al., this study reports that SIRT4 physically interacts with cGAS, deacetylates it, and thereby enhances cGAS DNA binding, dimerization, cGAMP production, and downstream STING-TBK1-IRF3 signaling. Functionally, SIRT4 protects against HSV-1 infection in cells and mice, while a SIRT4 inhibitor suppresses cGAS-driven signatures ex vivo (Trex1-deficient cells and SLE PBMCs). Overall, the work links SIRT4 to cGAS regulation and innate immunity, highlighting its potential as a disease-relevant node. Several mechanistic claims still need stronger experimental support, and some conclusions should be presented more cautiously.

Major comments:

1. The therapeutic implications are confusing. In the HSV-1 model, Sirt4 KO clearly worsens infection, showing SIRT4 is protective. Yet in Trex1-deficient cells and SLE PBMCs, SIRT4 inhibition is presented as beneficial. The authors should discuss this apparent contradiction and explain how SIRT4 can be both protective (in infection) and pathogenic (in autoimmunity).

Response: The reviewer's suggestion is very good and has been well taken. We added the following sentence to the **Discussion** section.

“The regulatory functions of SIRT4 in different disease models, its protective role

during viral infection, and its pathogenic effects in autoimmunity, illustrate that the innate immune response is a double-edged sword. It protects the host against pathogen invasion, but its abnormal and persistent activation in the body can be harmful, thus requiring precise regulation.”

2. Sirtuin activity is NAD⁺-dependent, and many studies show that NAD⁺ supplementation is generally beneficial for health and host defense. The authors should discuss how their findings fit with this broader literature, since their model seems to suggest the opposite effect.

Response: The reviewer`s suggestion is very good and has been well taken. We added the following sentence to the **Discussion** section.

“The activity of SIRT family molecules depends on NAD⁺. However, considering the functions of SIRT4 in antiviral and autoimmune processes, as well as the distinct roles of different SIRT family members in innate immunity, we may need to exercise caution when using NAD⁺ supplements, particularly in patients with autoimmune-related diseases.”

3. In Fig. 1A, acetyl-cGAS is compared but total cGAS also increases. Acetylation should be normalized to immunoprecipitated cGAS.

Response: The reviewer's suggestion is very good and has been well taken. Acetylation has been normalized to immunoprecipitated cGAS in Fig 1A.

Fig 1A

Fig 1 SIRT4 interacts with cGAS

(A) PMA-THP1 cells were infected with HSV-1 (MOI=1) for the indicated periods, with NAM (5mM) or DMSO as control. Afterward, the cells were lysed and subjected to immunoprecipitation (IP) and immunoblot (IB) analysis.

4. In Fig. 5B, the 24-h cGAMP rescue is too long to argue that SIRT4 acts upstream of cGAMP. Shorter time points (1-2 h) are needed.

Response: The reviewer's suggestion is very good and has been well taken. We first attempted to detect IFN- β levels by ELISA in wild-type mouse BMDMs transfected with HSV60 and cGAMP for 2 hours. The results showed that, compared to unstimulated conditions, the increase in IFN- β after 2 hours of stimulation was very limited, close to the lower detection limit of the ELISA kit and almost undetectable (Fig N1). Under this condition, comparing the effect of *Sirt4* deficiency on IFN- β secretion might be inaccurate. Considering the process of IFN- β secretion, it first requires the

activation of signaling pathways, including the phosphorylation of STING, TBK1, IRF3, and p65, followed by IFN- β transcription, and finally protein production and secretion. Therefore, the time point for ELISA detection is generally later, while the detection time points for phosphorylation and IFN- β RNA levels are earlier. Thus, we proceeded to examine the activation of signaling pathways and the transcriptional level of IFN- β . The results showed that at 1 or 2 hours of stimulation, *SIRT4* deficiency significantly impaired HSV60-induced activation of the innate immune signaling pathway and IFN- β production at the transcriptional level, but had no significant effect on cGAMP-induced responses (Appendix Fig S3C-D). Additionally, we observed similar results in HaCaT and THP1 cells (Fig N2A-B). In all, we added the following sentence to the **Results** section.

“However, in both human and mouse cells, cGAMP- or RNA viruses-triggered innate immune responses were not affected much by the impairment of SIRT4 expression (Fig 4A, 4F-G, EV2E, EV3B, EV3D, EV3G, and Appendix Fig S1D-F, and S3C-D), suggesting that SIRT4 specifically affected DNA virus-induced signaling pathway, and its target might be in the upstream of cGAMP.”

Fig N1

Fig N1 The induction of IFN- β by HSV60 and cGAMP in BMDMs

BMDMs were transfected with HSV60 (1 μ g/ml), or cGAMP (1 μ g/ml) for 2 h. The supernatants were collected and subjected to ELISA assays.

Appendix Fig S3C-D

Appendix Fig S3 *Sirt4* deficiency reduces antiviral innate immune responses against DNA virus in BMDMs and BMDCs

(C) Wild-type (WT) and *Sirt4*-deficient (KO) BMDMs were transfected with HSV60 (1 μ g/ml), VACV70 (1 μ g/ml), ISD (1 μ g/ml), or cGAMP (1 μ g/ml) for the indicated

periods. Then the cells were lysed for real-time PCR assays.

(D) Wild-type (WT) and *Sirt4*-deficient (KO) BMDMs were transfected with HSV60 (1 µg/ml) (Top), or cGAMP (1 µg/ml) (Bottom) for the indicated periods. The cells were then lysed for immunoblot assays.

Fig N2

Fig N2 Fig N2 SIRT4 did not significantly affect the response triggered by cGAMP

(A) HaCaT keratinocytes were transfected with the empty vector (Vec) or the SIRT4 expressing plasmid for 24 h and then transfected with HSV60 (1 µg/ml) or cGAMP (1 µg/ml) for the indicated periods. The cells were then lysed for immunoblot assays.

(B) Wild-type (WT) and *SIRT4*-deficient (KO) PMA-THP1 cells were transfected with HSV60 (1 µg/ml) or cGAMP (1 µg/ml) for the indicated periods. The cells were then

lysed for immunoblot assays.

5. Fig. 4H and 10K: lung H&E suggests more infiltration in KO mice, staining or quantification of macrophages would strengthen the data.

Response: The reviewer`s suggestion is very good and has been well taken. We have stained CD68 to evaluate the infiltration of macrophage in lungs and added the results to the **Appendix Figures**. As shown in Appendix Fig S2E and S4D, the CD68 staining results are consistent with the HE staining results. We added the following sentences to the **Results** section to describe these results.

“Further, as suggested by the H&E and CD68 staining assays, in response to HSV-1 infection, *Sirt4*-deficient mice exhibited more severe destructions and macrophage infiltration in lungs than wild-type mice (Fig 3H and Appendix Fig S2E).”

“Histopathological analysis of lung tissue sections also revealed that SIRT4-IN-1 exacerbated the severity of lung damage with more macrophage infiltration caused by HSV-1 infection (Fig 7L and Appendix Fig S4D).”

Appendix Fig S2E

Appendix Fig S2 SIRT4 protects mice from HSV-1 infection

(E) Sex and age-matched wild-type (WT) and *Sirt4*-deficient (KO) mice were intravenously infected with HSV-1 (1×10^7 PFU) for 24 h and lung sections were analyzed by CD68 staining. Scale bar, 100 μ m.

Appendix Fig S4D

Appendix Fig S4 SIRT4 deacetylates cGAS

(D) Wild-type (WT) mice were treated with SIRT4 inhibitor SIRT4-IN-1 (0, 10 mg/kg body weight), and then intranasally infected with HSV-1 (1×10^7 PFU) for 24 h. The lung sections were analyzed by H&E staining. Scale bar, 100 μm .

6. In Fig. 4, the infection phenotype in *Sirt4*^{-/-} mice is interpreted as consistent with impaired cGAS signaling, but this is not directly demonstrated. To strengthen this link, the authors could add pathway readouts in vivo (p-STING or p-TBK1 in lung tissue, or WB) to show that cGAS-STING signaling is indeed blunted in vivo.

Response: The reviewer's suggestion is very good and has been well taken. We examined the activation of signaling pathways in tissues from wild-type and *Sirt4*-deficient mice and found that after HSV-1 infection, the phosphorylation levels of STING and TBK1 in the livers, spleens, and lungs of *Sirt4*-deficient mice were significantly lower than those in wild-type mice (Appendix Fig S2D, and Fig N3). We added the following sentence to the **Results** section.

“Consistently, in a variety of organs, including liver, spleen, lung, and brain, the impaired innate immune responses were observed in *Sirt4*-deficient mice upon HSV-1 infection, compared to wild-type mice (Fig 3E-G, and Appendix Fig S2D).”

Appendix Fig S2D

Appendix Fig S2 SIRT4 protects mice from HSV-1 infection

(D) Wild-type (WT) and *Sirt4*-deficient (KO) mice (n=10) were intranasally infected with HSV-1 (1×10^7 PFU) for 24 h and then the livers, lungs, and spleens of the mice were subjected to immunoblot assays (Left). The levels of p-STING and p-TBK1 were normalized to total STING and TBK1, respectively (Right).

Fig N3

Fig N3 SIRT4 protects mice from HSV-1 infection

Wild-type (WT) and *Sirt4*-deficient (KO) mice were intranasally infected with HSV-1 (1×10^7 PFU) for 24 h and then the livers, lungs, and spleens of the mice were subjected to immunoblot assays.

7. To show directly that SIRT4 removes acetyl groups from cGAS, the authors should test purified SIRT4 on acetylated cGAS *in vitro*. Including conditions with and without NAD^+ , and with a catalytic mutant (H161Y), would make the evidence more solid.

Response: The reviewer's suggestion is very good and has been well taken. We examined the effect of purified SIRT4 on acetylated cGAS in an *in vitro* deacetylation

assay. As expected, in the presence of NAD⁺, the acetylation of immunopurified cGAS was decreased by *in vitro* translated SIRT4, but not by its catalytic mutant (H161Y) (Appendix Fig S4A-B). We added the following sentence to the **Results** section to describe these results.

“These phenomena were verified in an *in vitro* acetylation system (Appendix Fig S4A-B).”

Appendix Fig S4A-B

Appendix Fig S4 SIRT4 deacetylates cGAS

(A) HEK293T cells were transfected with Flag-cGAS. At 24 h after transfection, the cells were lysed and immunopurified cGAS was incubated with or without *in vitro*

translated SIRT4, in the presence of NAD⁺ or not, followed by western blot analysis of lysine acetylation.

(B) HEK293T cells were transfected with Flag-cGAS. At 24 h after transfection, the cells were lysed and immunopurified cGAS was incubated with or without *in vitro* translated SIRT4 or its H161Y mutant, in the presence of NAD⁺, followed by western blot analysis of lysine acetylation.

Minor comments:

1. Introduction should also mention aging-related cGAS activation.

Response: The reviewer's suggestion is very good and has been well taken. The following sentences have been added to the **Introduction** section.

“In addition, the activation of cGAS-STING signaling is engaged in ageing-related inflammation and neurodegeneration (Gulen et al, 2023).

Considering the essential role of cGAS in antiviral responses, autoimmunity, neurodegeneration, and ageing-related decline, it is not surprising that the ligand-binding ability, activity and stability of cGAS is tightly regulated by various posttranslational modifications (PTMs) to avoid aberrant activation (Zahid et al, 2020).”

2. The statement 'no sirtuin reported to deacetylate cGAS' should be revised, as Barthez et al., Cell Reports 2025, reported a direct interaction between SIRT2 and cGAS. Please update the text to reflect this.

Response: The reviewer`s suggestion is very good and has been well taken. We deleted this sentence and added the following sentence to the **Introduction** section.

“Later, Barthez et al. reported that SIRT2 interacts with cGAS directly, deacetylates cGAS, and suppresses ageing-associated cGAS activation and inflammation (Barthez et al, 2025).”

3. Quantification of WB bands is missing in some key figures.

Response: The reviewer`s suggestion is very good and has been well taken. Quantification of WB bands has been done in key figures Fig 1A, 2G, 5E, 6C-D, 6H, 7A-B, and Appendix Fig S2D.

Fig 1A

Fig 1 SIRT4 interacts with cGAS

(A) PMA-THP1 cells were infected with HSV-1 (MOI=1) for the indicated periods, with NAM (5mM) or DMSO as control. Afterward, the cells were lysed and subjected to immunoprecipitation (IP) and immunoblot (IB) analysis.

Fig 2G

Fig 2 SIRT4 positively regulates antiviral innate immune responses against DNA virus

(G) Wild-type (WT) and *SIRT4*-deficient (KO) PMA-THP1 cells were infected with HSV-1 (MOI=1) for 24 h. Immunoblot assays were performed.

Fig 5E

Fig 5 SIRT4 promotes the activation of cGAS

(E) HEK293T cells were transfected with indicated plasmids for 24 h and then the cell lysates were subjected to immunoprecipitation (IP) and immunoblot (IB) assays as indicated.

Fig 6C-D

Fig 6 SIRT4 deacetylates cGAS

(C) PMA-THP1 cells were transfected with control siRNA (SC) or *SIRT4*-specific siRNA (S2). At 24 h after transfection, the cells were infected with HSV-1 (MOI=1) for the indicated periods. The cells were lysed and subjected to immunoprecipitation (IP) and immunoblot (IB) assays as indicated.

(D) Wild-type (WT) and *SIRT4*-deficient (KO) PMA-THP1 cells were infected with HSV-1 (MOI=1) for the indicated periods. The cells were lysed and subjected to immunoprecipitation (IP) and immunoblot (IB) assays as indicated.

Fig 6H

Fig 6 SIRT4 deacetylates cGAS

(H) HeLa cells were transfected with indicated plasmids for 24 h and then the cell lysates were subjected to immunoprecipitation (IP) and immunoblot (IB) assays as indicated.

Fig 7A-B

Fig 7 SIRT4 inhibitor regulates DNA virus- or viral DNA-triggered innate immune responses

(A) HeLa cells were transfected with HA-cGAS and treated with SIRT4 inhibitor SIRT4-IN-1 (0, 10, 25 μ M). At 24 h after transfection, the cells were transfected with HSV60 (1 μ g/ml) for another 8 h. The cells were lysed and subjected to immunoprecipitation (IP) and immunoblot (IB) assays.

(B) HeLa (top) or PMA-THP1 (bottom) cells were treated with SIRT4 inhibitor SIRT4-IN-1 (0, 10, 25 μ M) for 12 h, and then transfected with HSV60 (1 μ g/ml) for another 8 h. The cells were lysed and subjected to immunoprecipitation (IP) and immunoblot (IB) assays.

Appendix Fig S2D

Appendix Fig S2 SIRT4 protects mice from HSV-1 infection

(D) Wild-type (WT) and *Sirt4*-deficient (KO) mice (n=10) were intranasally infected with HSV-1 (1×10^7 PFU) for 24 h and then the livers, lungs, and spleens of the mice were subjected to immunoblot assays (Left). The levels of p-STING and p-TBK1 were normalized to total STING and TBK1, respectively (Right).

4. Fig. 4I: IFN- β shows a downward trend, but with n=3 it is difficult to interpret; additional mice would likely clarify significance.

Response: The reviewer's suggestion is very good and has been well taken. The results shown in Figure 4I were actually repeated three times, and this is just one of the results. Due to the limited number of data points, a trend may appear to exist. We have compiled the results from all three replicates in all mice as the new Fig 3I. Furthermore, to further verify this phenomenon, we examined the expression of IFN- β and inflammatory factors in the serum and tissues after SeV infection. The results, as shown in Appendix Fig S2F-H, did not reveal any significant effect of SIRT4 deficiency on the innate immune response in SeV-infected mice. We added the following sentences to the **Results** section to describe these results.

“However, we did not observe significant difference in innate host defense between wild-type and *Sirt4*-deficient mice upon VSV infection (Fig 3I, and Appendix Fig S2F-H), suggesting SIRT4 does not regulate RNA virus-triggered antiviral innate

immune responses in mice.”

Fig 3I

Fig 3 SIRT4 protects mice from HSV-1 infection

(I) Wild-type (WT) and *Sirt4*-deficient (KO) mice (n = 9, 8-week-old) were intranasally infected with VSV (5×10^7 PFU) for 24 h. Bronchoalveolar lavage fluid (BALF) was collected and ELISA assays were performed.

Appendix Fig S2F-H

Appendix Fig 2 SIRT4 protects mice from HSV-1 infection

(F) ELISA of IFN-β, and IL-6 in serum of wild-type (WT) and *Sirt4*-deficient (KO)

mice (n=7, 8-week-old) 6 h after intravenous infection with VSV (5×10^7 PFU).

(G, H) Wild-type (WT) and *Sirt4*-deficient (KO) mice (n=10) were intranasally infected with VSV (5×10^7 PFU) for 24 h and then the lungs, livers, and spleens of the mice were subjected to ELISA assays (G) or real-time PCR assays (H).

5. Fig. 9H: in the 3KR mutant, acetyl-cGAS is already very low, making further "decrease" difficult to interpret.

Response: The reviewer's suggestion is very good and has been well taken. In the 3KR mutant, the key acetylation-related lysine residues are mutated, resulting in a significant decrease in overall acetylation levels of cGAS. To more clearly demonstrate the effect of SIRT4 on the acetylation of the 3KR mutant, we has quantified the acetylation in the new Fig 6H (the previous Fig 9H).

Fig 6H

Fig 6 SIRT4 deacetylates cGAS

(H) HeLa cells were transfected with indicated plasmids for 24 h and then the cell

lysates were subjected to immunoprecipitation (IP) and immunoblot (IB) assays as indicated.

6. In the Discussion, update the overview of sirtuins in innate immunity to include Barthez et al. Cell Reports 2025 (SIRT2-cGAS) alongside SIRT1/3/5 family reports.

Response: The reviewer`s suggestion is very good and has been well taken. The SIRT2-cGAS information has been updated in the following sentence in the **Discussion** section.

“Recently, accumulating evidence suggests that a few deacetylase Sirtuin family members are involved in antiviral innate immune responses, including SIRT1 targeting IRF3 and IFI16 (Qin et al, 2022; Wang et al, 2023), SIRT2 targeting G3BP1 and cGAS (Li et al., 2023; Barthez et al, 2025), SIRT3 targeting MAVS (Liu et al, 2024), and SIRT5 targeting DDX3 and MAVS (He et al, 2021; Liu et al, 2020).”

Referee #2:

The manuscript titled "SIRT4 regulates antiviral and autoimmune responses by promoting cGAS mediated signaling pathway" by Bo Yang et al report SIRT4 as a new regulator for cGAS and explored its function in several experimental models. The

authors find that SIRT4 specially interacted with cGAS and mediated its deacetylation, which leads to the enhancement of the association between cGAS with dsDNA. The authors also used genetic and biochemical approaches to demonstrate the specific regulatory role of SIRT4 in the cGAS-Sting pathway. Overall, the findings of this manuscript meet the scope of EMBO Reports, are innovative to some extent, and provide sufficient evidence to support the conclusions.

The reviewer supports the publication of this article in EMBO Reports, but there are some issues that need to be addressed.

1. The Sirtuin family has several members that have been reported to play roles in the cGAS-STING pathway, such as Sirt2, Sirt3 and Sirt6 (EMBO Rep. 2023 Dec 6;24(12):e57500. Redox Biol. 2024 Aug;74:103224. Cell Metab. 2019 Apr 2;29(4):871-885.e5. J Biol Chem. 2024 Aug;300(8):107554.). Although their targets may be not cGAS, the authors need to summarize the roles of this family in the cGAS-STING signaling pathway in the Introduction section in details. Additionally, the authors need to further demonstrate the differences between the actions of Sirt4 and those of other members of this family.

Response: The reviewer`s suggestion is very good and has been well taken. We have added the following sentence to the **Introduction** section to summarize the roles of this family in the cGAS-STING signaling pathway.

“A few SIRT6s have been reported to be involved in cGAS-STING signaling pathway, including SIRT2, SIRT3, and SIRT6 (Guo et al, 2024; Li et al., 2023; Simon et al, 2019; Zhou et al, 2024).”

In addition, we have added the following sentence to the **Introduction** section to further demonstrate the differences between the actions of SIRT4 and those of other members of this family.

“Early studies suggested that among the seven members of the SIRT family, SIRT1, SIRT2, and SIRT3 are robust deacetylases, while SIRT6 and SIRT7 also exhibit deacetylase activity. SIRT5 primarily catalyzes lysine desuccinylation, demalonylation, and deglutarylation. In contrast, SIRT4 lacks detectable histone deacetylase activity but possesses substrate-specific deacetylase activity (Kumar & Lombard, 2017). The absence of robust enzymatic activity posed challenges for studying the biological functions or developing modulators of SIRT4 (Li et al., 2018). However, recent studies have increasingly reported that SIRT4 can regulate critical physiological and pathological processes by directly deacetylating specific target proteins (Lv et al, 2025; Yu et al, 2025; Zhang et al, 2022). Moreover, specific inhibitors of SIRT4 have been identified, shedding light on the functional study of SIRT4 and suggesting its potential as a novel drug target (Pannek et al, 2024).”

2. The binding experiments conducted by the authors were mainly based on an overexpression system. The authors need to investigate whether cGAS interacts with and is modified by other members of the SIRT family during viral infection. Additionally, the authors need to investigate the co-localization of SIRT4 and cGAS during the process of viral infection.

Response: The reviewer`s suggestion is very good and has been well taken. We have revised and improved the manuscript in the following three aspects:

(1) We have supplemented the endogenous co-immunoprecipitation results of SIRT4 and cGAS in HeLa and PMA-THP1 cells under HSV-1 stimulation (Fig 1F). We added the following sentences to the **Results** section to describe these results.

“Furthermore, endogenous SIRT4-cGAS interaction was indicated by co-immunoprecipitation assays in HeLa and PMA-THP1 cells with HSV-1 infection or HSV60 transfection (Fig 1F).”

Fig 1F

Fig 1 SIRT4 interacts with cGAS

(F) PMA-THP1 (Top) or HeLa (Bottom) cells were infected with HSV-1 (MOI=1) for the indicated periods and then the cell lysates were subjected to immunoprecipitation (IP) and immunoblot (IB) assays as indicated.

(2) We investigated whether other SIRT4s interacted with endogenous cGAS and affected acetylation of endogenous cGAS during HSV-1 infection. As shown in Appendix Fig S1A, SIRT1-7 were transfected into HeLa cells, with or without HSV-1 infection and the co-IP results indicated that only SIRT4 exhibited a clear interaction with endogenous cGAS. It seemed that under unstimulated conditions, SIRT2 interacted very weakly to endogenous cGAS, but after HSV-1 stimulation, this binding became even less detectable (Appendix Fig S1A). Additionally, under unstimulated conditions, only SIRT4 effectively reduced the acetylation of endogenous cGAS (Appendix Fig S4C), which was consistent with the report by Li et al. that SIRT2 cannot directly deacetylate cGAS (EMBO Reports 2023, e57500). After HSV-1 infection, SIRT2 and SIRT5 appeared to reduce cGAS acetylation to some extent, but the extent of deacetylation by SIRT4 was more significantly (Appendix Fig S4C). Barthez et al.

have reported that SIRT2 binds to cGAS and can deacetylate it (Cell Reports 44, 115562, April 22, 2025). We notice that these results were obtained from HEK293T cells stimulated with HT DNA. Considering that cGAS and STING are generally thought to be expressed at low levels in HEK293T cells, it is possible that SIRT2's function in this context is cell-type or stimulus-specific. In summary, within our experimental system, among SIRT1-7, only SIRT4 can clearly bind to cGAS and deacetylate it, both with and without HSV-1 stimulation. However, it cannot be ruled out that in specific cell types or under specific stimulation conditions, other SIRT family proteins may also bind to cGAS and deacetylate it. The following sentences in the **Results** section have been revised to describe these findings.

“As shown in Fig 1B-C and Appendix Fig S1A, only SIRT4 co-immunoprecipitated with cGAS.”

“We investigated whether other SIRTs affected acetylation of endogenous cGAS and found that only SIRT4 effectively reduced the acetylation of endogenous cGAS with or without HSV-1 infection (Appendix Fig S4C).”

Appendix Fig S1A

Appendix Fig S1 SIRT4 interacts with cGAS and positively regulates antiviral innate immune responses against DNA virus

(A) HeLa cells were transfected with various combinations of plasmids as indicated. 24 h later, the cells were infected with HSV-1 (MOI=1) for 4 h. The cells were then lysed for immunoprecipitation (IP) and immunoblot (IB) analysis.

Appendix Fig S4C

Appendix Fig S4 SIRT4 deacetylates cGAS

(C) HeLa cells were transfected with various combinations of plasmids as indicated. 24

h later, the cells were infected with HSV-1 (MOI=1) for 4 h. The cells were then lysed for immunoprecipitation (IP) and immunoblot (IB) analysis.

(3) We have investigated the co-localization of SIRT4 and cGAS during the process of HSV-1 infection. As shown in Fig N4 and Fig 1H, SIRT4 co-localized with cGAS with or without HSV-1 infection, and SIRT4-cGAS could form puncta after HSV-1 infection, consistent with previously reported results indicating that cGAS exhibits liquid-liquid phase separation (LLPS) with DNA (Shi et al. The EMBO Journal e109272 | 2022, Du & Chen Science 2018). The relationship of this interaction with mitochondria will be addressed in detail in the next question.

Fig N4

Fig N4 SIRT4 co-localizes with cGAS

HeLa cells were infected with HSV-1 (MOI=1) for 2 h. Immunofluorescence assays

were performed using anti-cGAS (green) and anti-SIRT4 (purple). Nuclei were stained with DAPI. Mitochondria were stained with MitoTracker. Scale bar, 10 μ m.

Fig 1H

Fig 1 SIRT4 interacts with cGAS

(H) HeLa cells were infected with HSV-1 (MOI=1) for 2h. Immunofluorescence assays were performed using anti-cGAS (green) and anti-SIRT4 (purple). Nuclei were stained with DAPI. Mitochondria were stained with MitoTracker. Scale bar, 10 μ m. Arrows indicate cGAS puncta.

3. SIRT4 is primarily localized in the mitochondria. So, how does it interact with cGAS on the mitochondria? Is the assistance of other proteins required? Given that cGAS is primarily located in the cytoplasm, what fraction of the total cGAS pool is represented by the mitochondrial-localized cGAS?

Response: The reviewer's suggestion is very good and has been well taken. We

addressed this question from the following three aspects:

(1) We isolated mitochondria using a mitochondrial extraction kit and then assessed the distribution of cGAS and SIRT4 in the mitochondrial and cytosolic fractions. Western blot results showed that in HSV-1-infected THP-1 cells, SIRT4 was primarily localized to the mitochondria, while cGAS was distributed in both the mitochondrial and cytosolic fractions (Appendix Fig S1B-C). In Appendix Fig S1B, the mitochondrial fraction was loaded at 5 times the amount of the cytosolic fraction. Based on our estimation, approximately 20-40% of the total cGAS was localized in mitochondria. Similarly, in HSV-1-infected HeLa cells, both cGAS and SIRT4 were detected in both mitochondrial and cytosolic fractions (Appendix Fig S1C). With the mitochondrial fraction loaded at 5 times the cytosolic load in Appendix Fig S1C, we estimated that about 15-25% of the total cGAS was localized in mitochondrial. However, according to previous reports and our confocal microscopy results in Fig 1H, a fraction of cGAS is also localized in the nucleus. Therefore, the actual proportion of cGAS associated with mitochondria is likely even lower.

Appendix Fig S1

Appendix Fig S1 SIRT4 interacts with cGAS and positively regulates antiviral innate immune responses against DNA virus

(B-C) PMA-THP1 (B) or HeLa (C) cells were infected with HSV-1 (MOI=1) for the indicated periods. Mitochondria isolated from the cells were subjected to immunoblot assays as indicated.

Fig 1H

Fig 1 SIRT4 interacts with cGAS

(H) HeLa cells were infected with HSV-1 (MOI=1) for 2h. Immunofluorescence assays

were performed using anti-cGAS (green) and anti-SIRT4 (purple). Nuclei were stained with DAPI. Mitochondria were stained with MitoTracker. Scale bar, 10 μ m. Arrows indicate cGAS puncta.

(2) We performed co-immunoprecipitation experiments using mitochondrial extracts from HSV-1-infected THP1 and HeLa cells. As shown in Fig 1I, cGAS and SIRT4 co-immunoprecipitated from the mitochondrial fraction upon HSV-1 stimulation.

Figure 1I

Fig 1 SIRT4 interacts with cGAS

(I) HeLa cells were infected with HSV-1 (MOI=1) for the indicated periods. Mitochondria isolated from the cells were subjected to immunoprecipitation (IP) and immunoblot (IB) assays as indicated.

(3) We investigated the co-localization of SIRT4 and cGAS during the process of HSV-

1 infection. As shown in Fig N4 and Fig 1H, SIRT4 co-localized with cGAS with or without HSV-1 infection, and SIRT4-cGAS could form puncta after HSV-1 infection, consistent with previously reported results indicating that cGAS exhibits liquid-liquid phase separation (LLPS) with DNA (Shi et al. The EMBO Journal e109272 | 2022, Du & Chen Science 2018).

Fig N4

Fig N4 SIRT4 co-localizes with cGAS

HeLa cells were infected with HSV-1 (MOI=1) for 2 h. Immunofluorescence assays were performed using anti-cGAS (green) and anti-SIRT4 (purple). Nuclei were stained with DAPI. Mitochondria were stained with MitoTracker. Scale bar, 10 μ m.

Fig 1H

Fig 1 SIRT4 interacts with cGAS

(H) HeLa cells were infected with HSV-1 (MOI=1) for 2h. Immunofluorescence assays were performed using anti-cGAS (green) and anti-SIRT4 (purple). Nuclei were stained with DAPI. Mitochondria were stained with MitoTracker. Scale bar, 10 μ m. Arrows indicate cGAS puncta.

Collectively, our data indicate that cGAS and SIRT4 interact in the mitochondria. We propose that during HSV-1 infection, the leakage of mitochondrial DNA recruits cGAS and SIRT4, thereby promoting their assembly and potential phase separation on the mitochondrial surface or around the mitochondrial to orchestrate signal transduction. These findings and the subsequent sentences have been included in the **Results** section.

“In addition, both SIRT4 and cGAS can be found in mitochondria, and SIRT4 co-localized and co-immunoprecipitated with cGAS in mitochondrion, consistent with the report that SIRT4 is a known mitochondrial protein (Li et al., 2018) (Fig 1H-I and Appendix Fig S1B-C). Confocal imaging also revealed that SIRT4-cGAS

could form puncta after HSV-1 infection (Fig 1H), consistent with previously reported results indicating that cGAS exhibits liquid-liquid phase separation (LLPS) with DNA (Du & Chen, 2018; Shi et al, 2022).”

4. The results of Figure 2 and Figure 3 should be combined because they address the same issue, which would make the presentation more concise.

Response: The reviewer`s suggestion is very good and has been well taken. We have combined Figure 2 and Figure 3 as the new Figure 2. Some figures were moved to the new Figure EV1 and EV2.

Fig 2

Fig 2 SIRT4 positively regulates antiviral innate immune responses against DNA virus

(A) HEK293T cells were transfected with HA-SIRT4 and then transfected with control siRNA (SC) or *SIRT4*-specific siRNA (S1, S2, and S3). At 24 h after transfection, the cells were lysed for immunoblot assays.

(B) PMA-THP1 cells were transfected with control siRNA (SC) or *SIRT4*-specific siRNA (S2 and S3). At 24 h after transfection, the cells were infected with HSV-1 (MOI=1) for 24 h. The titers of HSV-1 were determined by standard plaque assays.

(C) PMA-THP1 cells were transfected with control siRNA (SC) or *SIRT4*-specific siRNA (S2 and S3). At 24 h after transfection, the cells were infected with HSV-1 (MOI=1) for 24 h. The supernatants were collected and subjected to ELISA assays.

(D) PMA-THP1 cells were transfected with control siRNA (SC) or *SIRT4*-specific siRNA (S2 and S3). At 24 h after transfection, the cells were infected with HSV-1 (MOI=1) for 8 h. The cells were lysed for Native PAGE or SDS-PAGE assays.

(E) Wild-type (WT) and *SIRT4*-deficient (KO) PMA-THP1 cells were subjected to immunoblot assays.

(F) Wild-type (WT) and *SIRT4*-deficient (KO) PMA-THP1 cells were infected with HSV-1, or VSV (MOI=1) for 24 h. The titers of HSV-1 or VSV were determined by standard plaque assays.

(G-H) Wild-type (WT) and *SIRT4*-deficient (KO) PMA-THP1 cells were infected with HSV-1 (MOI=1) for 24 h. Immunoblot (G) or fluorescence (H) assays were performed. Nuclei were stained with DAPI. Scar bars, 500 μ m.

(I) Wild-type (WT) and *SIRT4*-deficient (KO) PMA-THP1 cells were infected with HSV-1 (MOI=1) for 24 h. The supernatants were collected and subjected to ELISA assays.

(J) Wild-type (WT) and *SIRT4*-deficient (KO) PMA-THP1 cells were infected with HSV-1 (MOI=1) for the indicated periods. Then the cells were fractionated into cytosolic and nuclear subfractions. The immunoblot assays were performed as indicated.

Figure EV1

Figure EV1 SIRT4 expression patterns

(A) PMA-THP1 cells were infected with HSV-1 (MOI=1) (Top) or transfected with HSV60 (1 μ g/ml) (Bottom) for the indicated periods. The cells were then lysed for immunoblot assays.

(B-C) HaCaT keratinocytes were transfected with the empty vector (Vec) or the SIRT4 expressing plasmid for 24 h and then infected with HSV-1 (MOI=1) (B) or transfected with HSV60 (1 μ g/ml) (C) for the indicated periods. The cells were then lysed for immunoblot assays.

(D) HaCaT keratinocytes were transfected with the empty vector (Vec) or the SIRT4

expressing plasmid for 24 h and then infected with HSV-1 (MOI=1) for 8 h. The cells were then lysed for real-time PCR assays.

(E) PMA-THP1 cells were transfected with control siRNA (SC) or *SIRT4*-specific siRNA (S1, S2, and S3). At 24 h after transfection, the cells were lysed for real-time PCR assays.

(F) PMA-THP1 cells were transfected with control siRNA (SC) or *SIRT4*-specific siRNA (S2 and S3). At 24 h after transfection, the cells were transfected with HSV60 (1 µg/ml) for 24 h. The supernatants were collected and subjected to ELISA assays.

(G) PMA-THP1 cells were transfected with control siRNA (SC) or *SIRT4*-specific siRNA (S2). At 24 h after transfection, the cells were infected with HSV-1 (MOI=1) for 8 h. The cells were then lysed for real-time PCR assays.

(H) PMA-THP1 cells were transfected with control siRNA (SC) or *SIRT4*-specific siRNA (S2). At 24 h after transfection, the cells were transfected with HSV60 (1 µg/ml), VACV70 (1 µg/ml), HT DNA (1 µg/ml), poly(dA:dT) (1 µg/ml), or ISD(1 µg/ml) for 8 h. The cells were then lysed for real-time PCR assays.

(I) PMA-THP1 cells were transfected with control siRNA (SC) or *SIRT4*-specific siRNA (S2 and S3). At 24 h after transfection, the cells were transfected with HSV60 (1 µg/ml) for 8 h. Immunoblot assays were then performed as indicated.

(J) PMA-THP1 cells were transfected with control siRNA (SC) or *SIRT4*-specific siRNA (S2). At 24 h after transfection, the cells were infected with HSV-1 (MOI=1) for 8 h. The cells were fractionated into cytosolic and nuclear subfractions. Immunoblot assays were then performed as indicated.

Figure EV2

Figure EV2 SIRT4 deficiency reduces antiviral innate immune responses against cytosolic DNA in PMA-THP1 cells

(A) Wild-type (WT) and *SIRT4*-deficient (KO) PMA-THP1 cells were transfected with HSV60 (1 μ g/ml), VACV70 (1 μ g/ml), HT DNA (1 μ g/ml), or ISD (1 μ g/ml) for 24 h. The supernatants were collected and subjected to ELISA assays.

(B) Wild-type (WT) and *SIRT4*-deficient (KO) PMA-THP1 cells were infected with

HSV-1 (MOI=1) for 8 h. Then the cells were lysed for real-time PCR assays.

(C) Wild-type (WT) and *SIRT4*-deficient (KO) PMA-THP1 cells were transfected with HSV60 (1 µg/ml), VACV70 (1 µg/ml), HT DNA (1 µg/ml), poly(dA:dT) (1 µg/ml), or ISD (1 µg/ml) for 8 h. Then the cells were lysed for real-time PCR assays.

(D) Wild-type (WT) and *SIRT4*-deficient (KO) PMA-THP1 cells were infected with HSV-1 (MOI=1) for the indicated periods. Then Native-PAGE and SDS-PAGE assays were performed.

(E) Wild-type (WT) and *SIRT4*-deficient (KO) PMA-THP1 cells were transfected with HSV60 (1 µg/ml), VACV70 (1 µg/ml), HT DNA (1 µg/ml), ISD (1 µg/ml), or cGAMP (1 µg/ml) for 8 h. Then Native-PAGE and SDS-PAGE assays were performed.

5. The authors mentioned in the last part that the SIRT4 inhibitor regulates autoimmune responses. The reviewer suggests that the authors test the effects of the SIRT4 inhibitor or SIRT4 knockdown/knockout in an animal model, such as an SLE animal model. This work would not take a long time but could increase the reliability of the conclusions.

Response: This is a very good question. Indeed, we are conducting experiments in this area and have obtained some preliminary results. Our current work includes the following four aspects.

1. We have used wild-type and *Sirt4*-deficient mice to establish a TMPD-induced SLE model. TMPD induction is a commonly used mouse model of SLE. We have obtained

some preliminary results using this animal model. For instance, we observed differences in autoimmune responses and disease development between wild-type and *Sirt4*-deficient mice at 2 months and 6 months after TMPD injection. this experiment has only been completed once so far, and the other two groups of mice for replicate experiments have not yet reached the 6-month time point. We need to finish the replicate experiments to make the result solid. So we prefer not to show the preliminary data here.

2. We are breeding SIRT4/cGAS double-deficient mice to investigate the mechanism of SIRT4 in this SLE model. The role of the cGAS-STING signaling pathway in the TMPD-induced SLE model remains controversial. Pan et al. (Autophagy 2023, VOL. 19, NO. 2, 440–456) have reported that UXT1 attenuates the cGAS-STING1 signaling, and its deficiency leads to increased expression of ISGs and signs of exacerbated experimental lupus nephritis in the TMPD-induced murine lupus model. However, Motwani et al. (Front Immunol. 2021 Mar 29) have reported that both cGAS- and STING-deficiency not only fail to rescue mice from TMPD-induced SLE but also result in increased autoantibody production and higher proteinuria levels compared to cGAS/STING-sufficient mice. In addition, Kumpunya et al. (Front Immunol. 2022 Dec 16) have demonstrated that cGAS deficiency enhances inflammasome activation in macrophages and inflammatory pathology in pristane-induced lupus. Therefore, we aim to use SIRT4/cGAS double-deficient mice to further clarify the mechanism by which SIRT4 regulates SLE. However, we still need some time to complete the breeding and

conduct the experiments.

3. In the *Trex1* deficiency-induced SLE model, the roles of cGAS and STING are well-established, which is another aspect of our work. However, the *Trex1*-deficient mouse model requires heterozygous breeding, and even among homozygous individuals, the severity of autoimmunity varies significantly. For therapeutic studies using SIRT4 inhibitors, a large number of mice would be required to achieve statistical significance. We are currently still breeding and accumulating mice for this purpose.

4. Additionally, we are also breeding *Trex1/Sirt4* double-deficient mice, which will require another 6 months to a year to obtain experimental animals.

In summary, although we have some preliminary data from SLE animal models, given the complexity of these models and the long time period (1 to 2 years) to collect all the results, we prefer to develop this aspect as an independent project, serving as a continuation of the current study.

6. The authors need to add the molecular weights of the proteins in their Figures.

Response: The reviewer`s suggestion is very good and has been well taken. We have added molecular weights of the proteins in the Figures.

Referee #3:

The manuscript "SIRT 4 regulates antiviral and autoimmune responses by promoting cGAS mediated signaling pathway" describes the identification of the lysine deacetylase SIRT4 as a novel regulator of cGAS activity. The authors show that SIRT4, but not other sirtuin family members, can remove acetyl-groups from key lysine residues and thus facilitate cGAS activity, both in the human and murine system.

The work presented here is situated in a busy field of research and will be of interest to many cell biology, immunology, and virology researchers. These data contribute to our knowledge of the multitude of molecular mechanisms that govern and regulate cGAS activity. The exploratory experiments by the authors regarding treatment of sterile inflammatory diseases, such as SLE, by inhibition of SIRT4 further extends the readership of the manuscript.

The authors show high-quality data that generally well-support the claims made.

However, a few points need to be addressed:

Major points:

1. In the paper originally describing the acetylation of three lysine residues (K384,

K394 or K414) of human cGAS to regulate its activity (Dai et al. 2019), these authors showed the histone deacetylase HDAC3 to be responsible for deacetylating cGAS in THP1 cells. Ablation of HDAC3 expression via siRNAs prevented de-acetylation on protein level and substantially reduced cGAMP production. This is in direct contrast with the results reported here, where *SIRT4* knockout in THP1 cells reduces cGAMP production to the same extent (compare Fig. S3E in Dai et al. and Fig. 8B this manuscript). The authors ought to in the very least address this discrepancy in their discussion, but really should address this experimentally by repeating experiments using siRNAs against SIRT4 and HDAC3 in comparison. Here it would be good to include transfection agent-only conditions, since lipofectamine treatment alone can have effects on cGAS pathway activation. In addition, the authors should compare the effects of the SIRT4 inhibitor they used with small molecules targeting HDAC3.

Response: The reviewer`s suggestion is very good and has been well taken. We repeated these experiments using siRNAs against SIRT4 and HDAC3 in comparison and found that both siRNAs efficiently prevented the deacetylation of cGAS, reduced cGAMP production, and decreased the activation of innate immune signaling upon HSV60 transfection (Appendix Fig S5A-D). Consistently, both SIRT4 inhibitor and HDAC3 inhibitor RGFP966 exhibited similar effect on cGAS acetylation, cGAMP production, and host defense against virus DNA transfection (Appendix Fig S5E-H). In addition, we used transfection agent-only conditions with only lipofectamine treatment as control, and we observed similar results from lipo and SC group (Appendix Fig S5C-D).

In summary, our findings suggest that both SIRT4 and HDAC3 can deacetylate cGAS and subsequently influence its activity and signaling. This may be attributed to the critical role of cGAS, which necessitates multiple regulatory mechanisms in the organism. Such multi-layered regulation may provide diverse therapeutic targets when developing drugs targeting cGAS. Therefore, during the development of related drugs, it may be helpful to investigate whether the roles of SIRT4 and HDAC3 differ in various pathological processes, as well as the effects and side effects of their inhibitors and agonists, in order to identify the optimal therapeutic agents.

We added these findings to the **Results** Section using the following sentence.

“Histone deacetylase HDAC3 has been reported to be responsible for deacetylating cGAS and regulating its activity (Dai et al., 2019)). Thus, we compared the effect of SIRT4 and HDAC3 on cGAS acetylation and subsequent signaling pathways by knockdown and inhibitor methods respectively, and our findings suggest that both SIRT4 and HDAC3 could deacetylate cGAS and subsequently influence its activity and signaling (Appendix Fig S5A-H).”

We also discussed these findings in the **Discussion** Section using the following sentences.

“In addition, we compared the SIRT4 inhibitor with known inhibitors of cGAS deacetylation, RGFP966 (targeting HDAC3) and aspirin, and found that they have similar effects on regulating cGAS activity and its signaling pathway (Appendix Fig S5). This may be attributed to the critical role of cGAS, which necessitates multiple regulatory mechanisms in the organism. Such multi-layered regulation may provide diverse therapeutic targets when developing drugs targeting cGAS. Therefore, during the development of related drugs, it may be helpful to investigate whether the roles of these inhibitors differ in various pathological processes, including their effects and side effects, in order to identify the optimal therapeutic agents.”

Appendix Fig S5

Appendix Fig S5 SIRT4 inhibitor regulates innate immune responses

(A) PMA-THP1 cells were transfected with control siRNA (SC), *SIRT4*-specific siRNA (S2, siSIRT4), or *HDAC3*-specific siRNA (siHDAC3). At 24 h after transfection, the cells were transfected with HSV60 (1 µg/ml) for 6 h. The cells were lysed and subjected to immunoprecipitation (IP) and immunoblot (IB) assays as indicated.

(B) PMA-THP1 cells were transfected with control siRNA (SC), *SIRT4*-specific siRNA (S2, siSIRT4), or *HDAC3*-specific siRNA (siHDAC3). At 24 h after transfection, the cells were transfected with HSV60 (1 µg/ml) for 24 h. The cells were then collected and the cell lysates were subjected to ELISA analysis.

(C) PMA-THP1 cells were transfected with lipofectamine 2000 (Lipo), control siRNA (SC), *SIRT4*-specific siRNA (S2, siSIRT4), or *HDAC3*-specific siRNA (siHDAC3). At 24 h after transfection, the cells were transfected with HSV60 (1 µg/ml) for the indicated periods. Then the cells were lysed for immunoblot assays.

(D) PMA-THP1 cells were transfected with lipofectamine 2000 (Lipo), control siRNA (SC), *SIRT4*-specific siRNA (S2, siSIRT4), or *HDAC3*-specific siRNA (siHDAC3). At 24 h after transfection, the cells were transfected with HSV60 (1 µg/ml) for 4 h. Then the cells were lysed for real-time PCR assays.

(E) PMA-THP1 cells were treated with DMSO as control, *SIRT4* inhibitor (IN-1, 25 µM), or *HDAC3* inhibitor (RGFP966, 10 µM) for 12 h, and then transfected with HSV60 (1 µg/ml) for 6 h. The cells were lysed and subjected to immunoprecipitation (IP) and immunoblot (IB) assays as indicated.

(F) PMA-THP1 cells were treated with DMSO as control, *SIRT4* inhibitor (IN-1, 25 µM), or *HDAC3* inhibitor (RGFP966, 10 µM) for 12 h, and then transfected with

HSV60 (1 µg/ml) for 24 h. The cells were then collected and the cell lysates were subjected to ELISA analysis.

(G) PMA-THP1 cells were treated with DMSO as control, SIRT4 inhibitor (IN-1, 25 µM), or HDAC3 inhibitor (RGFP966, 10 µM) for 12 h, and then transfected with HSV60 (1 µg/ml) for the indicated periods. Then the cells were lysed for immunoblot assays.

(H) PMA-THP1 cells were treated with DMSO as control, SIRT4 inhibitor (IN-1, 25 µM), or HDAC3 inhibitor (RGFP966, 10 µM) for 12 h, and then transfected with HSV60 (1 µg/ml) for 4 h. Then the cells were lysed for real-time PCR assays.

2. In Figure 4D, Sirt4-KO mice show reduced IFN- β in BALF after HSV-1 infection when compared to WT mice. The same analysis after VSV infection shows very similar results (Fig. 4I). While statistical significance differs, with three data points (should be 4 in 4D?) the results do not allow the conclusion that RNA virus-induced interferon activation is unaffected by SIRT4. The authors need to change their wording and acknowledge that IFN responses to VSV might be affected by SIRT4 as well, albeit indirectly. If these data/samples exist, they should show data on IFN gene expression in tissues and cytokine levels in sera from mice treated with VSV, similar to what is shown in most of Fig. 4 for HSV-1.

Response: The reviewer's suggestion is very good and has been well taken. The results shown in Figure 4I were actually repeated three times, and this is just one of the results.

Due to the limited number of data points, a trend may appear to exist. We have compiled the results from all three replicates in all mice as the new Figure 3I. Furthermore, to further verify this phenomenon, we examined the expression of IFN- β and inflammatory factors in the serum and tissues after SeV infection. The results, as shown in Appendix Fig S2F-H, did not reveal any significant effect of SIRT4 deficiency on the innate immune response in SeV-infected mice. We added the following sentences to the **Results** section to describe these results.

“However, we did not observe significant difference in innate host defense between wild-type and *Sirt4*-deficient mice upon VSV infection (Fig 3I, and Appendix Fig S2F-H), suggesting SIRT4 does not regulate RNA virus-triggered antiviral innate immune responses in mice.”

Fig 3I

Fig 3 SIRT4 protects mice from HSV-1 infection

(I) Wild-type (WT) and *Sirt4*-deficient (KO) mice (n = 9, 8-week-old) were intranasally infected with VSV (5×10^7 PFU) for 24 h. Bronchoalveolar lavage fluid (BALF) was

collected and ELISA assays were performed.

Appendix Fig S2F-H

Appendix Fig 2 SIRT4 protects mice from HSV-1 infection

(F) ELISA of IFN- β , and IL-6 in serum of wild-type (WT) and *Sirt4*-deficient (KO) mice (n=7, 8-week-old) 6 h after intravenous infection with SeV (1×10^7 PFU).

(G, H) Wild-type (WT) and *Sirt4*-deficient (KO) mice (n=10) were intranasally infected with SeV (1×10^7 PFU) for 24 h and then the lungs, livers, and spleens of the mice were subjected to ELISA assays (G) or real-time PCR assays (H).

3. Control experiments, where cells are treated with cGAMP, are only shown for murine cells (Fig. 5). The authors should demonstrate that THP1 cells and HaCaT cells, WT and SIRT4 KO/overexpression, also display unchanged responses to STING agonist treatment.

Response: The reviewer's suggestion is very good and has been well taken. We

examined the role of SIRT4 in THP1 and HaCaT cells upon cGAMP treatment. As shown in Appendix Fig S1D, we found that overexpression of SIRT4 in HaCaT cells enhanced the activation of the innate immune response induced by HSV60, but did not significantly affect the response triggered by cGAMP. Similarly, consistent with the previous findings in EV2C (now EV2E in the new version) of the manuscript, knockout of SIRT4 in THP1 cells attenuated HSV60-induced innate immune activation, while having no obvious impact on the response elicited by cGAMP (Fig N5). We also evaluated the effects under short-term stimulation and observed similar outcomes after 1 or 2 hours of cGAMP treatment (Fig N2A-B). To further validate these findings, we tested the responses in HeLa cells and again found that SIRT4 potentiated the activation of the innate immune response triggered by HSV60, but not by cGAMP (Appendix Fig S1E-F). The following statements have been added in the **Results** section to describe these results.

“However, SIRT4 overexpression did not significantly affect the response triggered by cGAMP (Appendix Fig S1D-F).”

“However, in both human and mouse cells, cGAMP- or RNA viruses-triggered innate immune responses were not affected much by the impairment of SIRT4 expression (Fig 4A, 4F-G, EV2E, EV3B, EV3D, EV3G, and Appendix Fig S1D-F, and S3C-D), suggesting that SIRT4 specifically affected DNA virus-induced signaling pathway, and its target might be in the upstream of cGAMP.”

Appendix Fig S1D-F

Appendix Fig S1

(D) HaCaT keratinocytes were transfected with the empty vector (Vec) or the SIRT4 expressing plasmid for 24 h and then transfected with HSV60 (1μg/ml) or cGAMP (1μg/ml) for 4 h. The cells were then lysed for immunoblot assays.

(E) HeLa cells were transfected with the empty vector (Vec) or the SIRT4 expressing plasmid for 24 h and then transfected with HSV60 (1μg/ml) or cGAMP (1μg/ml) for the indicated periods. The cells were then lysed for immunoblot assays.

(F) HeLa cells were transfected with the empty vector (Vec) or the SIRT4 expressing plasmid for 24 h and then transfected with HSV60 (1μg/ml) or cGAMP (1μg/ml) for 4 h. The cells were then lysed for real-time PCR assays.

Fig EV2E

Figure EV2 *SIRT4* deficiency reduces antiviral innate immune responses against cytosolic DNA in PMA-THP1 cells

(E) Wild-type (WT) and *SIRT4*-deficient (KO) PMA-THP1 cells were transfected with HSV60 (1 μ g/ml), VACV70 (1 μ g/ml), HT DNA (1 μ g/ml), ISD (1 μ g/ml), or cGAMP (1 μ g/ml) for 8 h. Then Native-PAGE and SDS-PAGE assays were performed.

Fig N2

Fig N2 SIRT4 did not significantly affect the response triggered by cGAMP

(A) HaCaT keratinocytes were transfected with the empty vector (Vec) or the SIRT4 expressing plasmid for 24 h and then transfected with HSV60 (1 μg/ml) or cGAMP (1 μg/ml) for the indicated periods. The cells were then lysed for immunoblot assays.

(B) Wild-type (WT) and *SIRT4*-deficient (KO) PMA-THP1 cells were transfected with HSV60 (1 μg/ml) or cGAMP (1 μg/ml) for the indicated periods. The cells were then lysed for immunoblot assays.

Fig N5

Fig N5 SIRT4 did not significantly affect the response triggered by cGAMP

Wild-type (WT) and *SIRT4*-deficient (KO) PMA-THP1 cells were transfected with HSV60 (1 µg/ml), or cGAMP (1 µg/ml) for 8 h. The cells were then lysed for immunoblot assays.

Minor points:

4. Related to point 1, Dai et al. used treatment of cells with aspirin to enforce increased cGAS acetylation and thus reduced cGAS activity. It would be very informative if the authors could compare the SIRT4 inhibitor they used with aspirin treatment in the context of autoimmune responses (Fig. 11).

Response: The reviewer's suggestion is very good and has been well taken. We compared SIRT4 inhibitor with aspirin in *Trex1*-deficiency-induced autoimmune

responses and found that they had similar effects on autoimmune responses (Appendix Fig S5I-K). We added the following sentences to the **Results** section to describe these results.

“It has been reported that aspirin can directly acetylate cGAS and effectively suppress self-DNA-induced autoimmunity. Thus, we compared the role of SIRT4 inhibitor with aspirin in *Trex1*-deficient PMs and BMDCs, and found that they had similar effects on autoimmune responses (Appendix Fig S5I-K).”

We also discussed these findings in the **Discussion** Section using the following sentences.

“In addition, we compared the SIRT4 inhibitor with known inhibitors of cGAS deacetylation, RGFP966 (targeting HDAC3) and aspirin, and found that they have similar effects on regulating cGAS activity and its signaling pathway (Appendix Fig S5). This may be attributed to the critical role of cGAS, which necessitates multiple regulatory mechanisms in the organism. Such multi-layered regulation may provide diverse therapeutic targets when developing drugs targeting cGAS. Therefore, during the development of related drugs, it may be helpful to investigate whether the roles of these inhibitors differ in various pathological processes, including their effects and side effects, in order to identify the optimal therapeutic agents.”

Appendix Fig S5I-K

Appendix Fig S5 SIRT4 inhibitor regulates innate immune responses

(I, J) Wild-type (WT) and *Trex1*-deficient (KO) PMs (I) or BMDCs (J) were treated with SIRT4 inhibitor SIRT4-IN-1 (25 μ M) or aspirin (4 mM) for 12 h. Then the cells were lysed for immunoblot assays.

(K) Wild-type (WT) and *Trex1*-deficient (KO) BMDCs were treated with SIRT4 inhibitor SIRT4-IN-1 (25 μ M) or aspirin (4 mM) for 12 h. Then the cells were lysed for real-time PCR assays.

5. The data in Fig. 8A and 8B are not referenced anywhere in the results text.

Response: The reviewer's suggestion is very good and has been well taken. Sorry for the mistake. After submission, we revisited the cGAMP measurement system and made improvements based on the method described by Wu et al (Science 2013). The detailed procedure has been included in the **Materials and Methods** section. With this

optimized approach, we obtained more robust results, which are now presented as new Fig 5A, 7C, Appendix Fig S5B, and Appendix Fig S5F. The following statement has been added to the **Results** section to describe this update.

Materials and Methods

cGAMP quantification

For the measurement of 2'3'-cGAMP, a commercial ELISA kit based on the competition between 2'3'-cGAMP and 2'3'-cGAMP-HRP was used (Catalog No. 501700, Cayman). Briefly, PMA-THP1 cells were seeded in 6-well plate. After HSV60 transfection or HSV-1 infection, cell pellets were lysed using a cell homogenization buffer (10 mM Tris-HCl, pH 7.4; 10 mM KCl; 1.5 mM MgCl₂) and centrifuged at 13,000 rpm for 20 minutes. The supernatant was then heated at 95 °C for 5 minutes and centrifuged again at 12,000 rpm for 10 minutes to remove denatured proteins. The supernatant was collected for cGAMP quantification according to the manufacturer's instructions.

“Next, we examined whether SIRT4 regulated the production of cGAMP, which is synthesized by cGAS. As shown in Fig 5A, SIRT4 deficiency impaired HSV-1-triggered production of cGAMP, suggesting that SIRT4 promotes the synthesis of cGAMP by cGAS.”

“Additional, SIRT4-IN-1 treatment inhibited the synthesis of cGAMP triggered by HSV60 transfection (Fig 7C).”

Histone deacetylase HDAC3 has been reported to be responsible for deacetylating cGAS and regulating its activity (Dai et al., 2019). Thus, we compared the effect of SIRT4 and HDAC3 on cGAS acetylation and subsequent signaling pathways by knockdown and inhibitor methods respectively, and our findings suggest that both SIRT4 and HDAC3 could deacetylate cGAS and subsequently influence its activity and signaling (Appendix Fig S5A-H).

Fig 5A

Fig 5 SIRT4 promotes the activation of cGAS

(A) Wild-type (WT) and *SIRT4*-deficient (KO) PMA-THP1 cells were infected with HSV-1 (MOI=1) for 24 h. The cells were then collected and the cell lysates were subjected to ELISA analysis.

Fig 7C

Fig 7 SIRT4 inhibitor regulates DNA virus- or viral DNA-triggered innate immune responses

(C) PMA-THP1 cells were treated with SIRT4 inhibitor (0, 25 μ M) for 12 h, and then transfected with HSV60 (1 μ g/ml) for 24 h. The cells were then collected and the cell lysates were subjected to ELISA analysis.

Appendix Fig S5

Appendix Fig S5 SIRT4 inhibitor regulates innate immune responses

(B) PMA-THP1 cells were transfected with control siRNA (SC), *SIRT4*-specific siRNA (S2, siSIRT4), or *HDAC3*-specific siRNA (siHDAC3). At 24 h after transfection, the

cells were transfected with HSV60 (1 µg/ml) for 24 h. The cells were then collected and the cell lysates were subjected to ELISA analysis.

(F) PMA-THP1 cells were treated with DMSO as control, SIRT4 inhibitor (IN-1, 25 µM), or HDAC3 inhibitor (RGFP966, 10 µM) for 12 h, and then transfected with HSV60 (1 µg/ml) for 24 h. The cells were then collected and the cell lysates were subjected to ELISA analysis.

6. In Figure 9I, overexpression of WT cGAS in the absence of SIRT4 leads to phosphorylation of IRF3 (i.e. PRR pathway activation), while expression of the 3KR mutant does not. In contrast, indistinguishable *IFNB1* transcript levels can be observed for both constructs in Fig. 9J. Could the authors comment on this discrepancy?

Response: The reviewer's suggestion is very good and has been well taken. The results presented in Fig 9I and 9J were reproduced more than three times. Upon reviewing all previous replicates, we found that in all other repetitions, 3KR mutant did not lead to the phosphorylation of IRF3 or significantly activate *IFNB1* transcript levels. Therefore, the *IFNB1* transcript level data shown in Fig 9J (**the new Fig 6J**) is not representative. Consequently, we have replaced this panel with a set of results from a different experiment that we believe is more representative of our overall findings.

Fig 6

Fig 6 SIRT4 deacetylates cGAS

(J) *cGAS*-deficient HeLa cells were transfected with indicated plasmids for 24 h and then the cell lysates were subjected to real-time PCR assays.

7. The manuscript would benefit from some grammatical and semantic proof-reading to ease understanding in certain instances. The authors should make sure that they refer to SIRT4 as a "deacetylase" and not "deacylase" as these are two distinct processes. In addition, the authors should be consistent in correctly referring to human and mouse gene names and italicize where necessary. Many figure panels that describe mouse data have gene names in all capital letters (i.e. human). Where mouse gene names are used, they should start with a capital letter.

Response: The reviewer's suggestion is very good and has been well taken. We have changed "deacylase" to "deacetylase" referring to SIRT4. We have checked and revised

for all the gene names. The manuscript were grammatical and semantic proof-reading by a native speaker and several sentences were rewritten.

8. The figure legends need to be more precise when describing how many times experiments were performed independently, and whether technical replicates of one experiment or pooled data are shown. What is written under statistics in the methods section is not always congruent with figure legends. Regarding mouse experiments, were they repeated? If yes, the authors should show data from all individual animals.

Response: The reviewer`s suggestion is very good and has been well taken. Yes, all the mouse experiments were repeated. We have combined all the repeats together to show data from all individual animals in Fig 3C-G, Fig 3I, Fig 7I-K, EV 5F-H, and Appendix Fig S2D-H. We have added **Data information** to each figure legend to describe the biological and technical replicates.

We would like to specifically note that in the previous Fig 4C, we presented data on TNF- α levels in mouse sera. However, during repeated experiments, the TNF- α values were found to be very close to the lower detection limit of the assay kit, resulting in undetectable TNF- α levels in some mouse samples (Fig N6). This is also the reason why we discontinued measuring this indicator in both sera and bronchoalveolar lavage fluid (BALF) in subsequent inhibitor-related animal experiments. Although statistical analysis of TNF- α values from all mice still showed significant differences, we

considered these results insufficiently robust. Therefore, to ensure greater data reliability and consistency, we have removed the TNF- α results from the new Figures 3C and 3D.

Fig N6

Fig N6 SIRT4 protects mice from HSV-1 infection

ELISA of IFN- β , TNF- α , and IL-6 in serum of wild-type (WT) and *Sirt4*-deficient (KO) mice (n=10, 8-week-old) 6 h after intravenous infection with HSV-1 (1×10^7 PFU).

9. The authors ought to describe their methodology in more detail. For many assays, only a reference but no technical detail is provided. This is not sufficient for being able to reproduce the data shown here. For the RT-qPCR primers used in the study, a reference is missing entirely.

Response: The reviewer's suggestion is very good and has been well taken. We have described the methodology in more detail in the **Materials and Methods** sections for Preparations of MEFs, Immunoprecipitation and immunoblot analysis, IRF3 dimerization assay, *In situ* PLA, Nuclear Extracts, cGAMP quantification, and *In vitro*

deacetylation assay. We have added the reference to the RT-qPCR primers. Also, we added Appendix Table S1 to show all the primers used.

10. In the introduction, IKK-epsilon is referred to as IKKi, whereas it is called IKKe in the graphical abstract. This should be consistent.

Response: The reviewer's suggestion is very good and has been well taken. We have changed IKKi to IKK ϵ in the **Introduction** section in the following sentence.

“Activated STING serve as a platform to recruit and activate TANK-binding kinase 1 (TBK1) and IKK ϵ , leading to the phosphorylation and activation of the transcription factor IFN regulatory factor 3 (IRF3) and NF-kB, thus inducing the expression of IFN-I and proinflammatory factors.”

Dear Prof. Wang,

Thank you for the submission of your revised manuscript to our editorial offices. I have now received the reports from the three referees that I asked to re-evaluate the study, you will find below. As you will see, the referees now fully support publication of your study in EMBO reports. Referee #1 has two further requests, I ask you to address in a final revised manuscript. Please also provide a final p-b-p-response regarding the remaining referee points and the editorial requests.

Editorial requests:

- I would suggest this slightly modified title:

SIRT4 regulates antiviral and autoimmune responses by promoting cGAS-mediated signaling pathways

- Please provide the abstract written in present tense throughout.

- Please add a title to the reference section and reduce the number of keywords to five. Please order the manuscript sections like this, using only these names:

Title page - Abstract (max. 175 words) - Keywords (up to five) - Introduction - Results - Discussion - Methods - Data availability section - Acknowledgements (please include here all the funding information) - Disclosure and Competing Interests Statement - References - Figure legends - Expanded View Figure legends

- Please check again that the number "n" for how many independent experiments were performed, their nature (biological versus technical replicates), the bars and error bars (e.g. SEM, SD) and the test used to calculate p-values is indicated in the respective figure legends (main, EV and Appendix figures). Please also check that all the p-values are explained in the legend, and that these fit to those shown in the figure. Please provide statistical testing where applicable. Please avoid the phrase 'independent experiment' but clearly state if these were biological or technical replicates. Please also indicate (e.g. with n.s.) if testing was performed, but the differences are not significant. In case n=2, please show the data as separate datapoints without error bars and statistics. See also:

<https://link.springer.com/journal/44319/submission-guidelines#cms-Figure-and-data-presentation>

If n<5, please show single datapoints for diagrams. Moreover:

- Please note that the exact p values are not provided in the legends of figures 2b,c,f,i; 3b-g; 4a-d; 5a; 6g,j; 7c,d,f,h,l,j,k; 8a,b,d,e,f; EV-1d-h; EV-2a-c; EV-3a-d; EV-4c,d; EV-5a,c,e-h; Appendix Fig-S1f; Appendix Fig-S2d; Appendix Fig-S3a,b,c; Appendix Fig-S5b,d,f,h,k.

- Please add scale bars of similar style and thickness to all microscopic images (main, EV and Appendix figures), using clearly visible black or white bars (depending on the background). Please place these in the lower right corner of the images themselves. Please do not write on or near the bars in the image but define the size in the respective figure legend. Presently, many scale bars are rather hard to see. Please improve.

- Please add all sequence and primer information to the Reagents & Tools Table and remove this from the Methods. Please add call outs to the Reagents & Tools table where appropriate. Finally, please remove the instructions from the final R&T table.

- Please provide the Appendix file as pdf. Moreover, please add a title to the first page of the Appendix ('Appendix for ...'), but do not put author names again.

- Please confirm that for all Western blot panels (main, EV, or Appendix figures) the loading control was run on the same gel as the other proteins detected. Please note that we discourage comparisons between samples on different gels/blots, even if the samples derive from one experiment, as confounding factors reduce comparability. If unavoidable, the figure legend must state that the samples derive from the same experiment and that gels/blots were processed in parallel. If a 'representative' loading control is shown for multiple gels/blots, the intra-gel controls should be shown in the source data files, and the figure legends should describe the data displayed accurately. See our author guidelines:

<https://link.springer.com/journal/44319/submission-guidelines#cms-Figure-and-data-presentation> (section 'Electrophoretic gels and blots').

In addition, I would need from you uploaded separately:

- a short, two-sentence summary of the manuscript (not more than 35 words).

- two to four short (!) bullet points highlighting the key findings of your study (two lines each).

- a schematic summary figure as separate file that provides a sketch of the major findings (not a data image) in jpeg or tiff format (with the exact width of 550 pixels and a height of not more than 400 pixels) that can be used as a visual synopsis on our website.

I look forward to seeing the further revised version of your manuscript when it is ready. Please let me know if you have questions regarding the revision.

Best,

Referee #1:

Overall, the authors have addressed most of my previous comments satisfactorily. I have only two remaining points:

Appendix S2E and S4D: The authors addressed the initial request by adding CD68 staining, but without quantification the staining is not interpretable. Please add quantitative analysis of CD68⁺ cells/area. The CD68 staining protocol should also be included in the Methods.

Appendix Fig S1A and Appendix Fig S4C:

In the rebuttal, the authors acknowledge SIRT2-cGAS interaction and deacetylation by SIRT2 and SIRT5 after HSV-1 infection. However, the Results still state that ONLY SIRT4 binds and deacetylates cGAS, which is inconsistent with their own data. The text should be revised to reflect these observed but weaker effects from SIRT2/SIRT5 instead of presenting them as absent.

I would be able to recommend the manuscript for publication once these points are addressed.

Referee #2:

The authors addressed my concerns.

Referee #3:

The authors provide an impressive number of new experiments/data, that addressed all my comments and substantially improved the manuscript.

Editorial requests:

- I would suggest this slightly modified title:

SIRT4 regulates antiviral and autoimmune responses by promoting cGAS-mediated signaling pathways

Response: We fully agree with this revision. We have already made the modifications according to the stated requirements.

- Please provide the abstract written in present tense throughout.

Response: We fully agree with this revision. We have already made the modifications according to the stated requirements.

- Please add a title to the reference section and reduce the number of keywords to five.

Please order the manuscript sections like this, using only these names:

Title page - Abstract (max. 175 words) - Keywords (up to five) - Introduction - Results

- Discussion - Methods - Data availability section - Acknowledgements (please include here all the funding information) - Disclosure and Competing Interests Statement -

References - Figure legends - Expanded View Figure legends

Response: We fully agree with this revision. We have already made the modifications

according to the stated requirements.

- Please check again that the number "n" for how many independent experiments were performed, their nature (biological versus technical replicates), the bars and error bars (e.g. SEM, SD) and the test used to calculate p-values is indicated in the respective figure legends (main, EV and Appendix figures). Please also check that all the p-values are explained in the legend, and that these fit to those shown in the figure. Please provide statistical testing where applicable. Please avoid the phrase 'independent experiment' but clearly state if these were biological or technical replicates. Please also indicate (e.g. with n.s.) if testing was performed, but the differences are not significant. In case n=2, please show the data as separate datapoints without error bars and statistics.

See also:

<https://link.springer.com/journal/44319/submission-guidelines#cms-Figure-and-data-presentation>

If n<5, please show single datapoints for diagrams. Moreover:

- Please note that the exact p values are not provided in the legends of figures 2b,c,f,i; 3b-g; 4a-d; 5a; 6g,j; 7c,d,f,h,I,j,k; 8a,b,d,e,f; EV-1d-h; EV-2a-c; EV-3a-d; EV-4c,d; EV-5a,c,e-h; Appendix Fig-S1f; Appendix Fig-S2d; Appendix Fig-S3a,b,c; Appendix Fig-S5b,d,f,h,k.

Response: We fully agree with and comply with this requirement. We have checked all

the items and provided the exact p values in the legends of the figures that were mentioned above.

- Please add scale bars of similar style and thickness to all microscopic images (main, EV and Appendix figures), using clearly visible black or white bars (depending on the background). Please place these in the lower right corner of the images themselves. Please do not write on or near the bars in the image but define the size in the respective figure legend. Presently, many scale bars are rather hard to see. Please improve.

Response: We fully agree with and comply with this requirement. We have checked all the image figures and improve the bars according to these instructions.

- Please add all sequence and primer information to the Reagents & Tools Table and remove this from the Methods. Please add call outs to the Reagents & Tools table where appropriate. Finally, please remove the instructions from the final R&T table.

Response: We fully agree with and comply with this requirement. We have prepared the Reagents & Tools Table following the instructions and added it to the Methods.

- Please provide the Appendix file as pdf. Moreover, please add a tile to the first page of the Appendix ('Appendix for ...'), but do not put author names again.

Response: We fully agree with and comply with this requirement. We have prepared and provided the Appendix file as pdf according to the instructions.

- Please confirm that for all Western blot panels (main, EV, or Appendix figures) the loading control was run on the same gel as the other proteins detected. Please note that we discourage comparisons between samples on different gels/blots, even if the samples derive from one experiment, as confounding factors reduce comparability. If unavoidable, the figure legend must state that the samples derive from the same experiment and that gels/blots were processed in parallel. If a 'representative' loading control is shown for multiple gels/blots, the intra-gel controls should be shown in the source data files, and the figure legends should describe the data displayed accurately.

See our author guidelines:

<https://link.springer.com/journal/44319/submission-guidelines#cms-Figure-and-data-presentation> (section 'Electrophoretic gels and blots').

Response: We fully agree with and comply with this requirement. We have checked and confirmed all the WB panels.

In addition, I would need from you uploaded separately:

- a short, two-sentence summary of the manuscript (not more than 35 words).

Responses: SIRT4 deacetylates cGAS and promotes cGAS-mediated antiviral and autoimmune responses. SIRT4 inhibitors may represent a promising therapeutic strategy for managing DNA virus infections and modulating autoimmune responses.

- two to four short (!) bullet points highlighting the key findings of your study (two lines each).

Highlights:

1 SIRT4 interacts with cGAS deacetylates cGAS, thus regulating DNA virus- or cytoplasmic DNA-triggered innate immune responses.

2 SIRT4 deficiency or SIRT4 inhibitor treatment results in impaired antiviral innate immune signaling against DNA virus or cytoplasmic DNA *in vitro* and *in vivo*.

3 SIRT4 inhibitor attenuates the type I IFN signaling response in both *Trex1*-deficient cells and the PBMCs from SLE patients.

- a schematic summary figure as separate file that provides a sketch of the major findings (not a data image) in jpeg or tiff format (with the exact width of 550 pixels and a height of not more than 400 pixels) that can be used as a visual synopsis on our website.

Responses: Appendix Fig S6 is a schematic summary figure that provides a sketch of the major findings, only much bigger. We provided the figure with the exact size and we will cancel the Appendix Fig S6 if needed.

Referee #1:

Overall, the authors have addressed most of my previous comments satisfactorily. I have only two remaining points:

Appendix S2E and S4D: The authors addressed the initial request by adding CD68 staining, but without quantification the staining is not interpretable. Please add quantitative analysis of CD68⁺ cells/area. The CD68 staining protocol should also be included in the Methods.

Responses: The reviewer's suggestion is very good and has been well taken. We have added the quantitative analysis results of CD68⁺ cells/area into Appendix S2E and S4D.

We have added the CD68 staining protocol to the **Methods** section.

Appendix Fig S2E

Appendix Fig S2 SIRT4 protects mice from HSV-1 infection

(E) Sex and age-matched wild-type (WT) and *Sirt4*-deficient (KO) mice (n=5) were intravenously infected with HSV-1 (1×10^7 PFU) for 24 h and lung sections were analyzed by CD68 staining (Left). Quantification of CD68⁺ cells was shown in the right panel. Scale bar, 100 μ m.

Appendix Fig S4D

Appendix Fig S4 SIRT4 deacetylates cGAS

(D) Wild-type (WT) mice were treated with SIRT4 inhibitor SIRT4-IN-1 (0, 10 mg/kg body weight) or control DMSO (n=5), and then intranasally infected with HSV-1 (1×10^7 PFU) for 24 h. The lung sections were analyzed by CD68 staining (Left). Quantification of CD68⁺ cells was shown in the right panel. Scale bar, 100 μ m. Scale bar, 100 μ m.

Tissue section staining

The tissue sections were prepared and stained according to experimental SOP of Servicebio. Briefly, paraffin-embedded sections were dewaxed, hydrated, and antigen-retrieved. Endogenous peroxidase was blocked with 3% H₂O₂. After serum blocking, sections were incubated overnight with primary CD68 antibody at 4 °C, followed by HRP-conjugated secondary antibody. DAB was used for visualization. For H&E staining, separate sections were stained with hematoxylin and eosin. All slides were then dehydrated, cleared, and mounted with neutral gum for microscopic examination. For each slide, five non-overlapping fields with the highest density of positive staining were selected at high-power magnification (400x) and CD68-positive cells were manually counted by a trained observer blinded to the sample groups to obtain the average value.

Appendix Fig S1A and Appendix Fig S4C:

In the rebuttal, the authors acknowledge SIRT2-cGAS interaction and deacetylation by SIRT2 and SIRT5 after HSV-1 infection. However, the Results still state that ONLY

SIRT4 binds and deacetylates cGAS, which is inconsistent with their own data. The text should be revised to reflect these observed but weaker effects from SIRT2/SIRT5 instead of presenting them as absent.

Responses: The reviewer`s suggestion is very good and has been well taken. We have revised the description of these results in **Results** section and **Discussion** section.

Results

“As shown in Fig 1B-C and Appendix Fig S1A, SIRT4 co-immunoprecipitated with cGAS. It seemed that under unstimulated conditions, SIRT2 interacted very weakly to endogenous cGAS, but after HSV-1 stimulation, this binding became even less detectable (Appendix Fig S1A).”

“We investigated whether other SIRTs affected acetylation of endogenous cGAS and found that SIRT4 effectively reduced the acetylation of endogenous cGAS under unstimulated conditions (Appendix Fig S4C). After HSV-1 infection, SIRT2 and SIRT5 appeared to reduce cGAS acetylation to some extent, but the extent of deacetylation by SIRT4 was more significantly (Appendix Fig S4C).”

Discussion

“In addition, coimmunoprecipitation assays indicate SIRT4 interacts with cGAS.”

Dear Prof. Wang,

Thank you for the submission of your further revised manuscript to our editorial offices. I now went through this and your p-b-p-response and consider the remaining points of referee #1 as adequately addressed. However, before I can proceed with formal acceptance, I have these further editorial requests I ask you to address in a final revised manuscript:

- Please note that corresponding authors are required to supply an ORCID ID upon submission of a revised manuscript. Please do this for co-corresponding author Yinming Liang. The ORCID needs to be added by the author to his profile in the EMBO reports submission portal. This can't be done from our side.
- Please add scale bars to the images in Fig. 3H, left panels. Please do not write on or near the bars in the image but define the size in the respective figure legend.
- Most figures are extremely crowded, in particular Figs. 1, 3, 4, 6, 7, EV1, EV3 and EV4, and some of the smaller panels won't show very well in the final online version of the paper. Please re-assemble the figures, remove some data, and move these to the Appendix (as further Appendix Figures or adding the information to existing Appendix figures). Please then also update all the callouts in the manuscript text accordingly.

<https://media.springernature.com/original/springer-cms/rest/v1/content/27825798/data/v1>

- You state in many of the data information sections: 'Data shown are from at least three independent biological replicates.' Please be more specific. Please indicate clearly how many replicates (technical or biological) were used for the data in each panel. For the Western blot panels please state that one representative (?) replicate experiment is shown (see also below).
- For panels 2B, 2F, 4B, 4D and 4F you indicate that the data shown are from two independent biological replicates. Nevertheless, 3 data points are shown and statistics is provided. Please check. In case n is indeed 2, then please show the data as separate datapoints without error bars and statistics in these panels.
- Many of the data information sections just start with: ',Two-tailed unpaired Student's t test'. Please add more information. What was calculated using this test and in which panel.
- Please remove the sentence ',Source data for this figure are available online.' from the figure legends. The source data will be linked to the figures so this will be obvious for the readers.
- I am not convinced that in all cases the Western blots shown in each panel (main, EV and Appendix figures) derive from the same gel. Please confirm again that for all Western blot panels shown the loading control was run on the same gel as the other proteins detected and shown in the panel. If this is not the case, please state in the figure legend that the samples derive from the same experiment and that gels/blots were processed in parallel. If a 'representative' loading control is shown, then show the intra-gel controls in the source data files and describe this accurately in figure legends. See our author guidelines:

<https://link.springer.com/journal/44319/submission-guidelines#cms-Figure-and-data-presentation> (section 'Electrophoretic gels and blots').

Finally, please add a detailed description for the Western blot procedure to the Methods section.

- Please also have your final manuscript file carefully proofread by a native speaker.

Please contact me in case you have questions regarding the revision.

Best,

Dear editor of *EMBO reports* :

Thank you very much for giving us a chance to revise our paper. Attached please find our revised manuscript, entitled “SIRT4 regulates antiviral and autoimmune responses by promoting cGAS-mediated signaling pathways” to *EMBO reports*.

We have had the manuscript proofread by a native English speaker, who performed comprehensive revisions to grammar and spelling throughout the article. In the figure legends, we have provided detailed descriptions for each panel regarding biological replicates, technical replicates, whether the images are representative, and the statistical methods used. For all Western blot figures, we have verified the original raw data to confirm that any directly compared data originate from the same membrane. For all internal loading controls (e.g., p-TBK1 relative to total TBK1), we confirm the results were obtained from the same membrane after stripping. For certain WB figures (e.g., those displaying both Native PAGE and SDS-PAGE), we have stated that all samples derive from a single, concurrent experiment. Most figures have been reformatted to reduce visual clutter and improve readability. We have uploaded the Source Data files for the supplementary figures (Appendix figures) to facilitate any future verification. Throughout this revision, in addition to addressing all of the editor's comments, we have also consulted several recently published papers in your journal as a reference for style. All changes in the main manuscript text are highlighted in yellow. As the submitted Appendix file is a PDF, we have uploaded the clean version. If required, we are happy to provide a version with the tracked changes marked.

As this round of revisions focused primarily on formatting, I did not update the point-by-point response to the reviewers' comments. However, I would be pleased to provide it upon request.

Thank you very much for helping Professor Yinming Liang resolve his account and ORCID issues. We appreciate the editor's helpful comments, which helped us improve

the manuscript. We hope that our revised manuscript will now be acceptable for publication in *EMBO reports* and share our data with the readers of the Journal. Thanks for your consideration and kind help.

Best Regards, and Happy New Year,

Jie Wang

Professor

Xinxiang Medical University

601 Jinsui Road, Xinxiang

Henan Province, China

Prof. Jie Wang
Xinxiang Medical University, Xinxiang
School of Medical Technology
601 Jinsui Road
Xinxiang, Henan 453003
China

Dear Prof. Wang,

I am very pleased to accept your manuscript for publication in the next available issue of EMBO reports. Thank you for your contribution to our journal.

You may qualify for financial assistance for your publication charges - either via a Springer Nature fully open access agreement or an EMBO initiative. Check your eligibility: <https://link.springer.com/journal/44319/how-to-publish-with-us>

Yours sincerely,

>>> Please note that it is EMBO Reports policy for the transcript of the editorial process (containing referee reports and your response letter) to be published as an online supplement to each paper. If you do NOT want this, you will need to inform the Editorial Office via email immediately. More information is available here: <https://link.springer.com/partners/embo-press/editorial-policies#Peer%20review>